# A missense in HSF2BP causing primary ovarian insufficiency affects meiotic recombination by its novel interactor C19ORF57/BRME1

Natalia Felipe-Medina[1†], Sandrine Caburet[2,3†], Fernando Sánchez-Sáez[1], Yazmine B Condezo[1], Dirk G de Rooij[4], Laura Gómez-H[1], Rodrigo Garcia-Valiente[1], Anne Laure Todeschini[2,3], Paloma Duque[1], Manuel Adolfo Sánchez-Martin[5,6], Stavit A Shalev[7,8], Elena Llano[1,9], Reiner A Veitia[2,3,10]*, Alberto M Pendás[1]*

[1]Molecular Mechanisms Program, Centro de Investigación del Cáncer and Instituto de Biología Molecular y Celular del Cáncer (CSIC-Universidad de Salamanca), Salamanca, Spain; [2]Université de Paris, Paris Cedex, France; [3]Institut Jacques Monod, Université de Paris, Paris, France; [4]Reproductive Biology Group, Division of Developmental Biology, Department of Biology, Faculty of Science, Utrecht University, Utrecht, Netherlands; [5]Transgenic Facility, Nucleus platform, Universidad de Salamanca, Salamanca, Spain; [6]Departamento de Medicina, Universidad de Salamanca, Salamanca, Spain; [7]The Genetic Institute, "Emek" Medical Center, Afula, Israel; [8]Bruce and Ruth Rappaport Faculty of Medicine, Technion, Haifa, Israel; [9]Departamento de Fisiología y Farmacología, Universidad de Salamanca, Salamanca, Spain; [10]Université Paris-Saclay, Institut de Biologie F. Jacob, Commissariat à l'Energie Atomique, Fontenay aux Roses, France

*For correspondence:
reiner.veitia@ijm.fr (RAV);
amp@usal.es (AMP)

†These authors contributed equally to this work

Competing interests: The authors declare that no competing interests exist.

**Abstract** Primary Ovarian Insufficiency (POI) is a major cause of infertility, but its etiology remains poorly understood. Using whole-exome sequencing in a family with three cases of POI, we identified the candidate missense variant S167L in *HSF2BP*, an essential meiotic gene. Functional analysis of the HSF2BP-S167L variant in mouse showed that it behaves as a hypomorphic allele compared to a new loss-of-function (knock-out) mouse model. *Hsf2bp*[S167L/S167L] females show reduced fertility with smaller litter sizes. To obtain mechanistic insights, we identified C19ORF57/BRME1 as a strong interactor and stabilizer of HSF2BP and showed that the BRME1/HSF2BP protein complex co-immunoprecipitates with BRCA2, RAD51, RPA and PALB2. Meiocytes bearing the HSF2BP-S167L variant showed a strongly decreased staining of both HSF2BP and BRME1 at the recombination nodules and a reduced number of the foci formed by the recombinases RAD51/DMC1, thus leading to a lower frequency of crossovers. Our results provide insights into the molecular mechanism of HSF2BP-S167L in human ovarian insufficiency and sub(in)fertility.

## Introduction

The process of gametogenesis is one of the most complex and highly regulated differentiation programs. It involves a unique reductional cell division, known as meiosis, to generate highly specialized cells: the gametes. Indeed, the outcome of meiosis is the production of oocytes and spermatozoa, which are the most distinctive cells of an adult organism and are essential for the faithful transmission of the genome across generations.

The meiotic division is an orderly process that results in the pairing and synapsis of homologous chromosomes and crossover (CO) formation, which ultimately enable homologous chromosomes segregation (*Hunter, 2015*; *Loidl, 2016*; *Zickler and Kleckner, 2015*). In mammals, pairing of homologs is dependent on the repair of self-induced double-strand breaks (DSBs) during prophase I by homologous recombination (*Handel and Schimenti, 2010*) and it leads to the intimate alignment of homologous chromosomes (synapsis) through the zipper-like synaptonemal complex (SC) (*Cahoon and Hawley, 2016*). The SC is a proteinaceous tripartite structure that provides the structural framework for DSBs repair (*Baudat et al., 2013*), as epitomized by the tight association of the recombination nodules (RNs, multicomponent recombinogenic factories) and the axial elements of the SC (*Zickler and Kleckner, 2015*).

Meiotic DSBs repair is an evolutionarily conserved pathway that is highly regulated to promote the formation of at least one CO per bivalent. This chromosome connection between bivalents through chiasmata is required for a correct reductional division. As other DNA repair processes, proper meiotic recombination is essential for genome stability and alterations can result in infertility, miscarriage and birth defects (*Geisinger and Benavente, 2017*; *Handel and Schimenti, 2010*; *Webster and Schuh, 2017*).

Infertility refers to failure of a couple to reproduce and affects 10–15% of couples (*Isaksson and Tiitinen, 2004*). Infertility can be due to female factors, male factors, a combination of both or to unknown causes, each category representing approximately 25% of cases (*Isaksson and Tiitinen, 2004*; *Matzuk and Lamb, 2008*). There are several underlying causes and physiological, genetic and even environmental and social factors can play a role. Forward and reverse genetic analyses in model organisms have identified multiple molecular pathways that regulate fertility and have allowed to infer reasonable estimates of the number of protein-coding genes essential for fertility (*de Rooij and de Boer, 2003*; *Schimenti and Handel, 2018*).

Primary ovarian insufficiency (POI) is a major cause of female infertility and affects about 1–3% of women under 40 years of age. It is characterized by cessation of ovarian function before the age of 40 years. POI results from a depletion of the ovarian follicle pool and can be isolated or syndromic. Genetic causes of POI account for approximately 20% of cases (*Rossetti et al., 2017*). Although infertility-causing pathogenic variants are inherently unlikely to spread in a population, they can be observed within families, especially when there is consanguinity. Such cases provide crucial insights into the function of the genes and molecular mechanisms that they disrupt. Over the last decade, causative variants in several genes have been found using whole exome sequencing in 'POI pedigrees'. In particular, pathogenic variants in genes involved in DNA replication, recombination or repair, such as *STAG3, SYCE1, HFM1, MSH5* and *MEIOB* have been formally implicated in this condition by ourselves and others (*Caburet et al., 2014*; *Caburet et al., 2019a*; *de Vries et al., 2014*; *Guo et al., 2017*; *Primary Ovarian Insufficiency Collaboration et al., 2014*).

In this study, we have identified in a consanguineous family with POI the candidate S167L missense variant in *HSF2BP*, an essential yet poorly studied meiotic gene. *HSF2BP* encodes an interactor of the heat-shock response transcription factor HSF2 (*Yoshima et al., 1998*). During the course of this work and, in agreement with our results, two independent groups showed that HSF2BP is essential for meiotic recombination through its ability to interact with BRCA2 (*Brandsma et al., 2019*; *Zhang et al., 2019*). Here, we report that the introduction of the missense variant HSF2BP-S167L in mouse leads to subfertility and DNA repair defects during prophase I. In addition, we identified a protein complex composed of BRCA2, HSF2BP, and the as yet unexplored C19ORF57/BRME1 (meiotic double-stranded break BRCA2/HSF2BP complex associated protein) as a key component of the meiotic recombination machinery. Our studies show that a single substitution (S167L) in HSF2BP leads to a reduced loading of both BRME1 and HSF2BP at the RNs. Furthermore, our results suggest that meiotic progression requires a critical threshold level of HSF2BP/BRME1 for the ulterior loading of the recombinases to the RNs.

## Results

### Clinical cases

The parents are first-degree cousins of Israeli Arab origin. Of the five daughters, three are affected with POI and presented with early secondary amenorrhea. They had menarche at normal age (at 13–

14) but with irregular menses that stopped around 25. Only one of the patients affected by POI could have a child with the help of a fertility treatment (see pedigree in *Figure 1*). In order to identify the genetic basis of this familial POI case, we performed whole exome sequencing on genomic DNA from two POI patients, III-2 and III-3, and their fertile sister III-10 (*Supplementary file 1a*). Variants were filtered on the basis of (i) their homozygosity in the patients, (ii) their heterozygosity or absence in the fertile sister, (iii) their absence in unrelated fertile in-house controls and (iv) a minor allele frequency (MAF) below 0.01 in all available databases (*Supplementary file 1b*). This filtering process led to the identification of a missense substitution located in the *HSF2BP* gene: rs200655253 (21:43630396 G > A, GRCh38). The variant lies within the sixth exon of the reference transcript ENST00000291560.7 (NM_007031.2:c.500C > T) and changes a TCG codon into a TTG (NP_008962.1:p.Ser167Leu). It is very rare (Variant Allele Frequency/VAF 0.0001845 in the GnomAD database and 0.0005 in the GME Variome dedicated to Middle-East populations) and absent in a

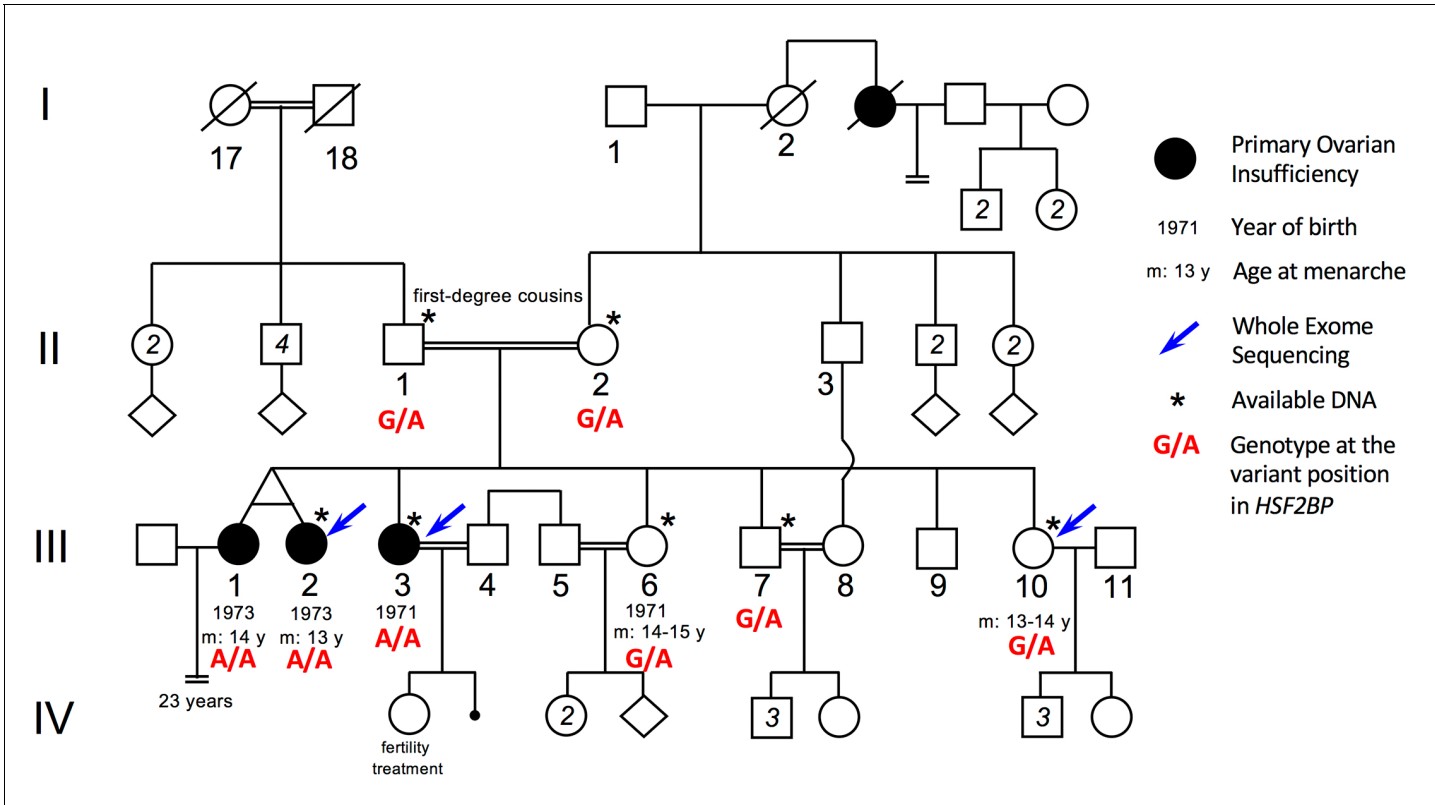

**Figure 1.** Pedigree of the consanguineous family with the variant HSF2BP-S167L. III-1 and III-2 are monozygotic twins, who appear phenotypically dizygotic. Clinical investigation confirmed POI, with normal 46, XX karyotype (500 bands and SKY spectral karyotyping). Year of birth and age of menarche are indicated when known. III-1 became amenorrheic at age 24 and III-2 at age 25, both after irregular menstruations since menarche. III-1 presents with a short stature (152 cm, within the 3–5 percentile), a normal neck, cubitus valgus and metacarpal shortening of 4–5. Ultrasound investigation showed normal uterus and ovaries. Her g-banding karyotyping was normal 46, XX (500 bands) and variants in FMR1 gene were ruled out. III-2 displays a normal secondary sexual development with no dysmorphic sign. Clinical investigation confirmed POI, with normal 46, XX karyotype (500 bands and SKY spectral karyotyping). The elder sister III-3 was also diagnosed with POI, with no further clinical information. She is 160.5 cm. She had one normal pregnancy with the help of 'fertility treatment', and a second unsuccessful attempt. The two fertile sisters, III-6 and III-10 had their menarche at 14–15 and 13–14 respectively, with regular menstruations ever since. They are respectively 150 cm and 151 cm, with no clinical sign, and each one had several children without difficulties. The fertile brother III-7 is 171 cm and shows no health or fertility problem. He developed frontal baldness since the age of 30. The genotype of each individual at the variant genomic position in *HSF2BP* is shown in red, as determined by Sanger sequencing for available DNAs (See *Figure 1—figure supplement 1*).

The online version of this article includes the following figure supplement(s) for figure 1:

**Figure supplement 1.** Segregation of the S167L Variant in HSF2BP in the consanguineous family shows the chromatograms obtained by Sanger sequencing of the HSF2BP-S167L variant in the family.

**Figure supplement 2.** Strong conservation of the Ser167 residue in HSF2BP protein in 99 mammals.

**Figure supplement 3.** Strong conservation of the Ser167 residue in HSF2BP in 48 birds and reptiles and 64 fish species.

homozygous state from all available databases. The variant was verified by Sanger sequencing and was found to segregate in a Mendelian fashion within the family: the affected twin III-1 was homozygous for the variant and both parents and fertile siblings were heterozygous carriers (*Figure 1—figure supplement 1*). Therefore, there was no homozygous males identified in this family, preventing the analysis of the impact of this variant on male fertility. Serine 167 is a highly conserved position and the S167L variant is predicted to be pathogenic or deleterious by 11 out of the 18 pathogenicity predictors available in dbNSFP 3.5. (*Supplementary file 1c*, *Figure 1—figure supplement 2* and *Figure 1—figure supplement 3*).

## Mice with the HSF2BP S167L variant show a partial reduction of fertility

During the course of this work, two independent groups showed that HSF2BP is essential for meiotic recombination through its ability to interact with the armadillo repeats of BRCA2 (*Zhang et al., 2019*). Both groups showed that genetic disruption of *Hsf2bp* in mouse leads to the accumulation in the chromosomes axes of DNA repair proteins such as γH2AX (ATR-dependent phosphorylation of H2AX marks DSBs) and the single stranded-DNA binding protein RPA, a strong reduction of the recombinases DMC1 and RAD51 at the RNs and a lack of COs as labelled by MLH1 (*Baker et al., 1996*). The end result is male sterility (*Brandsma, 2006*; *Brandsma et al., 2019*; *Zhang et al., 2019*). However, loss of HSF2BP in female mice showed a milder meiotic phenotype (*Zhang et al., 2019* and our own data, see below) and a weak albeit non-statistical significant reduction of fertility (*Brandsma et al., 2019*) despite all of the mutants are nulls though in different genetic backgrounds.

In order to confirm the causality of the S167L variant in this POI family, we generated a knock-in mouse $Hsf2bp^{S167L/S167L}$ by genome editing (*Figure 2—figure supplement 1a*). We also generated a loss-of-function model ($Hsf2bp^{-/-}$) for direct comparison (*Figure 2—figure supplement 1b–d*). $Hsf2bp^{S167L/S167L}$ male and female mice were able to reproduce but females showed a significant reduction in the number of litters (*Figure 2a*), whilst males only showed a slight non-significant reduction in fertility (*Figure 2a*), suggesting that the S167L variant impacts murine fertility.

Histological analysis of $Hsf2bp^{S167L/S167L}$ ovaries revealed no apparent differences in the number of follicles in comparison to wild-type (WT) animals (*Figure 2b–c* and *Figure 2—figure supplement 2a*), in contrast with the drastic reduction of the follicle pool in $Hsf2bp^{-/-}$ ovaries (*Figure 2b–c*). Testes from $Hsf2bp^{S167L/S167L}$ mice displayed a reduced size (21% reduction compared to WT mice; testis/body weight ratio: S167L 0.26% ± 0.07 (n = 12) vs 0.33% ± 0.05 for WT controls (n = 14), \*\*p<0,01, *Figure 2d* and *Figure 2—figure supplement 2b*) and this reduction was stronger in $Hsf2bp^{-/-}$ testes (70% reduction compared to WT, testis/body weight ratio: $Hsf2bp^{-/-}$ 0.10% ± 0.005 (n = 6) vs 0.33% ± 0.05 for WT controls (n = 14) \*\*\*\*p<0,001, *Figure 2d* and *Figure 2—figure supplement 2b*). Histological analysis of adult $Hsf2bp^{S167L/S167L}$ testes revealed seminiferous tubules with a partial arrest with apoptotic spermatocytes (meiotic divisions) and their epididymis exhibited scarcer spermatozoa (*Figure 2e*). Consistent with these results, $Hsf2bp^{S167L/S167L}$ males showed increased numbers of meiotic divisions positive for TUNEL staining (*Figure 2f*) and a reduction in the number of spermatozoa in the epididymis ($3.3 \times 10^6$ in the $Hsf2bp^{S167L/S167L}$ mutant vs $4.3 \times 10^6$ in the WT; *Figure 2g*). During mouse spermatogenesis, the 12 stages of the epithelial cycle can be distinguished in seminiferous tubule sections by identifying groups of associated germ cell types (*Ahmed and de Rooij, 2009*). Following these criteria, the seminiferous epithelium of $Hsf2bp^{-/-}$ mice showed a stage IV arrest, characterized by a massive apoptosis of zygotene-like spermatocytes occurring at the same time that In spermatogonia divide into B spermatogonia (*Figure 2e*). The presence of spermatogonia, spermatocytes, Sertoli and Leydig cells was not altered in any of the mutants (*Figure 2e*). These results suggest that mice bearing the POI-causing variant only partially phenocopy the human disease.

## $Hsf2bp^{S167L/S167L}$ meiocytes show an altered meiotic homologous recombination

To further characterize meiotic defects, $Hsf2bp^{S167L/S167L}$ meiocytes were first analyzed for the assembly/disassembly of the SC by monitoring the distribution of SYCP1 and SYCP3. We did not observe any difference in synapsis and desynapsis from leptotene to diplotene in both oocytes and

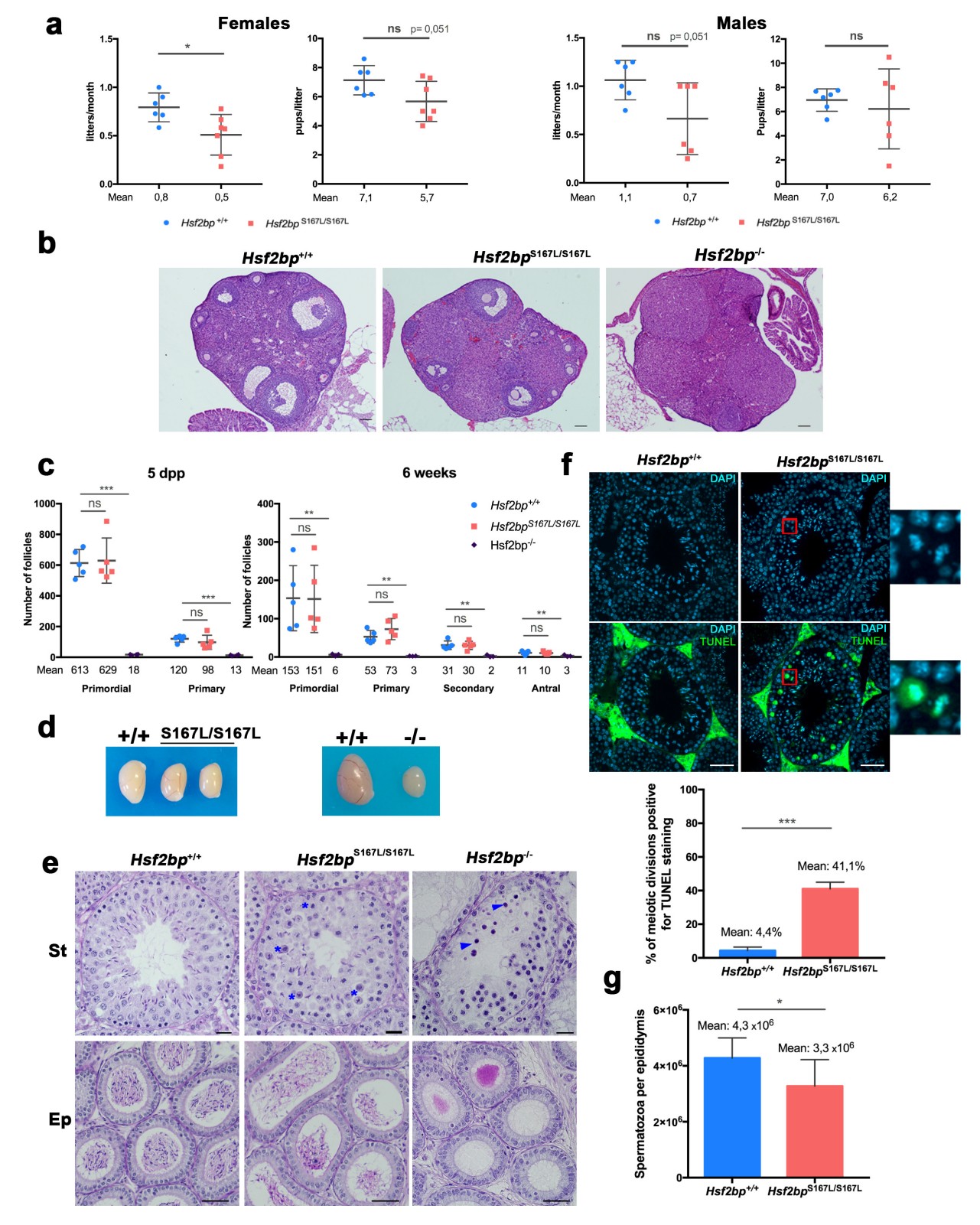

**Figure 2.** Mice carrying the HSF2BP S167L variant show a partial reduction of fertility. (**a**) Fertility assessment of males and female *Hsf2bp*^S167L/S167L^ and WT mice showing the number of litters per month and the number of pups per litter (see Materials and methods). Mice: *Hsf2bp*^+/+^n = 6 females/6 males, *Hsf2bp*^S167L/S167L^n = 7 females/6 males. Two-tailed Welch's t-test analysis: *p<0.05. (**b**) Hematoxylin and eosin stained sections of ovaries from adult (8 weeks) *Hsf2bp*^+/+^, *Hsf2bp*^S167L/S167L^ and *Hsf2bp*^-/-^ females. *Hsf2bp*^-/-^ ovaries but not *Hsf2bp*^+/+^ and *Hsf2bp*^S167L/S167L^ showed a strong depletion

*Figure 2 continued on next page*

Figure 2 continued

of follicles. Bar in panels 100 µm. (**c**) Quantification of the number of follicles (primordial, primary, secondary and antral follicles) per ovary in *Hsf2bp*^+/+^, *Hsf2bp*^S167L/S167L^ and *Hsf2bp*^-/-^ females at 5 dpp and 6 weeks of age showing no differences between *Hsf2bp*^+/+^ and *Hsf2bp*^S167L/S167L^ but a strong reduction in the oocyte pool in *Hsf2bp*^-/-^ females. Ovaries: five dpp/6 weeks = 5/5 ovaries from *Hsf2bp*^+/+^ and *Hsf2bp*^S167L/S167L^ and 4/3 from *Hsf2bp*^-/-^. Two-tailed Welch's t-test analysis: ns, no significant differences, **p<0.01, ***p<0.001. (**d**) Testis size of *Hsf2bp*^S167L/S167L^ (left, 21% reduction) and *Hsf2bp*^-/-^ mice (right, 70% reduction) in comparison with their WT counterparts. See *Figure 2—figure supplement 2b* for the quantification. (**e**) PAS and Hematoxylin stained testis sections. The S167L variant leads to a partial spermatogenic arrest with an elevated number of apoptotic meiotic divisions (blue asterisks) and a reduction of the number of spermatozoa in the epididymides in comparison with the WT control (*Hsf2bp*^+/+^). The null allele (*Hsf2bp*^-/-^) showed a complete spermatogenic arrest at epithelial stage IV and absence of spermatozoa. Massive apoptosis of spermatocytes is indicated (blue arrowheads). Bar: upper panels 10 µm, lower panels 20 µm. (St) Seminiferous tubules, (Ep) Epididymides. (**f**) Immunohistochemical detection of apoptotic cells by TUNEL staining showing an increase of apoptotic meiotic divisions in stage XII tubules from *Hsf2bp*^S167L/S167L^ males (magnified panel). Plot under the panel represents the quantification. Mice: n = 3 adult mice for each genotype. Two-tailed Welch's t-test analysis: ***p<0.001. Bar in panel, 25 µm. (**g**) Quantification of epididymal sperm in *Hsf2bp*^+/+^ and *Hsf2bp*^S167L/S167L^ adult mice. Epididymides: n = 8 for each genotype. Two-tailed Welch's t-test analysis: *p<0.05.

The online version of this article includes the following figure supplement(s) for figure 2:

**Figure supplement 1.** Generation and genetic characterization of *Hsf2bp* S167L and *Hsf2bp*-deficient mice.

**Figure supplement 2.** Fertility defects in *Hsf2bp*^S167L/S167L^ mice.

spermatocytes (*Figure 3—figure supplement 1a–b*). However, we observed an elevated number of apoptotic meiotic divisions in *Hsf2bp*^S167L/S167L^ males (*Figure 2—figure supplement 2c*). These results are consistent with the partial arrest observed in the histological analysis (*Figure 2e*). As expected, this phenotype was exacerbated in *Hsf2bp*^-/-^ spermatocytes that were arrested at a zygo-tene-like stage (*Figure 3—figure supplement 1c*). *Hsf2bp*^-/-^ oocytes showed a delay in prophase I progression with the majority of cells at zygotene stage in 17.5 days post-coitum (dpc) females, whilst the WT oocytes were mainly at pachytene stage. Additionally, we observed increased num-bers of oocytes showing synapsis defects in the *Hsf2bp*^-/-^ oocytes (*Hsf2bp*^-/-^: 45,5% ± 1,5 vs WT: 7,5% ± 1,5; n = 2 (both genotypes), **p<0,01, *Figure 3—figure supplement 1d*). These results strongly suggest that the POI variant S167L is a hypomorphic allele.

Next, we analyzed whether the POI-inducing variant affects the loading/stability of HSF2BP by immunolabeling meiocytes from *Hsf2bp*^S167L/S167L^ mice. We observed a striking reduction of HSF2BP staining at the axes during prophase I in both spermatocytes and oocytes (*Figure 3a–b*). Western blot analysis of WT, *Hsf2bp*^S167L/S167L^ and *Hsf2bp*^-/-^ in whole testis extracts from 13 days post-par-tum (dpp) animals (*Figure 3c*) revealed that the reduced labeling observed by immunofluorescence correlated with a reduced protein expression level, suggesting that the mutation leads to a reduced expression and/or stability.

Given that HSF2BP is essential for DNA repair, we carried out a comparative staining analysis of γH2AX, the ssDNA-binding protein RPA, the recombinases RAD51 and DMC1, the ssDNA-binding protein SPATA22 (complexed to RPA during resection) and CO formation in meiocytes from *Hsf2bp*^S167L/S167L^, *Hsf2bp*^-/-^ and WT animals (*Figures 4*, *5* and *6*, *Figure 4—figure supplement 1*, *Figure 5—figure supplement 1* and *Figure 6—figure supplement 1*). Our results revealed that *Hsf2bp*^S167L/S167L^ spermatocytes showed an increased labeling of γH2AX at pachytene (*Figure 4a*), an accumulation of RPA at the chromosome axis (*Figure 4b* and *Figure 4—figure supplement 1a*), a reduction of the recombinases DMC1 and RAD51 staining (*Figure 5a–b* and *Figure 5—figure sup-plement 1a–b*), an accumulation of SPATA22 (*Figure 6a* and *Figure 6—figure supplement 1a*), and a decreased number of COs (measured as MLH1, *Figure 6b* and *Figure 6—figure supplement 1b*). In accordance with the reduction of COs, we observed the presence of univalents in the XY pair at pachynema as well as univalents in metaphase I spermatocytes from *Hsf2bp*^S167L/S167L^ mice (*Figure 6d* and *Figure 6—figure supplement 1c*). These results would explain the elevated number of apoptotic metaphases observed (*Figure 2e–f* and *Figure 2—figure supplement 2c*).

Our analysis in females showed accumulation of γH2AX staining (*Figure 4c*) but no accumulation in RPA labeling in *Hsf2bp*^S167L/S167L^ and *Hsf2bp*^-/-^ oocytes (*Figure 4d* and *Figure 4—figure supple-ment 1b*). Similar to the spermatocytes, DMC1 and RAD51 staining showed a reduction in both *Hsf2bp*^S167L/S167L^ and *Hsf2bp*^-/-^ oocytes (*Figure 5c–d* and *Figure 5—figure supplement 1c–d*). SPATA22 labeling in females showed a clear accumulation in *Hsf2bp*^-/-^ but only a trend towards accumulation in *Hsf2bp*^S167L/S167L^ oocytes (*Figure 6a* and *Figure 6—figure supplement 1a*). In

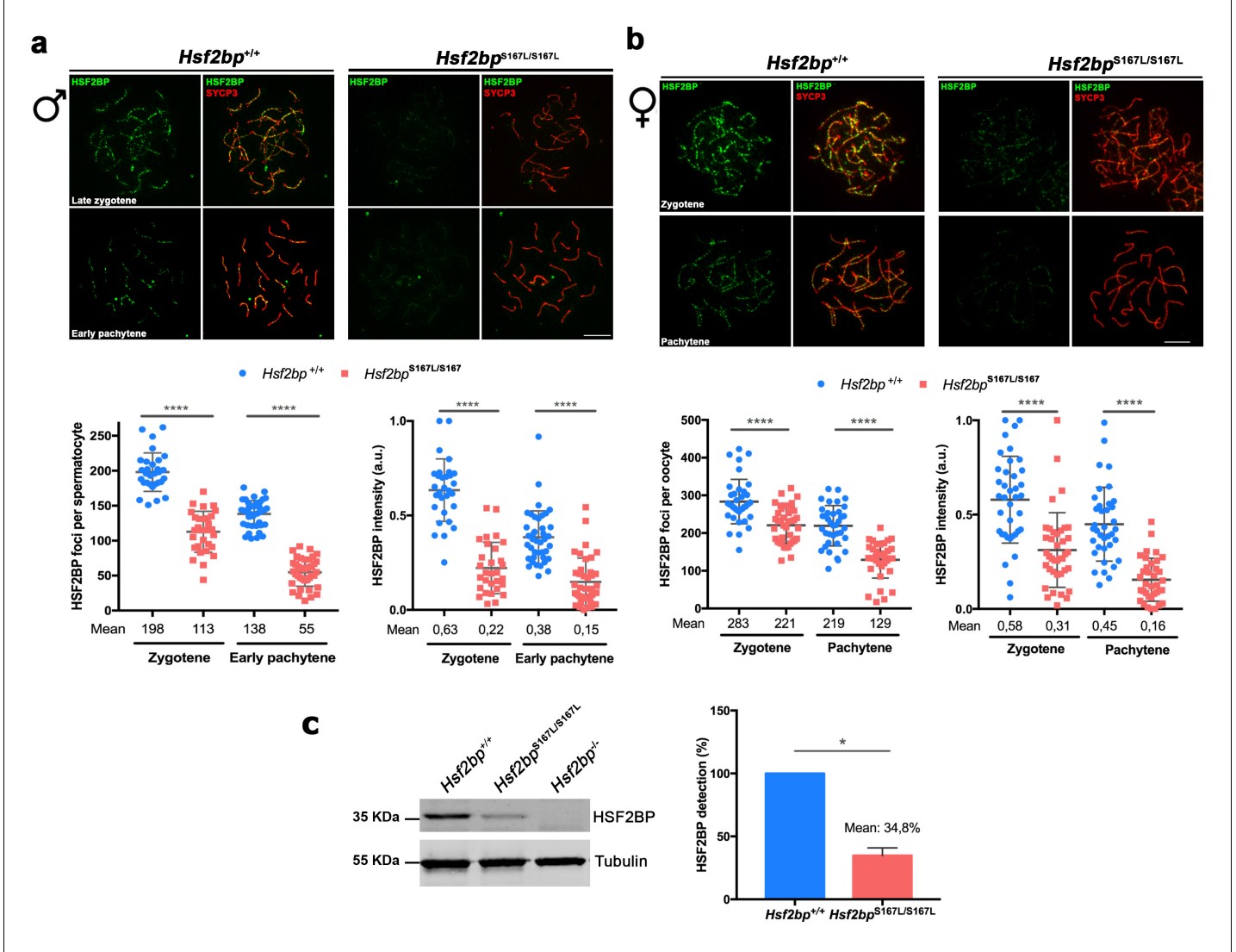

**Figure 3.** Meiocytes from *Hsf2bp*^S167L/S167L^ mice show a decrease in the expression of HSF2BP. (a–b) Double immunofluorescence of HSF2BP (green) and SYCP3 (red) in *Hsf2bp*^+/+^ and *Hsf2bp*^S167L/S167L^ (a) spermatocyte and (b) oocyte spreads showing a strong reduction in the labeling of HSF2BP at the chromosome axis. Plots under the panels show the quantification. Nuclei analyzed: 30 zygonemas and 40 pachynemas from two adult male mice of each genotype. In females 38/39 zygonemas and 37/35 pachynemas from two 17.5 dpc embryos of *Hsf2bp*^+/+^ and *Hsf2bp*^S167L/S167L^, respectively. Two-tailed Welch's t-test analysis: ****p<0.0001. (c) Western blot analysis of protein extracts from 13 dpp WT, *Hsf2bp*^S167L/S167L^ and *Hsf2bp*^-/-^ testes using polyclonal antibodies against HSF2BP. Tubulin was used as loading control. Graph on the right represents the relative quantification of the immunoblotting. Mice: n = 2 *Hsf2bp*^+/+^, *Hsf2bp*^S167L/S167L^ and *Hsf2bp*^-/-^. Two-tailed Welch's t-test analysis: *p<0.05. Bar in panels a-c, 10 µm.

The online version of this article includes the following figure supplement(s) for figure 3:

**Figure supplement 1.** *Hsf2bp*^S167L/S167L^ mice do not show synapsis defects.

agreement with the lower presence of recombinases, the number of COs (measured as interstitial CDK2 foci) was also reduced in *Hsf2bp*^S167L/S167L^ oocytes, and a stronger reduction was observed in *Hsf2bp*^-/-^ oocytes (*Figure 6c* and *Figure 6—figure supplement 1d*). Overall, male and female *Hsf2bp*^S167L/S167L^ mice share alterations in the meiotic recombination pathway although with different reproductive outcome.

We next sought to understand how the *HSF2BP* pathogenic variant was mediating the observed meiotic alteration. HSF2BP has been shown to bind BRCA2, an essential protein for meiotic homologous recombination (*Martinez et al., 2016*; *Sharan et al., 2004*), by a direct interaction that involves Arg200 in HSF2BP and the Gly2270-Thr2337 region within the C-terminal fragment of BRCA2

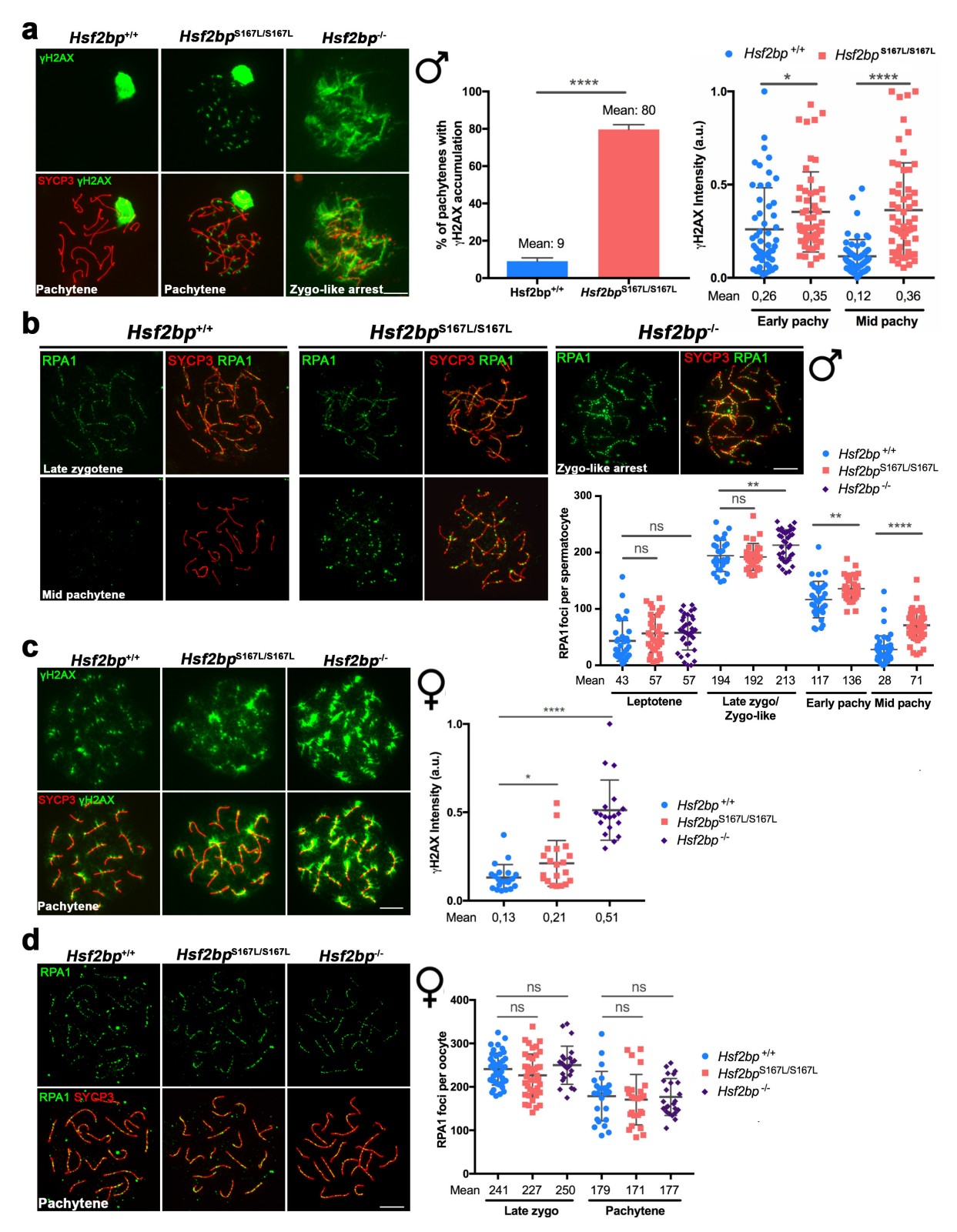

**Figure 4.** DNA repair in *Hsf2bp*[S167L/S167L] mice. (**a, c**) Double labeling of γH2AX (green) and SYCP3 (red) in (**a**) spermatocyte and (**c**) oocyte spreads from WT, *Hsf2bp*[S167L/S167L] and *Hsf2bp*[-/-] mice. (**a**) Males display an accumulation of γH2AX patches in *Hsf2bp*[S167L/S167L] pachynemas and a strong accumulation in the whole nucleus in *Hsf2bp*[-/-] zygotene-like arrested cells. Plots on the right of the panel represent the percentage of pachynemas with γH2AX labeling (Nuclei: 364 *Hsf2bp*[+/+] and 376 *Hsf2bp*[S167L/S167L] from three adult mice) and the quantification of γH2AX intensity on autosomes at

*Figure 4 continued on next page*

Figure 4 continued

early and mid-pachytene stages (Nuclei: 53 early and 60 mid pachynemas from three adult mice of each genotype). Two-tailed Welch's t-test analysis: *p<0.05, ****p<0.0001. (c) In females there is an accumulation of γH2AX in $Hsf2bp^{S167L/S167L}$ pachynemas that is stronger in those from $Hsf2bp^{-/-}$ females. Nuclei: n = 21/20/19 pachynemas from 2 $Hsf2bp^{+/+}/Hsf2bp^{S167L/S167L}/Hsf2bp^{-/-}$ embryos (17.5 dpc). Two-tailed Welch's t-test analysis: *p<0.05, ****p<0.0001. (b, d) Double immunolabeling of RPA1 (green) and SYCP3 (red) in (b) spermatocyte and (d) oocyte spreads from $Hsf2bp^{+/+}$, $Hsf2bp^{S167L/S167L}$ and $Hsf2bp^{-/-}$. (b) In males, RPA1 accumulates at early and mid-pachytene in S167L spermatocytes and in the zygotene-like arrested cells from $Hsf2bp^{-/-}$. Plot on the right of the panel represents the quantification. Nuclei: n = 31/34/37 leptonemas, n = 29/29/37 zygonemas/zygonemas-like from three adult $Hsf2bp^{+/+}$, $Hsf2bp^{S167L/S167L}$ and $Hsf2bp^{-/-}$ mice respectively, n = 33 early and 46 mid pachynemas from three adult $Hsf2bp^{+/+}$ and $Hsf2bp^{S167L/S167L}$ mice. Two-tailed Welch's t-test analysis: ns, no significant differences; **p<0.01, ****p<0.0001. (d) In females, RPA1 labeling is similar in $Hsf2bp^{+/+}$, $Hsf2bp^{S167L/S167L}$ and $Hsf2bp^{-/-}$ oocytes at zygotene and pachytene. Plot on the right of the panel represents the quantification. Nuclei: $Hsf2bp^{+/+}/Hsf2bp^{S167L/S167L}/Hsf2bp^{-/-}$ n = 42/41/23 zygonemas from two embryos (16.5 dpc) and n = 25/25/24 pachynemas from two embryos (17.5 dpc). Two-tailed Welch's t-test analysis: ns, no significant differences. Bar in all panels, 10 μm. Extended panels for RPA1 figures in *Figure 4—figure supplement 1*.

The online version of this article includes the following figure supplement(s) for figure 4:

**Figure supplement 1.** RPA localization in *Hsf2bp* mutants.

(*Brandsma et al., 2019*). Given the impossibility to detect endogenous BRCA2 by immunofluorescence in mouse spermatocytes, we carried out co-localization/interaction assays in a heterologous system by transfecting BRCA2-C (i.e. its C-term) and HSF2BP in U2OS/HEK293T. Our results showed that BRCA2-C co-immunoprecipitates with both HSF2BP-WT and HSF2BP-S167L in similar ways (*Figure 6—figure supplement 2a*). In single transfections, HSF2BP localized in the nucleus and cytoplasm whereas BRCA2-C showed nuclear localization. This pattern changed drastically to a nuclear dotted pattern when co-transfected (*Figure 6—figure supplement 2b*). This re-localization was independent of the HSF2BP variant, suggesting that the HSF2BP variant effects are not directly mediated by BRCA2 delocalization.

## BRME1, a novel interactor of HSF2BP

In order to further understand the mechanism underlying the pathogenicity of the HSF2BP-S167L variant, we searched for proteins that interact with the murine HSF2BP through a yeast two hybrid (Y2H) screening. The analysis of the clones with putative interactors revealed that 19 out of 98 analyzed clones matched the uncharacterized gene 4930432K21Rik, which corresponds to human C19ORF57, hereby dubbed BRME1 for Break Repair Meiotic recombinase recruitment factor 1. This HSF2BP interactor consists of 600 amino acids with a high content of acidic residues, has no recognizable functional domains and is intrinsically disordered. The interaction was validated by transiently transfecting plasmids driving the expression of HSF2BP and BRME1. Both HSF2BP-S167L and WT interacted with BRME1 (*Figure 7a*). We further validated this interaction in vivo by co-immunoprecipitation (co-IP) of both proteins from mouse whole testis extracts (*Figure 7b*). To identify the regions required for this interaction, we split the BRME1 protein into three fragments (N-terminal, central region and C-terminal). We mapped the HSF2BP/BRME1-interacting domain to the C-term fragment of BRME1 (spanning residues 475–600 of the murine protein, *Figure 7c*). In line with this, the $Brme1^{\Delta142-472/\Delta142-472}$ mutant mice, expressing the BRME1 protein devoid of its central part, were fertile and did not show defects in chromosome synapsis or an alteration of HSF2BP loading to axes, further indicating that a large fraction of the coding protein of BRME1 is not essential for BRME1/HSF2BP function in vivo (*Figure 7—figure supplement 1*).

We also sought to characterize the involvement of BRME1 in meiosis through immunofluorescence. BRME1 localized to the chromosome axes of WT meiocytes from zygotene to pachytene with a pattern of discrete foci that mimics the RNs (*Figure 7—figure supplement 2a–b*). In agreement with the yeast two hybrid and co-IP results, BRME1 perfectly co-localized with HSF2BP on the chromosome axes (*Figure 7d* and *Supplementary file 1d* for quantification). This co-localization was verified by super-resolution microscopy (*Figure 7e*). In accordance with the tight association of BRME1 with HSF2BP and with a role in DSB repair, both HSF2BP and BRME1 colocalized with RPA and DMC1 foci. During prophase I, HSF2BP and BRME1 showed higher levels of spatio-temporal colocalization at the RNs with RPA than with DMC1 (*Figure 7—figure supplement 3a–b* and *Supplementary files 1d-e* for quantification). We also analyzed the HSF2BP-dependent localization of BRME1 in $Hsf2bp^{-/-}$ and $Hsf2bp^{S167L/S167L}$ mutants. Immunofluorescence analysis of meiocytes

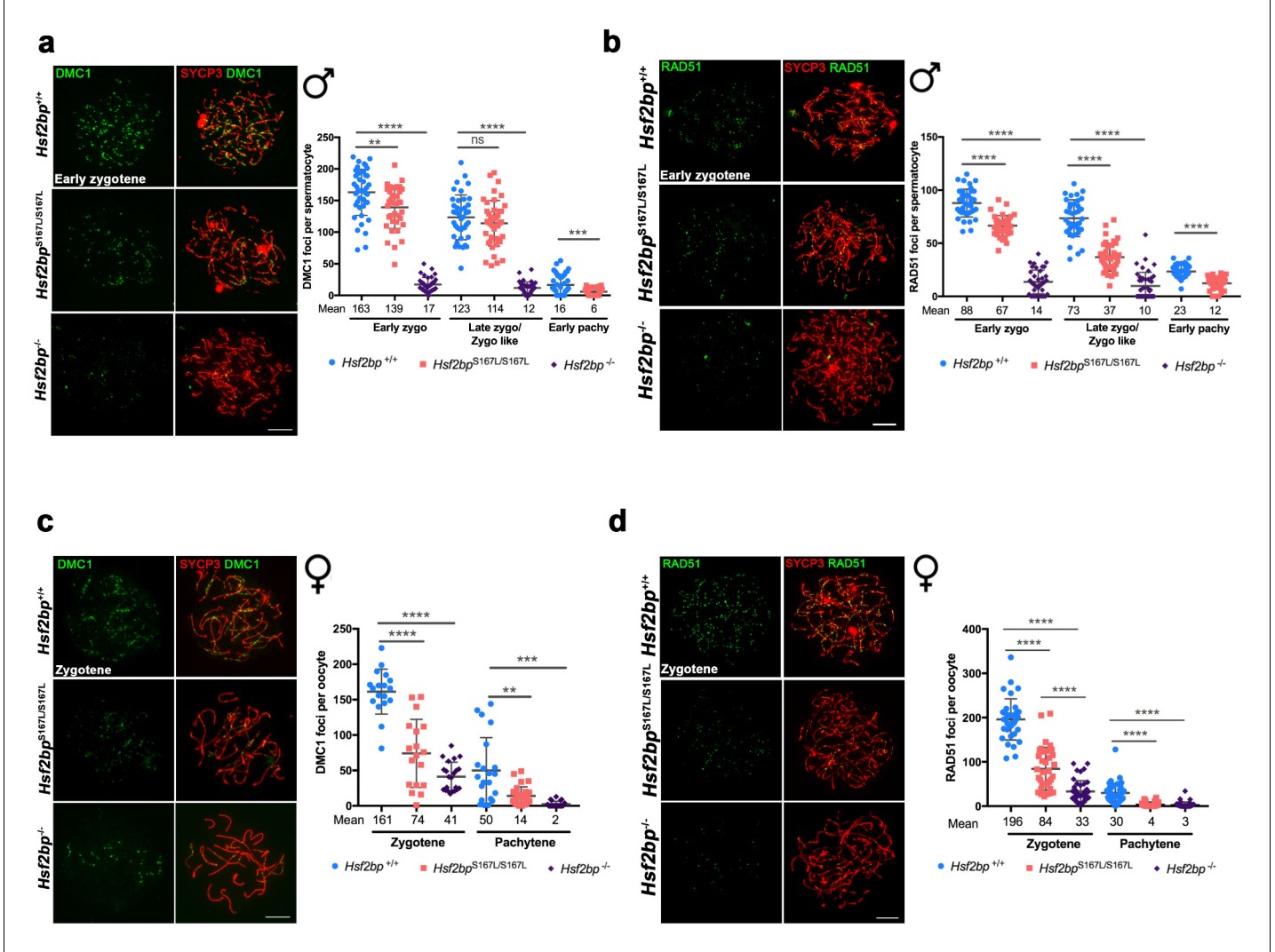

**Figure 5.** The loading of recombinases is compromised in *Hsf2bp*^S167L/S167L mice. Double immunolabeling of (**a, c**) DMC1 or (**b, d**) RAD51 (green) and SYCP3 (red) in *Hsf2bp*^+/+, *Hsf2bp* ^S167L/S167L and *Hsf2bp*^-/- (**a, b**) spermatocytes and (**c, d**) oocytes showing a strong reduction (*Hsf2bp*^-/-) and mild reduction (*Hsf2bp* ^S167L/S167L) in the number of foci in comparison with their WT counterparts. Plots on the right of the panels represent the quantification of foci on each genotype and stage. Male nuclei DMC1: *Hsf2bp*^+/+/*Hsf2bp* ^S167L/S167L/*Hsf2bp*^-/-, respectively, n = 43/38/37 early zygonemas, 41/43/37 late zygonemas/zygonemas-like and 44/37 early pachynemas from two adult mice of each genotype. Male nuclei RAD51: *Hsf2bp*^+/+/*Hsf2bp* ^S167L/S167L/*Hsf2bp*^-/- respectively n = 39 early zygonemas from all genotypes, 37/40/43 late zygonemas/zygonema-like and 37/39 early pachynemas from two adult mice of each genotype. Oocyte nuclei DMC1: *Hsf2bp*^+/+/*Hsf2bp* ^S167L/S167L/*Hsf2bp*^-/- n = 18/18/22 zygonemas from two embryos and n = 21/30/23 pachynemas from two embryos (17.5 dpc). Oocyte nuclei RAD51: *Hsf2bp*^+/+/*Hsf2bp* ^S167L/S167L/*Hsf2bp*^-/- n = 35/35/42 zygonemas and n = 42/42/40 pachynemas from two embryos (17.5 dpc). Two-tailed Welch's t-test analysis: ns, no significant differences, **p<0.01, ***p<0.001, ****p<0.0001. Bar in all panels, 10 µm. Extended panels for these figures in *Figure 5—figure supplement 1*.
The online version of this article includes the following figure supplement(s) for figure 5:

**Figure supplement 1.** Defective loading of recombinases in *Hsf2bp*^S167L/S167L mice.

showed a complete lack of BRME1 staining in the absence of HSF2BP and a significant reduction of foci number in the presence of HSF2BP-S167L variant (35% reduction in males and 72% in females at pachytene; *Figure 7f–g* and *Figure 7—figure supplement 4a*). Western blot analysis also revealed a drastic reduction of BRME1 expression in *Hsf2bp*^S167L/S167L and *Hsf2bp*^-/- spermatocytes, suggesting an HSF2BP-dependent stabilization of BRME1 (*Figure 7h*).

To assess if HSF2BP and/or BRME1 had DNA-binding activity (targeting to DSBs), we carried out an in vitro binding assay using HSF2BP and BRME1 proteins expressed in a transcription and translation coupled reticulocyte system (TNT; *Loregian et al., 2004*; *Souquet et al., 2013*) in which there

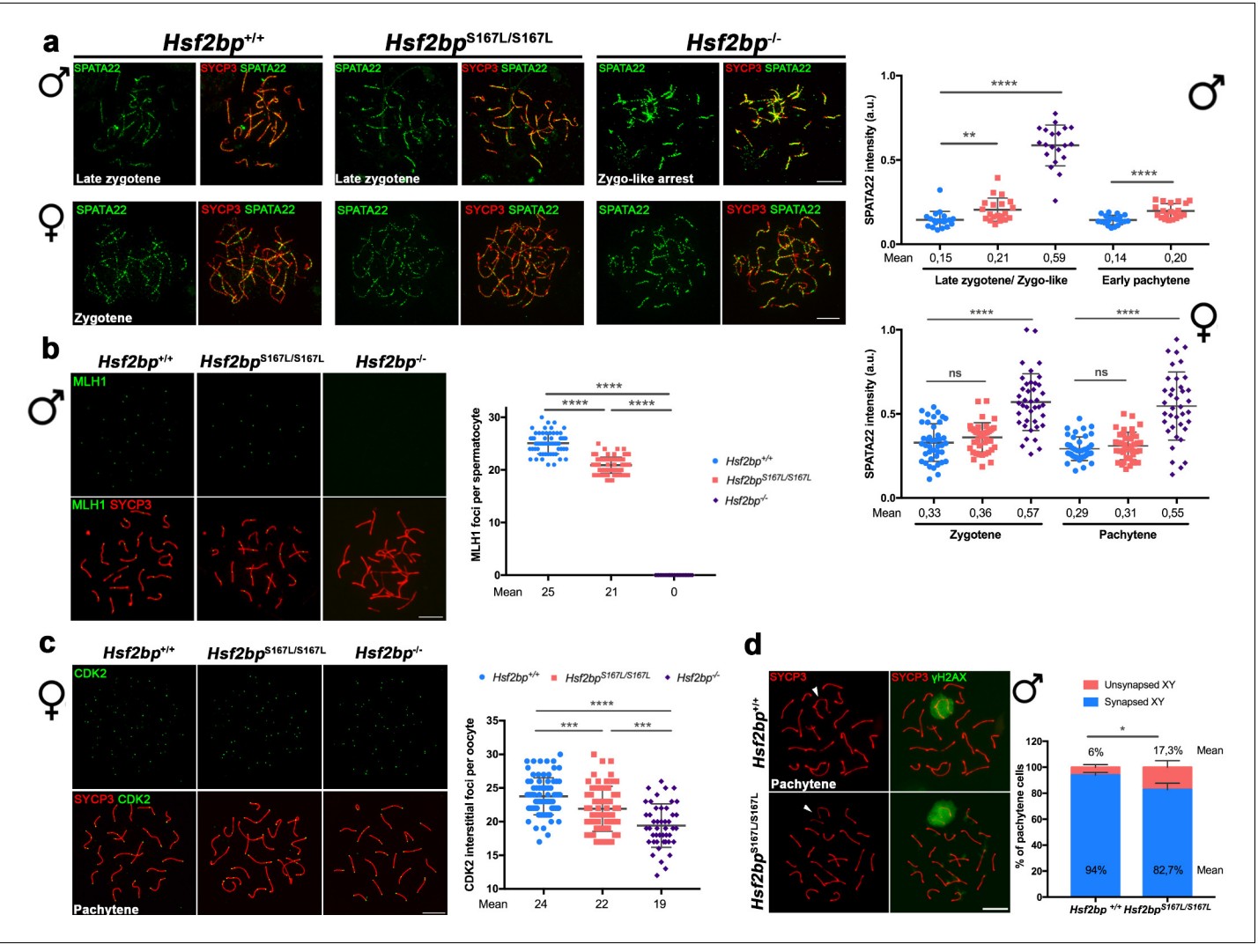

**Figure 6.** Recombination proficiency is decreased in *Hsf2bp*^S167L/S167L^ mice. (**a**) Double labeling of SPATA22 (green) and SYCP3 (red) in spermatocyte (upper panel) and oocyte (lower panel) spreads from WT, *Hsf2bp*^-/-^ and *Hsf2bp*^S167L/S167L^ mice. SPATA22 is accumulated in knock-out spermatocytes and oocytes and shows a milder accumulation in the *Hsf2bp*^S167L/S167L^ spermatocytes. *Hsf2bp*^S167L/S167L^ oocytes show a slight but not significant accumulation. Plots on the right of the panel represents the quantification of SPATA22 labeling. Males nuclei: n = 20 cells for each stage from two adult mice of each genotype. Females nuclei: *Hsf2bp*^+/+^/*Hsf2bp*^S167L/S167L^/*Hsf2bp*^-/-^ n = 41/40/40 zygonemas from two embryos and n = 40/39/38 pachynemas from two embryos (17.5 dpc). Two-tailed Welch's t-test analysis: ns, no significant differences, **p<0.01, ****p<0.0001. (**b**) Double immunofluorescence of MLH1 (green) and SYCP3 (red) in spermatocyte spreads from WT, *Hsf2bp*^S167L/S167L^ and *Hsf2bp*^-/-^. MLH1 foci are significantly reduced in the *Hsf2bp*^S167L/S167L^ spermatocytes and absent in the knock-out. The plot on the right shows the quantification. See also *Figure 6—figure supplement 1b* for the plot showing the percentage of bivalents without CO. Nuclei: n = 61 for *Hsf2bp*^+/+^, 89 for *Hsf2bp*^S167L/S167L^ and 60 for *Hsf2bp*^-/-^ from three adult mice of each genotype. Two-tailed Welch's t-test analysis: ****p<0.0001. (**c**) Double labeling of CDK2 (green) and SYCP3 (red) in oocyte spreads from 17.5 dpc *Hsf2bp*^+/+^, *Hsf2bp*^S167L/S167L^ and *Hsf2bp*^-/-^ embryos. During meiotic prophase I, CDK2 localizes to the telomeres of chromosomes from leptotene to diplotene. However, around mid-pachytene additional interstitial CDK2 signals appear at CO sites, colocalizing with MLH1. As a measure of COs, just interstitial CDK2 foci (non-telomeric) have been counted. *Hsf2bp*^-/-^ and *Hsf2bp*^S167L/S167L^ females show a high and moderate reduction in the number of COs, respectively. Plot on the right of the panel show the quantification. See also *Figure 6—figure supplement 1d* for the plot showing the percentage of bivalents without CO. Nuclei: *Hsf2bp*^+/+^/*Hsf2bp*^S167L/S167L^/*Hsf2bp*^-/-^ n = 79/67/46 from three embryos (17.5 dpc). Two-tailed Welch's t-test analysis: ***p<0.0001, ****p<0.0001. (**d**) Double immunofluorescence of γH2AX (green) and SYCP3 (red) in spermatocyte spreads from WT and *Hsf2bp*^S167L/S167L^ mice. At pachytene, γH2AX allows the identification of the XY bivalent. Diagram on the right represents the quantification of the pachynemas with unsynapsed sex chromosomes from *Hsf2bp*^S167L/S167L^ and WT mice. Nuclei: n = 150 pachynemas from three adult mice of each genotype. Two-tailed Welch's t-test analysis: *p<0.05. Bar in all panels, 10 μm.

The online version of this article includes the following figure supplement(s) for figure 6:

**Figure supplement 1.** Meiotic recombination is affected in *Hsf2bp*^S167L/S167L^ mice.

**Figure supplement 2.** Comparative interaction of HSF2BP-S167L and HSF2BP-WT with BRCA2.

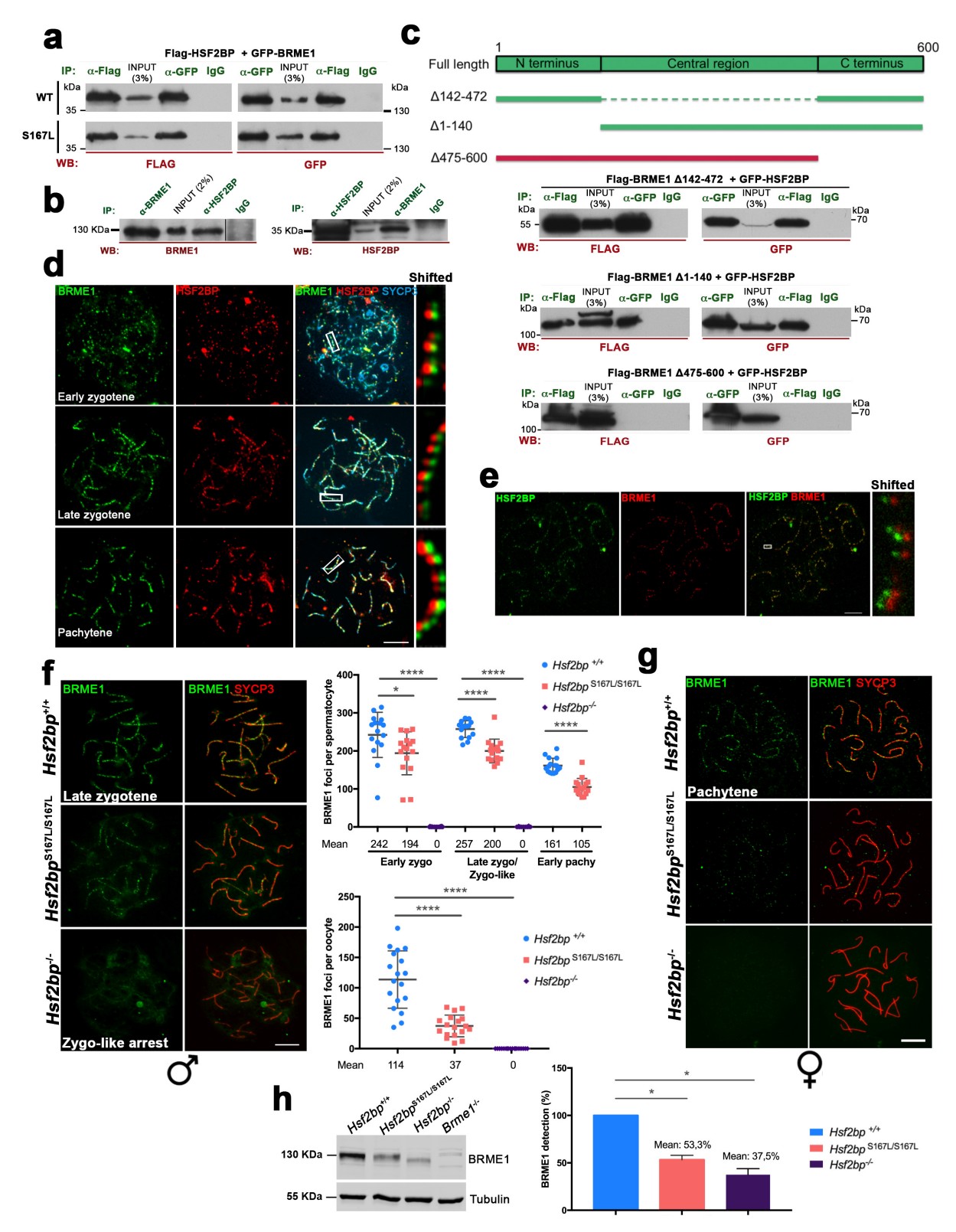

**Figure 7.** BRME1, a novel HSF2BP interactor that colocalizes to the recombination nodules. (**a**) HEK293T cells were transfected with Flag-HSF2BP (WT, upper panel; S167L, lower panel) and its novel interactor GFP-BRME1. Protein complexes were immunoprecipitated (IP: green text) with either anti-Flag or anti-EGFP or IgGs (negative control) and analysed by immunoblotting with the indicated antibody (WB: red text). Both HSF2BP variants (WT and S167L) co-immunoprecipitated similarly with BRME1. (**b**) IP of testis extracts with antibodies against BRME1, HSF2BP and IgGs as a negative control (IP:

*Figure 7 continued on next page*

Figure 7 continued

green text) and western blot with the indicated antibodies (WB: red text) (c) Schematic representation of full-length BRME1 protein and the corresponding deletion (Δ) constructs (filled boxes) generated to decipher the essential BRME1 region for interacting with HSF2BP (green positive interaction and red no interaction). Western blots under the scheme show the Co-IP experiments. HEK293T cells were transfected with GFP-HSF2BP and the different delta constructs of Flag-BRME1. The Δ475–600 abolishes the interaction, indicating that the C terminus of BRME1 is the essential region of interaction with HSF2BP. (d) Triple immunofluorescence of BRME1 (green), HSF2BP (red) and SYCP3 (blue) in WT spermatocyte spreads showing high colocalization between BRME1 and HSF2BP at late zygotene and pachytene (See *Supplementary file 1d* for quantification). Bar in panel, 10 µm. (e) Double immunolabeling of spermatocyte spread preparations with HSF2BP (green) and BRME1 (red) analyzed by Stimulated emission depletion (STED) microscopy. Bar in panel, 5 µm. (f–g) Double immunofluorescence of BRME1 (green) and SYCP3 (red) in (f) spermatocytes and (g) oocyte spreads from $Hsf2bp^{+/+}$, $Hsf2bp^{S167L/S167L}$ and $Hsf2bp^{-/-}$ showing a strong reduction of BRME1 staining in the S167L mutant and absence in the $Hsf2bp$ knock-out. Plots next to the panel represent the quantification. See also extended *Figure 7—figure supplement 4a*. Male nuclei: $Hsf2bp^{+/+}$/$Hsf2bp^{S167L/S167L}$/$Hsf2bp^{-/-}$: n = 16/16/20 early and 15/15/16 late zygonemas, 16/16 /- pachynemas from two adult mice. Female nuclei: n = 18 pachynemas from two embryos (17.5 dpc) of each genotype. Two-tailed Welch's t-test analysis: *p<0.05, ****p<0.0001. Bar in panels, 10 µm. (h) Western blot analysis of protein extracts from 13 dpp WT, $Hsf2bp^{S167L/S167L}$ and $Hsf2bp^{-/-}$ testes using an antibody against BRME1. Tubulin was used as loading control. Graph on the right represents the relative quantification of the immunoblotting. Mice: n = 2 $Hsf2bp^{+/+}$, $Hsf2bp^{S167L/S167L}$ and $Hsf2bp^{-/-}$. Two-tailed Welch's t-test analysis: *p<0.05.

The online version of this article includes the following figure supplement(s) for figure 7:

**Figure supplement 1.** *Brme1 Δ142–472* mutants do not show meiotic defects.
**Figure supplement 2.** C19ORF57/BRME1 localizes at meiotic RNs.
**Figure supplement 3.** Colocalization analysis of BRME1 and HSF2BP with RPA and DMC1.
**Figure supplement 4.** BRME1 localization depends on HSF2BP and none of them has DNA-binding abilities.
**Figure supplement 5.** Generation and genetic characterization of $Rnf212^{-/-}$ and $Hei10^{-/-}$ mice.
**Figure supplement 6.** BRME1 loading depends on DSBs generation but not on synapsis.

are no nuclear proteins and chromatin (*Melton et al., 1984*) and used RPA as positive control. Our results show that both proteins lacked direct DNA-binding abilities, in contrast to the strong activity of RPA (*Figure 7—figure supplement 4b–c*).

To determine the role of BRME1 in recombination and DNA repair, we analyzed its cytological distribution pattern in different mutants lacking synapsis/recombination-related proteins. These mutants were the meiotic cohesin REC8 (*Bannister et al., 2004*), the central element protein of the SC SIX6OS1 (*Gómez-H et al., 2016*), the E3 ligases involved in the stabilization of recombinogenic proteins RNF212 and HEI10 (*Qiao et al., 2014*), the spermatoproteasomal subunit PSMA8 (*Gómez-H et al., 2019*), and the nuclease SPO11 required for DSBs generation ($Rnf212^{-/-}$, $Hei10^{-/-}$ and $Spo11^{-/-}$ mouse mutants are described in this work, see Materials and methods and *Figure 7—figure supplement 5* and *Figure 7—figure supplement 6a*, *Baudat et al., 2000*). HSF2BP staining was also carried out for a direct comparison. We were able to show that none of the recombination-deficient mutants abrogate BRME1 labeling at zygotene (or the corresponding meiotic stage at which the mutant spermatocytes are arrested), in contrast to its absence of loading in SPO11-deficient mice (*Figure 7—figure supplement 6b*, left). These results are very similar to those obtained for HSF2BP in these mutants (*Figure 7—figure supplement 6b*, right) and indicate that SPO11-dependent DSBs are essential for targeting HSF2BP/BRME1 to the RNs, and that the heterocomplex can be positioned at early events soon after DSBs generation.

To functionally analyze the role of BRME1 in mouse fertility, we generated a $Brme1^{-/-}$ null mutant by genome editing (*Figure 8—figure supplement 1a–d*). $Brme1^{-/-}$ females, despite being fertile, showed a strong reduction of the follicle pool (*Figure 8a*). Male $Brme1^{-/-}$ mice were infertile, the average size of their testes was severely reduced (76% reduction compared to WT; testis weight/body weight ratio: $Brme1^{-/-}$ 0,08% ± 0004 (n = 6) vs 0,33% ± 0,05 for WT controls (n = 14), ****p<0.0001, *Figure 8b* and *Figure 2—figure supplement 2b*), and lacked spermatozoa (*Figure 8c*). Histological analysis showed a meiotic arrest at epithelial stage IV with apoptotic spermatocytes (*Figure 8c*). Double immunolabeling of SYCP3 and SYCP1 revealed that spermatocytes were partially synapsed and showed a partner-switch phenotype in which synapsis is not restricted to homologous pairs (*Figure 8d*). The arrest corresponds to a zygotene-like stage though a small fraction of cells (3,7% ± 1,9; n = 3) were able to escape this blockage reaching early pachytene. $Brme1^{-/-}$ oocyte spread analysis revealed the presence of a subset of fully synapsed pachynemas but an increased number of cells with different degree of asynapsis (47,9% ± 2,2 vs 12% ± 5,7 in the WT; n = 2 (both genotypes), *p<0,05 *Figure 8e* and *Figure 8—figure supplement 1e*). Given the

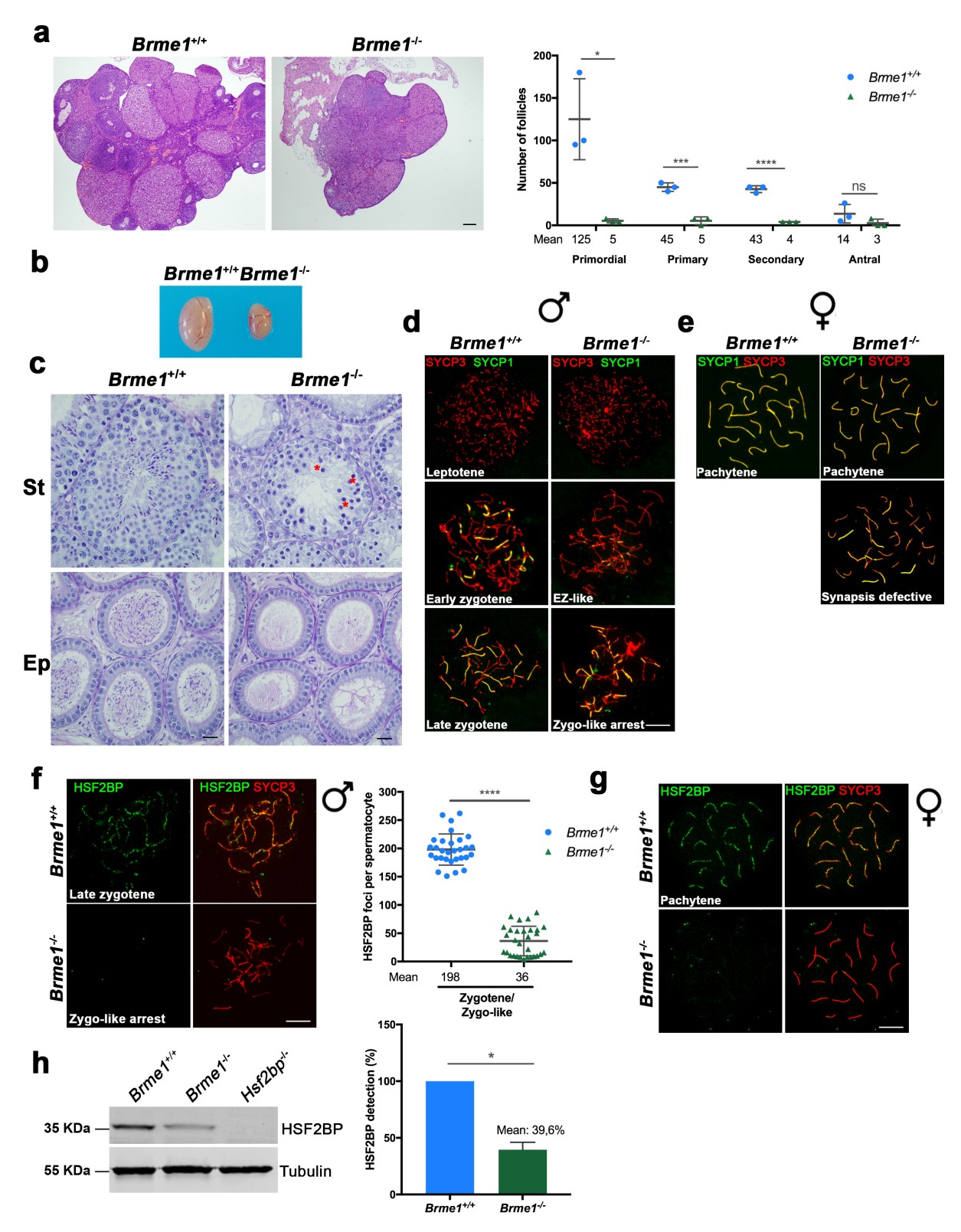

**Figure 8.** *Brme1⁻/⁻* mice show severe fertility defects. (**a**) Hematoxylin+eosin stained sections of ovaries from adult *Brme1⁻/⁻* females showing a strong depletion of follicles. Plot on the right represents the quantification in 3 months-old females. Ovaries: n = 3 ovaries for each genotype. Two-tailed Welch's t-test analysis: *p<0.05, ***p<0.001, ****p<0.0001. Bar in panels, 50 µm. (**b**) Testes from adult *Brme1⁻/⁻* males show a strong reduction of the testis size. See quantification of testis weight/body weight at ***Figure 2—figure supplement 2b***. (**c**) Spermatogenesis is arrested at epithelial stage IV in
*Figure 8 continued on next page*

*Figure 8 continued*

*Brme1*$^{-/-}$ as shown in PAS+hematoxylin stained testis sections. Massive apoptosis of spermatocytes is indicated (red asterisks). The spermatogenic arrest leads to empty epididymides and non-obstructive azoospermia. (St) Seminiferous tubules. (Ep) Epididymides. Bar in panels, 10 μm. (d–e) Double labeling of (d) spermatocyte and (e) oocyte spreads from WT and *Brme1*$^{-/-}$ mice with SYCP3 (red) and SYCP1 (green). *Brme1*$^{-/-}$ spermatocytes arrest in a zygotene-like stage and show synapsis between non-homologous chromosomes. (e) *Brme1*$^{-/-}$ females showed a subset of fully-synapsed pachynemas (18.5 dpc) but increased numbers of synapsis-defective cells. See extended panel for females at *Figure 8—figure supplement 1e*. Bar in panels, 10 μm. (f–g) Double labeling with HSF2BP (green) and SYCP3 (red) of (f) spermatocyte and (g) oocyte spreads from *Brme1*$^{-/-}$ mice showing faint HSF2BP labeling in spermatocytes and total absence of labeling in oocytes. Plot on the right of (f) panel represents de quantification of HSF2BP foci in *Brme1*$^{-/-}$ spermatocytes. Nuclei: n = 30 zygonemas/zygonemas-like from two adult mice of each genotype (*Brme1*$^{+/+}$ values from *Figure 3a*) Two-tailed Welch's t-test analysis: ****$p<0.0001$. Bar in panels, 10 μm. (h) Western blot analysis of protein extracts from 13 dpp WT, *Brme1*$^{-/-}$ and *Hsf2bp*$^{-/-}$ testes with a specific antibody against HSF2BP. Tubulin was used as loading control. Graph on the right represents the relative quantification of the immunoblotting. Mice: n = 2 *Brme1*$^{+/+}$, *Brme1*$^{-/-}$ and *Hsf2bp*$^{-/-}$. Two-tailed Welch's t-test analysis: *$p<0.05$.

The online version of this article includes the following figure supplement(s) for figure 8:

**Figure supplement 1.** Generation and genetic characterization of *Brme1* knock-out mice.

interaction between HSF2BP and BRME1, we tested whether HSF2BP localization depended on BRME1 by immunolabeling of HSF2BP in *Brme1*$^{-/-}$ spermatocytes and oocytes. Our results showed a strong reduction of HSF2BP staining in BRME1-null spermatocytes (*Figure 8f*) and a total absence in oocytes (*Figure 8g*). Western blot analysis of HSF2BP in 13 dpp testis extracts from *Brme1*$^{-/-}$ mice showed a strong reduction in comparison with the WT control (*Figure 8h*), suggesting again that BRME1 is necessary for HSF2BP protein stabilization.

Immunostaining of *Brme1*$^{-/-}$ spermatocytes for γH2AX, RPA, the recombinases RAD51 and DMC1 or SPATA22 revealed an accumulation of γH2AX and RPA on zygonema-like spermatocytes (*Figure 9—figure supplement 1a–b*), a drastic reduction of RAD51/DMC1 foci in early and late zygonema (*Figure 9a and c*, *Figure 9—figure supplement 2a and c*) and a strong accumulation of SPATA22 (*Figure 9e* and *Figure 9—figure supplement 2e*). According to the meiotic arrest at the zygotene-like stage, MLH1 staining revealed a total absence of COs (*Figure 9g*). As in males, *Brme1*$^{-/-}$ oocytes showed an accumulation of γH2AX (*Figure 9—figure supplement 1c*) and SPATA22 (*Figure 9f* and *Figure 9—figure supplement 2f*) and a reduced staining of DMC1 and RAD51 leading to a reduced number of COs (measured as interstitial CDK2; *Figure 9b, d and h*, *Figure 9—figure supplement 2b and d*). However, *Brme1*$^{-/-}$ oocytes did not show RPA accumulation (*Figure 9—figure supplement 1d*). These results are similar to the phenotypes described for HSF2BP mutants (*Figures 4*, *5* and *6*, *Figure 4—figure supplement 1*, *Figure 5—figure supplement 1*, *Figure 6—figure supplement 1* and (*Brandsma et al., 2019*) see *Supplementary file 1f* for a complete comparison among mutants and their meiotic alterations). Thus, both HSF2BP and BRME1 mutant mice show a highly similar phenotype including sexual dimorphism.

## BRME1 and HSF2BP form a multimeric complex with PALB2 and BRCA2

To further delineate the interactome of BRME1, we immuno-precipitated BRME1 from testis extracts coupled to mass-spectrometry. We identified as expected HSF2BP as the main interactor, but also BRCA2, PALB2, RAD51 and RPA, strongly suggesting that they form a large multimeric complex (*Supplementary files 1g-1h*). For validation, we transfected the corresponding expression plasmids in HEK293T cells for co-IP analysis. BRME1 co-immunoprecipitated with BRCA2 and HSF2BP when they were all co-transfected, but importantly BRME1 alone did not co-immunoprecitate with BRCA2 (*Figure 10a–b*). The reciprocal co-IP of BRCA2 with HSF2BP and BRME1 was also positive. We also observed modest but positive co-IP of HSF2BP with RPA, PALB2 and RAD51; and of BRME1 with RAD51 and RPA but not with PALB2 (*Figure 10—figure supplement 1a*). These interactions were further analyzed in a cell-free TNT system coupled to co-immunoprecipitation assays. We observed an absence of direct interaction between any of them, with the exception of BRME1 and HSF2BP, as expected from the Y2H analysis (*Figure 10—figure supplement 1b*). These results suggest that these proteins belong to a complex (or complexes) in vivo (likely through BRCA2) and that the HSF2BP-S167L variant could be altering BRME1 interaction with partners of major BRCA2-containing recombination complexes.

Finally, given the interaction of BRME1 and HSF2BP, we analysed their interdependence in U2OS cells. Transfected HSF2BP was localized diffusely in the cytoplasm and the nucleus (*Figure 10c*).

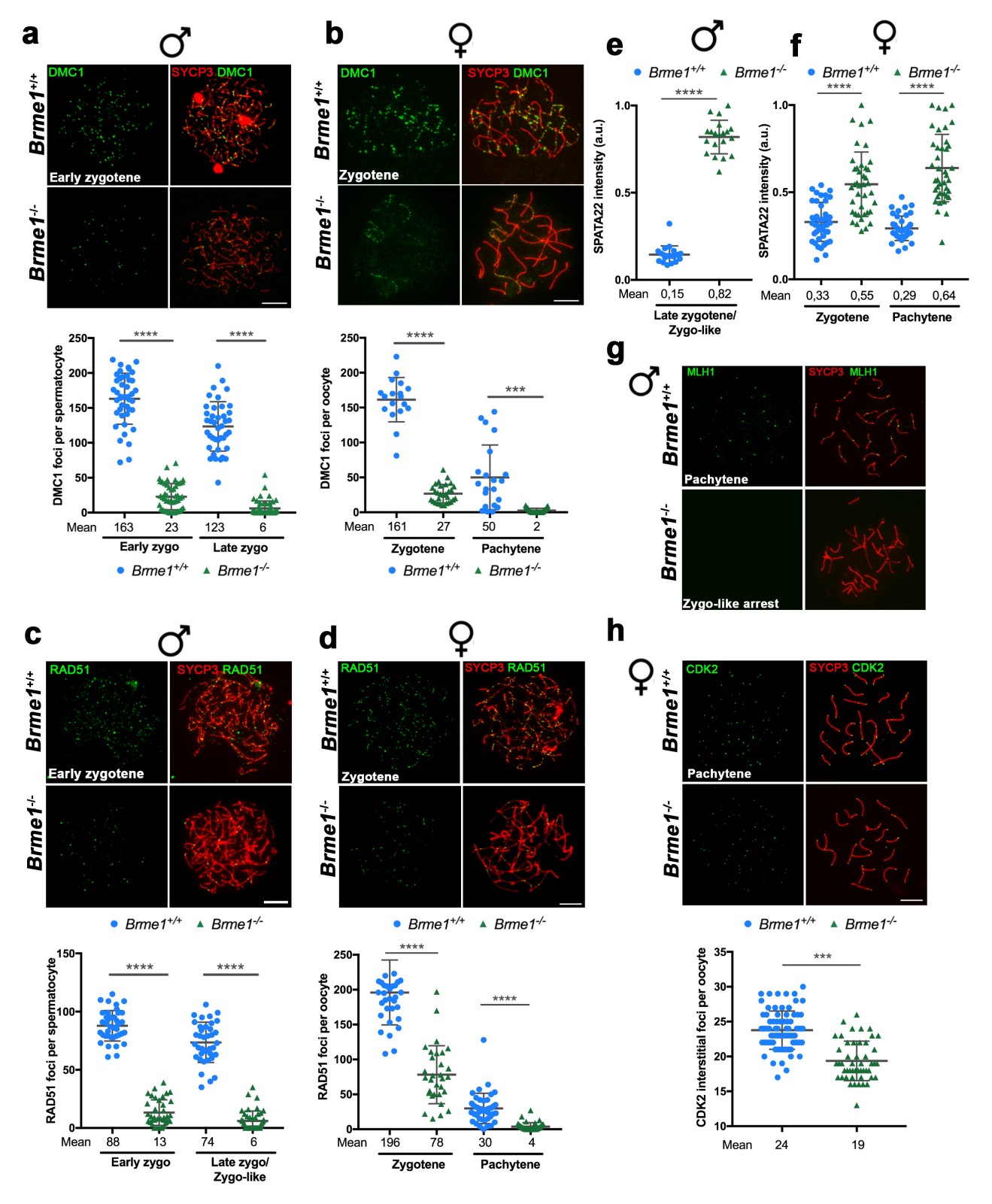

**Figure 9.** BRME1 is essential for meiotic recombination. (**a–b**) Double immunofluorescence of DMC1(green) and SYCP3 (red) in *Brme1*^{+/+} and *Brme1*^{-/-} (**a**) spermatocytes and (**b**) oocytes showing a reduction in the number of DMC1 foci. Plots under each panel represent the quantification. See also extended panels on *Figure 9—figure supplement 2a–b*. Male nuclei for DMC1: *Brme1*^{+/+}/*Brme1*^{-/-} n = 43/50 early and 41/52 late zygonemas/ zygonemas like from two adult mice of each genotype (*Brme1*^{+/+} values from *Figure 5a*). Female nuclei for DMC1: *Brme1*^{+/+}/*Brme1*^{-/-} n = 18/30

*Figure 9 continued on next page*

*Figure 9 continued*

zygonemas and 21/31 pachynemas from two embryos (17.5 dpc) of each genotype (*Brme1$^{+/+}$* values from *Figure 5c*). Two-tailed Welch's t-test analysis: ***p<0.001, ****p<0.0001. Bar in panels, 10 μm. (**c–d**) Double immunofluorescence of RAD51 (green) and SYCP3 (red) in *Brme1$^{+/+}$* and *Brme1$^{-/-}$* (**c**) spermatocytes and (**d**) oocytes showing a reduction in the number of RAD51 foci in the *Brme1$^{-/-}$* in comparison to the WT. Plots under each panel represent the quantification. See also extended panels on *Figure 9—figure supplement 2c–d*. Male nuclei for RAD51: *Brme1$^{+/+}$*/*Brme1$^{-/-}$* n = 39/39 early and 37/45 late zygonemas/zygonemas like from two adult mice of each genotype (*Brme1$^{+/+}$* values from *Figure 5b*). Female nuclei for RAD51: *Brme1$^{+/+}$*/*Brme1$^{-/-}$* n = 35/31 zygonemas and 42/40 pachynemas from two embryos (17.5 dpc) of each genotype (*Brme1$^{+/+}$* values from *Figure 5d*). Two-tailed Welch's t-test analysis: ****p<0.0001. (**e–f**) Quantification of SPATA22 intensity in (**e**) spermatocytes and (**f**) oocyte spreads from *Brme1$^{+/+}$* and *Brme1$^{-/-}$*. See immunofluorescences in *Figure 9—figure supplement 2e–f*. Nuclei: Males, n = 20 cells from two adult mice of each genotype (*Brme1$^{+/+}$* from *Figure 6a*). Females, *Brme1$^{+/+}$*/*Brme1$^{-/-}$* n = 41/40 zygonemas and 40/40 pachynemas from two embryos of each genotype (17.5 dpc) (*Brme1$^{+/+}$* from *Figure 6a*). Two-tailed Welch's t-test analysis: ****p<0.0001. (**g**) Double immunofluorescence of MLH1 (green) and SYCP3 (red) in *Brme1$^{+/+}$* and *Brme1$^{-/-}$* spermatocytes showing the absence of MLH1 labeling in the knock-out. (**h**) Double labeling of CDK2 (green) and SYCP3 (red) in oocyte spreads from 17.5 dpc *Brme1$^{+/+}$* and *Brme1$^{-/-}$* embryos. During meiotic prophase I, CDK2 localizes to the telomeres of chromosomes from leptotene to diplotene. However, around mid-pachytene additional interstitial CDK2 signals appear at CO sites, colocalizing with MLH1. As a measure of COs, just interstitial CDK2 foci (non-telomeric) have been counted. *Brme1$^{-/-}$* females show a strong reduction in the number of COs. Plot under the panel show the quantification. Nuclei: *Brme1$^{+/+}$*/*Brme1$^{-/-}$* n = 79/49 from three embryos (17.5 dpc) in WT and two embryos in *Brme1$^{-/-}$* (*Brme1$^{+/+}$* from *Figure 6c*). Two-tailed Welch's t-test analysis: ***p<0.0001. Bar in all panels, 10 μm.

The online version of this article includes the following figure supplement(s) for figure 9:

**Figure supplement 1.** DSBs are formed but not properly repaired in *Brme1*-deficient mice and mimic the phenotype of *Hsf2bp*-deficient mice.

**Figure supplement 2.** Altered dynamic of recombinational proteins in the absence of BRME1.

However, when HSF2BP was co-overexpressed with BRME1, its pattern changed to an intense nucleoplasm staining with nuclear invaginations that resemble nucleoplasmic reticulum (*Figure 10c*; *Malhas et al., 2011*). Interestingly, such invaginations were reduced when BRME1 was co-transfected with HSF2BP-S167L (*Figure 10c*). In addition, the intensity of the fluorescence signal of HSF2BP-S167L was lower than for HSF2BP-WT and both intensities increased when HSF2BP was co-expressed with BRME1 (*Figure 10c*). Western blot analysis indicated a reduced protein stability of the S167L variant (*Figure 10d*), in agreement with the results observed in vivo (*Hsf2bp$^{S167L/S167L}$* mutant, *Figure 3*). Interestingly, the protein expression level of transfected HSF2BP increased when co-transfected with BRME1 and was partially dependent on proteasome degradation (*Figure 10d*), indicating a role of BRME1 in stabilizing HSF2BP. Taken altogether and given the low protein expression of BRME1 in the *Hsf2bp$^{S167L/S167L}$*, these results suggest a functional interdependence between BRME1 and HSF2BP that leads to their lower protein stability/expression in mutant meiocytes, which might induce recombination defects.

## Discussion

Using exome sequencing, we identified the S167L missense variant in *HSF2BP* in a consanguineous family with three cases of POI with secondary amenorrhea. All affected family members are homozygous for the variant, and the healthy relatives are heterozygous carriers. The causality of the HSF2BP-S167L variant is supported by the meiotic phenotype and the subfertility observed in *Hsf2bp$^{S167L/S167L}$* female mice. Furthermore, the DNA repair defects in murine *Hsf2bp$^{S167L/S167L}$* meiocytes, displayed by the reduced number of RAD51/DMC1 foci on DSBs and the subsequent reduction in the number of COs, provide evidence that this missense variant alters meiotic recombination. This conclusion was further supported by the comparative analysis of the S167L allele with the *Hsf2bp* null allele, which revealed that the missense variant can be considered as a hypomorphic allele. This is in agreement with the secondary amenorrhea observed in the patients, and the residual (medically-assisted) fertility in one of the affected sisters. Our identification of *HSF2BP* as a gene implicated in POI is in line with recent reports of POI-causing variants in genes that are required for DNA repair and recombination, such as *MCM8, MCM9, SYCE1, MSH4, PSMC3IP, FANCM* or *NBN* (*AlAsiri et al., 2015*; *Carlosama et al., 2017*; *de Vries et al., 2014*; *Fouquet et al., 2017*; *He et al., 2018*; *Tenenbaum-Rakover et al., 2015*; *Tucker et al., 2018*; *Wood-Trageser et al., 2014*; *Zangen et al., 2011*).

Meiotic mouse mutants often exhibit sexually dimorphic phenotypes (*Cahoon and Libuda, 2019*). These differences can have a structural basis, given that the organization of the axial elements is known to be different between sexes. This is supported by the difference in length of the axes and

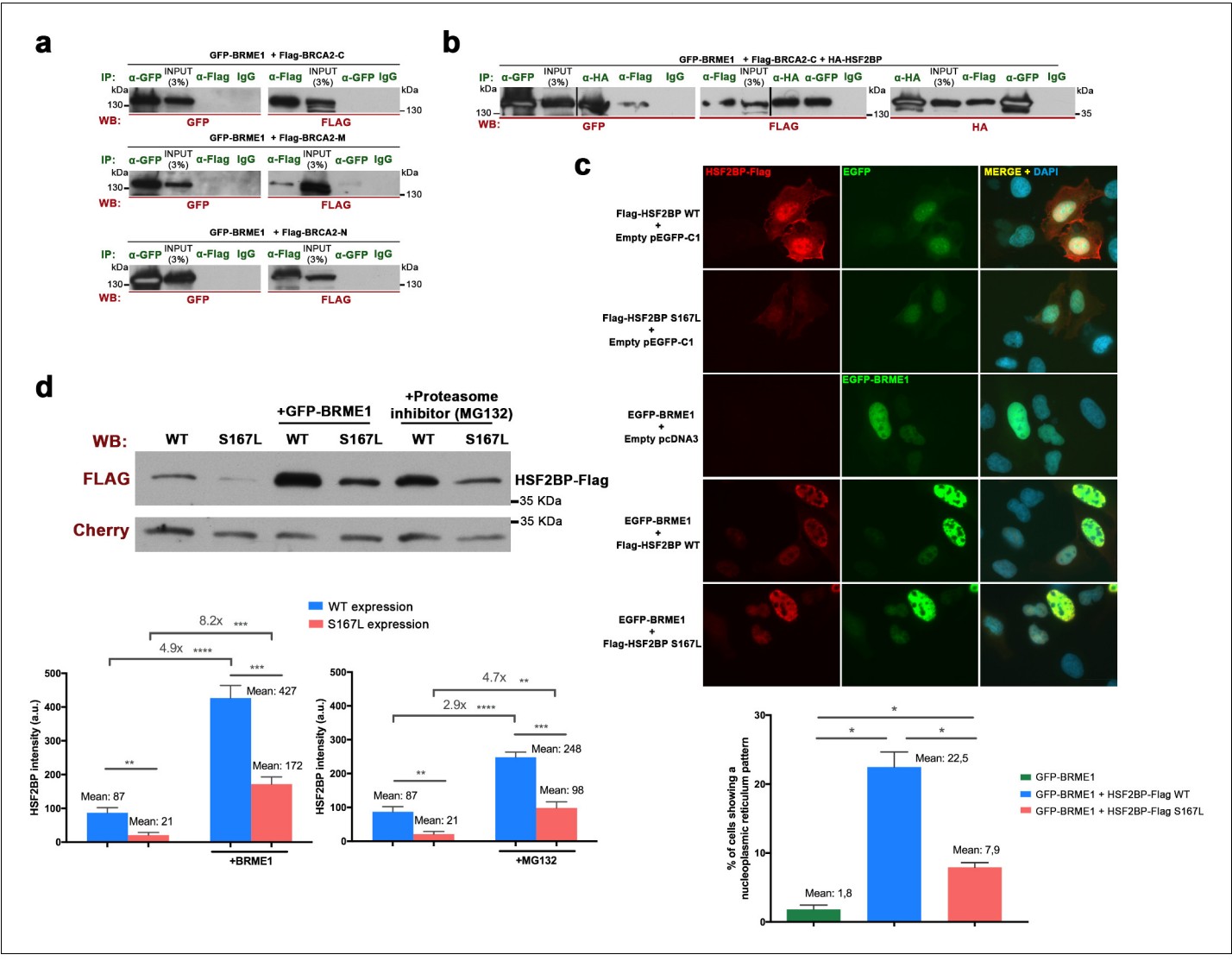

**Figure 10.** BRME1 forms a complex with BRCA2 and HSF2BP and stabilizes HSF2BP. (a–b) HEK293T cells were co-transfected with GFP-BRME1, Flag-BRCA2-C and HA-HSF2BP. Protein complexes were immunoprecipitated (IP: green text) with either an anti-Flag, anti-EGFP, anti-HA or IgGs, and analyzed by western blot with the indicated antibody (WB: red text). (a) BRME1 does not co-immunoprecipitate with BRCA2-N, BRCA2-M or BRCA2-C. (b) In the presence of HA-HSF2BP (triple co-transfection) BRCA2-C and BRME1 coimmunoprecipitate (co-IPs between HSF2BP and BRCA2-C are shown in *Figure 6—figure supplement 2a*). (c) Transfected U2OS cells with plasmids encoding Flag-HSF2BP (WT or S167L) and EGFP-BRME1 alone or together were immuno-detected with antibodies against Flag (red) and EGFP (green). Transfected HSF2BP (WT and S167L) labels the whole cell (S167L less intense) whereas BRME1 shows nuclear localization. When co-expressed, BRME1 and HSF2BP change their patterns and form nuclear invaginations that resemble nucleoplasmic reticulum. This phenotype is milder in the presence of HSF2BP-S167L than with the WT (graph under the panel: quantification of the number of cells showing a nucleoplasmic reticulum pattern). n > 400 cells from two independent transfections of each condition. Two-tailed Welch's t-test analysis: *p<0.05. Bar in panel, 20 μm. (d) HEK293T cells were transfected with Flag-HSF2BP (WT and S167L) alone or with GFP-BRME1. Additionally, cells transfected with Flag-HSF2BP were treated with the proteasome inhibitor (MG132, 10 μM) and analyzed by western blot. Cherry was used as transfection efficiency control. HSF2BP-WT was expressed at higher levels than HSF2BP-S167L and their detection (both the WT and the S167L variant) was increased when co-transfected with BRME1. The increase was greater for the HSF2BP-S167L variant in comparison with the WT. Incubation with MG132 increased the detection levels of transfected HSF2BP mimicking the effect of co-transfecting BRME1. n = 3 independent transfections for each condition. Two-tailed Welch's t-test analysis: **p<0.01, ***p<0.001, ****p<0.0001.

The online version of this article includes the following figure supplement(s) for figure 10:

**Figure supplement 1.** Co-immunoprecipitations of BRME1, HSF2BP, RPA, RAD51 and PALB2 expressed from transfected HEK293 cells and from TNT assays.

by the essential role that the meiotic cohesin subunit RAD21L plays in males but not in females (*Herrán et al., 2011*). In general, meiotic recombination mutants appear to proceed further in female than in male, because in males the asynapsis of the sex bivalent leads to a loss of silencing of the Y chromosome, and perhaps also because of the presence of less stringent checkpoints in oogenesis (*Hunt and Hassold, 2002*). A sexually dimorphic phenotype is also observed here in our HSF2BP-S167L mutant mice. Both sexes show a significant decrease in the number of COs and consequently an increase in the frequency of meiocytes showing bivalents without CO. This leads to a significant reduction of spermatozoa in the epididymis, while the number of oocytes and their distribution in the follicular pool are not affected in females. Our failure to detect any significant impact of the variant on male fertility could be due to the high variability in litter frequency and size in our study. However, it is known that a strong reduction of the spermatozoa count (up to 60%) does not affect male mouse fertility (*Schürmann et al., 2002*), which would explain the normal fertility of male mice bearing the HSF2BP-S167L variant despite the presence of overt meiotic alterations. By contrast, female mice with the HSF2BP-S167L variant show a mild sub-fertility phenotype with a reduction of litter frequency. This can be due to the very much fewer gametes available for fertilization in females in comparison to males but also to molecular differences in the meiotic recombination process in both sexes (*Cahoon and Libuda, 2019*), as displayed by the absence of RPA accumulation or the more pronounced decrease in DMC1 foci observed in HSF2BP-S167L oocytes.

Although we cannot exclude that a POI-like phenotype would appear over time in HSF2BP-S167L female mice, the sub-fertility observed in HSF2BP-S167L females appears to be milder in comparison to the phenotype of the human patients. This could be explained by a lower sensitivity of mice to hypomorphic alleles compared to humans, in a similar manner to the known lower gene dosage sensitivity of the former in the context of genes that are haploinsufficient in human (*Veitia, 2003*). In addition, the initial events of human female meiosis appear to be more error-prone than in mice, or even than human males, as evidenced by the increased incidence of synaptic defects in the human oocytes or the fact that MLH1 foci appear much earlier in prophase I (*Hassold et al., 2007*). Furthermore, it has recently been shown that human oocytes exhibit a specific CO maturation inefficiency (*Wang et al., 2017*). Indeed, despite a higher total number of COs in women, the frequency of bivalents without CO is paradoxically higher in women than in men. Altogether, these observations could explain the stronger phenotype of the human POI patients compared to the $Hsf2bp^{S167L/S167L}$ female mice.

The absence of homozygous male carriers in the consanguineous family studied here prevents the direct comparison of the impact of the HSF2BP-S167L variant on fertility phenotypes between human and mouse males. Although future studies might identify infertile men homozygous for the S167L variant, it is interesting to note that another variant of HSF2BP (G224*) was shown to affect recombination rate in males and that two siblings homozygous for this HSF2BP variant in the analyzed Icelandic population were healthy but without descendants, suggesting they were infertile (*Halldorsson et al., 2019*). This reinforces a conserved function of HSF2BP in human male fertility.

We have shown through a biochemical analysis that BRME1 immunoprecipitates mainly with its partner HSF2BP (constituting or belonging to a complex) but also with PALB2, RAD51, RPA and BRCA2 (in testis extracts). These interactions could possibly be mediated by the multidomain hub protein BRCA2 (*Siaud et al., 2011*) as HSF2BP interacts directly with BRCA2 (*Brandsma et al., 2019*) and with BRME1 (this work). In addition, BRCA2 also directly interacts with the DSBs recruiter PALB2, with the recombinases RAD51 and DMC1 (through different specific domains), and with DNA (*Siaud et al., 2011*). This BRCA2-containing complex participates in the orderly orchestration of events at DSBs such as the initial binding of RPA to the resected DNA, the exchange of RPA by RAD51/DMC1, and the loading of the MEIOB-SPATA22 complex to the RPA complexes (*Martinez et al., 2016*; *Zhao et al., 2015*). Interestingly, genes with recently identified variants in POI patients are implicated in the repair of induced DSBs at the early stages of meiosis and encode BRCA2-interacting factors, such as *MEIOB, DMC1* or *BRCA2* itself (*Caburet et al., 2019a*; *Caburet et al., 2020*; *Caburet et al., 2019b*; *He et al., 2018*). This highlights the crucial importance and the high sensitivity of this particular meiotic step, and the hub role of BRCA2 as a tightly regulated platform for correct meiotic recombination.

We have also shown by several complementary approaches that the proteins HSF2BP/BRME1 constitute in vivo a functional complex in which both subunits are essential for meiotic recombination and for their mutual protein expression and/or stability in vivo. Accordingly, the genetic depletion of

BRME1 or HSF2BP leads to similar if not identical phenotypes in which oogenesis is altered with severe defects in chromosome synapsis that promotes premature loss of ovarian follicles and spermatogenesis is arrested at zygotene-like stage resulting in a lack of spermatozoa. These meiocytes are not able to load the recombinases RAD51/DMC1, impairing the proper repair of DSBs leading to the generation of no COs or very few in males and females, respectively. As a consequence, zygonema-like spermatocytes accumulate the single strand binding proteins SPATA22 and RPA, whereas oocytes accumulates only SPATA22. During the course of the reviewing of this work, three *Brme1* knockouts have been described (*Zhang et al., 2020*; *Shang et al., 2020*; *Takemoto et al., 2020*). All the described male mutants show strong fertility defects and similar molecular alterations although with different severity. In females, the two works that address their analysis (Shang et al. and Takemoto et al.), describe normal fertility which is in contrast with the strong reduction in the follicle pool and meiotic defects observed in our *Brme1*$^{-/-}$ females. The higher severity of our male and female mutants could be explained on the basis of the different genetic background of the mice given that all of them are apparently similar.

The S167L human recessive POI variant behave as a hypomorphic allele in mice, which results in a reduction of the protein expression/stability of itself and of its partner BRME1 in vivo and in transfected cells. As a consequence, both male and female *Hsf2bp*$^{S167L/S167}$ mice show a similar but milder phenotype than that of the *Hsf2bp*$^{-/-}$ or *Brme1*$^{-/-}$, consisting in a reduction in the spermatozoa count while being fertile and a subtle reduction in female fertility. Molecularly, the reduction observed in the meiocytes of the mutant *Hsf2bp*$^{S167L/S167L}$ mice of RAD51/DMC1, the reduction of COs (both in males and females), the accumulation of RPA (only in males) and SPATA22 (in males and females) are also weaker than in the null mutant. The observed accumulation of RPA in males is likely to occur at the early stages of recombination because SPATA22 loading to the DSBs is also increased in the mutants of HSF2BP and BRME1. RPA, as part of a trimeric replication protein complex (RPA1-RPA2-RPA3), binds and stabilizes ssDNA intermediates that form during DNA repair. In meiosis, RPA is also forming a complex with two other essential meiotic players MEIOB (homologue of RPA) and SPATA22. However, the loading of this complex to DSBs is RPA-independent (*Shi et al., 2019*). It has been postulated that RPA functions in meiosis at two different stages; (i) during the early recombination stages when the DSBs ends are resected by the MRN complex and (ii) during the strand invasion into the homologous duplex that is carried out by RAD51/DMC1 and ssDNA is generated at the displacement loops (*Shi et al., 2019*). The observed lack of DNA-binding ability of HSF2BP/BRME1 points towards a model in which the absence of the complex HSF2BP/BRME1 through a direct interaction with BRCA2 impairs the replacement of RPA by RAD51/DMC1 in the foci that form on the DSBs of the spermatocytes. Similarly, the reduced expression at the protein level of HSF2BP/BRME1 as a consequence of the POI variant, which does not affect their heterodimerization, would make them less proficient in replacing RPA in the spermatocytes by the recombinases RAD51/DMC1 leading to a lower frequency of COs. Given the unknown function that RPA plays in vivo during oogenesis (*Shi et al., 2019*), it is tempting to speculate that the role of RPA in mediating the replacement of RAD51/DMC1 in female meiosis would be carried out by another protein complex such as SPATA22/MEIOB in a HSF2BP/BRME1-dependent manner.

Very recently, a high-resolution genome-wide recombination map revealed novel loci involved in the control of meiotic recombination and highlighted genes involved in the formation of the SC (SYCE2, RAD21L, SYCP3, SIX6OS1) and the meiotic machinery itself as determinants of COs (*Halldorsson et al., 2019*). Within the second category, variants of the SUMO ligase RNF212 and the ubiquitin ligase HEI10 have been largely documented as genetic determinants of the recombination rate in humans and, importantly, so were variants of HSF2BP. Consequently, gene dosage of RNF212 and HEI10 affects CO frequency through their activity in CO designation and maturation (*Lake and Hawley, 2013*; *Reynolds et al., 2013*). We found that both BRME1 and HSF2BP localization are unaffected in the loss-of-function mouse mutants of *Rnf212* and *Hei10 (Ccnb1ip1)*. This observation together with the proper co-localization of BRME1/HSF2BP with RPA allows us to map these proteins upstream in the recombination pathway.

It is worth noting that some of the genes affecting the recombination rate have also been described as 'fertility genes', such as *SYCP3, HFM1* and *HSF2BP* (*Geisinger and Benavente, 2017*; *Primary Ovarian Insufficiency Collaboration et al., 2014* and this work). Altogether, we propose that different variants of the same meiotic gene (alleles responsible for mild or strong phenotypes) can give rise to either an altered genome-wide recombination rate with no detrimental effect, or

cause infertility when the decreased recombination rate falls below the lower limit of one COs per bivalent. In the present POI family, the S167L variant in HSF2BP seems to be below that limit. To our knowledge, HSF2BP is one of the very few human genes with variants known to affect both the genome-wide recombination rate in the human population and meiotic chromosome missegregation (fertility) through a reduction of the recombination rate (*Halldorsson et al., 2019*). Along similar lines, it is conceivable that variants with additive effects (*Schimenti and Handel, 2018*) can lead to a genome-wide reduction of the recombination rate and thus to aneuploidy and infertility. Specifically, variants in genes involved in meiotic recombination and SC constituents could be responsible for a large fraction of genetic infertilities. These variants should be under purifying selection and would be removed or substantially reduced from the population. However, this is not the case for genes with sexual phenotypic dimorphism (*Gershoni and Pietrokovski, 2014*) as is apparent for a wide number of meiotic genes (*Cahoon and Libuda, 2019*), including HSF2BP and BRME1, where individuals of one of the sexes are fertile carriers.

In summary, we describe for the first time a human family where POI co-segregates with a genetic variant in HSF2BP (S167L) in a Mendelian fashion. Humanized mice reveal that the HSF2BP variant is a hypomorphic allele that promotes the lower protein expression and/or stability of the HSF2BP/BRME1 complex and phenocopy in a milder manner the meiotic defects observed in mice lacking either HSF2BP or its direct interactor BRME1.

## Materials and methods

### Whole exome sequencing

Written informed consent was received from participants prior to inclusion in the study and the institutions involved. Genomic DNA was extracted from blood samples by standards protocols.

For individuals III-3 and III-10, library preparation, exome capture, sequencing and initial data processing were performed by Beckman Coulter Genomics (Danvers, USA). Exon capture was performed using the hsV5UTR kit target enrichment kit. Libraries were sequenced on an Illumina HiSEQ instrument as paired-end 100 bp reads. For individual III-2, library preparation, exome capture, sequencing and data processing were performed by IntegraGen SA (Evry, France) according to their in-house procedures. Target capture, enrichment and elution were performed according to manufacturer's instructions and protocols (SureSelect Human All Exon Kits Version CRE, Agilent). The library was sequenced on an Illumina HiSEQ 2500 as paired-end 75 bp reads. Image analysis and base calling was performed using Illumina Real Time Analysis (RTA 1.18.64) with default parameters.

### Bioinformatic analysis

For the three individuals, sequence reads were mapped onto the human genome build (hg38/GRCh38) using the Burrows-Wheeler Aligner (BWA) tool. Duplicated reads were removed using sambamba tools. Whole exome sequencing metrics are provided in *Supplementary file 1a*. Variant calling, allowing the identification of SNV (Single Nucleotide Variations) and small insertions/deletions (up to 20 bp) was performed via the Broad Institute GATK Haplotype Caller GVCF tool (3.7). Ensembl VEP (Variant Effect Predictor, release 87) program was used for initial variant annotation. This tool considers data available in dbSNP (dbSNP147), the 1000 Genomes Project (1000G_phase3), the Exome Variant Server (ESP6500SI-V2-SSA137), the Exome Aggregation Consortium (ExAC r3.0), and IntegraGen in-house databases. Additional annotation data was retrieved using dbNSFP (version 3.5, https://sites.google.com/site/jpopgen/dbNSFP) and Varsome (https://varsome.com/). Minor allele frequencies were manually verified on GnomAD (http://gnomad.broadinstitute.org), ISB Kaviar (http://db.systemsbiology.net/kaviar/), and Great Middle Eastern variant database GME Variome (http://igm.ucsd.edu/gme/).

Variant filtering was performed on the following criteria:

- minimum depth at variant position of 10,
- correct segregation in the family, on the basis of homozygosity by descent: variants should be homozygous in both affected sisters III-2 and III-3, and heterozygous or homozygous for Reference allele in the fertile sister III-10,
- absence in unrelated in-house fertile controls,

- Minor Allele Frequency (MAF) below 1% in global and in each population in the GnomAD database,
- presence in the coding sequence (i.e not in UTRs, introns, intergenic,.)
- high predicted functional impact on the protein. Impact was evaluated based on the predictors included in dbNSFP3.5 (*Cahoon and Libuda, 2019*; *Carlosama et al., 2017*; *Cox and Mann, 2008*) (considered as pathogenic when the majority of the predictors agreed).

The number of variants fulfilling those criteria is provided in *Supplementary file 1b*. Visual inspection of the variant was performed using the IGV viewer.

## Sanger sequencing analysis

To confirm the presence and segregation of the variant, direct genomic Sanger DNA sequencing of *HSF2BP* was performed in the patients, the parents and non-affected siblings using specific primers: HSF2BP-ex6F: 5'-ctagaatcttctgtatcctgca-3' and HSF2BP-ex6R2: 5'-ggtctggaagcaaacaggcaa-3'. The resulting chromatograms are shown in *Figure 1—figure supplement 1*.

## Predictions of pathogenicity and sequence conservation

The S167L variant was predicted to be pathogenic or deleterious and highly conserved by 11 out of the 18 pathogenicity predictors available in dbNSFP 3.5 (*Supplementary file 1c*). Upon verification, it appears that the conflicting interpretation of this variant might stem from the single occurrence of a Leu at this position in zebrafish. As the change in zebrafish is the variant that we have in the human family, we checked all the available sequences (Ensembl Release 99, January 2020, removing the one-to-many relationships). Ser167 is very highly conserved in mammals, birds and reptiles and fish and is present in 208 of 212 orthologous sequences (*Figure 1—figure supplement 2* and *Figure 1—figure supplement 3*).

## Generation of CRISPR/Cas9-edited mice

For developing all the mutant mice models (*Hsf2bp*$^{-/-}$, *Hsf2bp*$^{S167L/S167L}$, *Brme1*$^{\Delta142-472/\Delta \Delta142-472}$, *Brme1*$^{-/-}$, *Spo11*$^{-/-}$, *Rnf212*$^{-/-}$ and *Hei10*$^{-/-}$) the different crRNAs were predicted at https://eu.idtdna.com/site/order/ designtool/index/CRISPR_CUSTOM. The crRNAs, the tracrRNA and the ssODNs were produced by chemical synthesis at IDT (crRNAs and ssODNs sequences are listed in *Supplementary files 1i-1j*). For the *Hsf2bp*$^{S167L}$ we introduced a mutation in the mouse counterpart residue (p.Ser171Leu) of the POI mutation found in the clinical case (p.Ser167Leu). However, for the shake of simplicity, on this manuscript we refer to the mutant allele by the acronym of the human mutation (S167L). The ssODN contains the mutation on the corresponding position of the mouse sequence (c.512C > T, p.Ser171Leu, see character in red in *Supplementary file 1j*) and the PAM mutations avoiding amino acid changes (see characters in bold in the *Supplementary file 1j*). For the *Spo11*$^{-/-}$ mice generation, the ssODN contains the mutations in the active site (TACTAC >TTCTTC p.YY137-138FF, see *Supplementary file 1j*) and the PAM mutations (bold characters in *Supplementary file 1j*). In all cases the crRNA and tracrRNA were annealed to obtain the mature sgRNA. A mixture containing the sgRNAs, recombinant Cas9 protein (IDT) and the ssODN (30 ng/μl Cas9, 20 ng/μl of each annealed sgRNA and 10 ng/μl ssODN) were microinjected into B6/CBA F2 zygotes (hybrids between strains C57BL/6J and CBA/J) (*Singh et al., 2015*) at the Transgenic Facility of the University of Salamanca. Edited founders were identified by PCR amplification (Taq polymerase, NZYtech) with primers flanking the edited region (see *Supplementary file 1k* for primer sequences). The PCRs products were direct sequenced or subcloned into pBlueScript (Stratagene) followed by Sanger sequencing, selecting the founders carrying the desired alelles. The selected founders were crossed with wild-type mice to eliminate possible unwanted off-targets. Heterozygous mice were re-sequenced and crossed to give rise to edited homozygous. Genotyping was performed by analysis of the PCR products produced from genomic DNA extracted from tail biopsies. The primers and the expected amplicon sizes are listed in the *Supplementary file 1k*. Mouse mutants for *Rec8*, *Six6os1* and *Psma8* have been previously described (*Bannister et al., 2004*; *Gómez-H et al., 2019*; *Gómez-H et al., 2016*).

## Ethics statement
All the experiments were approved by the Ethics Committee for Animal Experimentation of the University of Salamanca (USAL) and the Ethics committee of the Spanish Research Council (CSIC) under protocol #00–245. Accordingly, all the mouse protocols used in this work have been approved by the Animal Experimentation committees mentioned above. Specifically, mice were always housed in a temperature-controlled facility (specific pathogen free, spf) using individually ventilated cages, standard diet and a 12 hr light/dark cycle, according to EU law (63/2010/UE) and the Spanish royal law (53/2013) at the "Servicio de Experimentación Animal, SEA. In addition, animal suffering was always minimized, and we made every effort to improve animal welfare during the life of the animals. The mice analysed were between 2 and 4 months of age, except in those experiments where the age is indicated.

## Histology
For histological analysis, after the necropsy of the mice their testes or ovaries were removed and fixed in Bouin´s fixative or formol 10%, respectively. They were processed into serial paraffin sections and stained with haematoxylin-eosin (ovaries) or Periodic acid–Schiff (PAS) and hematoxylin (testes). The samples were analysed using a microscope OLYMPUS BX51 and images were taken with a digital camera OLYMPUS DP70. For TUNEL assay, sections were deparaffinized and apoptotic cells were detected with the In Situ Cell Death Detection Kit (Roche) and counterstained with DAPI.

## Follicle counting
The inner third of each ovary was serially sliced into 5 μm thick sections and follicles were counted every five sections and classified into four stages (primordial, primary, secondary and antral). Only those follicles in which the nucleus of the oocyte was clearly visible were counted.

## Epididymal sperm count
The epididymides were removed, minced and incubated in 1,5 ml of KSOM for 30 min at 37°C to release sperm into the medium. The suspension was incubated for 10 min at 60°C and the total sperm count was quantified by using a hemacytometer.

## Fertility assessment
$Hsf2bp^{+/+}$ and $Hsf2bp^{S167L/S167L}$ males and females (8 weeks old) were mated with WT females and males, respectively, over the course of 4–12 months. six mice per genotype (seven mice for $Hsf2bp^{S167L/S167L}$ females) were crossed. The presence of copulatory plug was examined daily and the number of pups per litter was recorded.

## Immunocytology and antibodies
Testes were detunicated and processed for spreading using a conventional 'dry-down' technique or for squashing (Gómez-H et al., 2016). Oocytes from fetal ovaries (E16.5, E17.5 and E19.5 embryos) were digested with collagenase, incubated in hypotonic buffer, disaggregated and fixed in paraformaldehyde. Rabbit polyclonal antibodies against HSF2BP and BRME1 were developed by ProteintechTM against a fusion protein of poly-His with full length HSF2BP or BRME1 (pUC57 vector) of mouse origin. Two antibodies (named R1 and R2) were generated against each protein (HSF2BP or BRME1) by immunization of two different host rabbits. Rabbit polyclonal antibody against DMC1 was developed by ProteintechTM against a DMC1 peptide (EESGFQDDEESLFQDIDLLQKHGINMADIKKLKSVGICTIKG). The primary antibodies used for immunofluorescence were rabbit αHSF2BP R2 (1:30, ProteintechTM), rabbit αBRME1 R2 (1:100, ProteintechTM), mouse αSYCP3 IgG sc-74569 (1:100, Santa Cruz), rabbit α-SYCP3 serum K921 (provided by Dr. José Luis Barbero, Centro de Investigaciones Biológicas, Spain), rabbit αSYCP1 IgG ab15090 (1:200, Abcam), rabbit anti-γH2AX (ser139) IgG #07–164 (1:500, Millipore), mouse αMLH1 51-1327GR (1:20, BD Biosciences), mouse αCDK2 (1:20; Santa Cruz Sc-6248) rabbit αRAD51 PC130 (1:50, Calbiochem), rabbit αRPA1 serum ¨Molly¨ (1:30, provided by Dr. Edyta Marcon, Medical Research University of Toronto, Canada), rat αRPA2 2208S (1:100, Cell Signaling), rabbit αDMC1 (1:500, ProteintechTM), rabbit αSPATA22 16989–1-AP (1:60, Proteintech), mouse αFlag IgG (1:100; F1804, Sigma-Aldrich).

## Image acquisition and analysis

Slides were visualized at room temperature using a microscope (Axioplan 2; Carl Zeiss, Inc) with 63 × objectives with an aperture of 1.4 (Carl Zeiss, Inc). Images were taken with a digital camera (ORCA-ER; Hamamatsu) and processed with OPENLAB 4.0.3 and Photoshop (Adobe). The slides from the different genotypes used for comparative analyses were all freshly prepared in parallel and immunofluorescence were also carried out in parallel with the same freshly prepared cocktail of antibodies. Slides were not frozen to avoid differences in the background and antigen reactivity. All the images acquired were taken with constant exposure times for comparison. Quantification of foci and fluorescence intensity were performed using Image J software. Only the axis-associated foci were counted. For colocalization analysis, the same nucleus was quantified without rotation (experiment) and after rotating 90 degrees one of the images. This condition allows to determine non-specific colocalization (random). Background was subtracted for intensity quantification. Squashed preparations were visualized with a Delta vision microscopy station. Stimulated emission depletion (STED) microscopy (SP8, Leica) was used to generate the super-resolution images. Secondary antibodies for STED imaging were conjugated to Alexa 555 and 488 (Invitrogen) and the slides were mounted in Prolong Antifade Gold without DAPI.

## Generation of plasmids

Full-length cDNAs encoding HSF2BP, BRME1 (full length and delta constructs), RPA1, BRCA2 (N, M and C constructs), PALB2, RAD51, and PSMA8 were RT-PCR amplified from murine testis RNA. The cDNAs were cloned into the EcoRV pcDNA3-2XFlag, SmaI pcDNA3-2XHA or SmaI pEGFP-C1 expression vectors under the CMV promoter. In frame cloning was verified by Sanger sequencing.

## Y2H assay and screening

Y2H assay was performed using the Matchmaker Gold Yeast Two-Hybrid System (Clontech) according to the manufacturers' instructions. Mouse *Hsf2bp* cDNA was subcloned into the vector pGBKT7 and was used as bait to screen a mouse testis Mate and Plate cDNA library (Clontech Laboratories Inc). Positive clones were initially identified on double dropout SD (synthetic dropout)/–Leu /– Trp/X-α-Gal/Aureobasidin A plates before further selection on higher stringency quadruple dropout SD /–Ade /– His /– Leu /– Trp/X-α-Gal/Aureobasidin A plates. Pray plasmids were extracted from the candidate yeast clones and transformed into *Escherichia coli*. The plasmids from two independent bacteria colonies were independently grown, extracted and Sanger sequenced.

## DNA pull-down assay

ssDNA/dsDNA pull down assays were performed using the protocol previously described by *Souquet et al., 2013*. A HPLC-purified biotinylated oligonucleotide was used for the DNA pull down assays: ss60-mer F: 5'-GAT CTG CACGACGCACACCGGACGTATCTGCTATCGCTCATG TCAACCGCTCAAGCTGC/3'BiotinTEG/ (IDT) and ss60-mer R (No biotinylated): 5'- GCAGC TTGAGCGGTTGACATGAGCGATAGCAGATACGTCCGGTGTGCGTCGTGCAGATC-3'. Double-stranded DNA annealing was carried out in 50 mM NaCl, 25 mM Tris-HCl, pH 7.5 buffer with complementary sequences at molecular equivalence by a denaturing step (5 min at 95°C) and a slow return to room temperature. DNA was immobilized onto Dynabeads M-280 Streptavidin (Dynal) following the manufacturer instructions (0.2 pmol per 1 µg of beads). Protein extracts were obtained from in vitro coupled transcription/translation systems (TNT T7 Coupled Reticulocyte Lysate Systems, Promega) according to manufacturer's protocol. 15 µl of Flag-tagged proteins from TNT assays were pre-incubated on ice for 10 min in modified DBB (DBB: 50 mM Tris HCl, 100 mM NaCl, 10% (w/v) glycerol, Complete Protease inhibitor, 1 mM 2-mercaptoethanol pH 7,4 modified with 25 mM Tris-HCl, 1 mM EDTA plus 5 mg/ml BSA). After this preincubation 500 µg Dynabeads with immobilized ss- or ds-DNA were added and incubated for 1 hr at 4°C under agitation. Then the beads were washed three times (5 min rotating at RT) in 700 µl of modified DBB without BSA, before being washed once in 700 µl of rinsing buffer (modified DBB with 150 mM NaCl). Finally, DNA-binding proteins were eluted by resuspending the beads in 30 µl of Laemmli buffer boiling the samples for 5 min. The samples were analyzed by western blot.

## Cell lines and transfections

HEK293T and U2OS cell lines were obtained from the ATCC and transfected with Jetpei (PolyPlus) according to the manufacturer protocol. Cell lines were tested for mycoplasma contamination using the Mycoplasma PCR ELISA (Sigma).

## Immunoprecipitation and western blotting

HEK293T cells were transiently transfected and whole cell extracts were prepared in a 50 mM Tris-HCl pH 7,4, 150 mM NaCl, 1 mM EDTA, 1% Triton X-100 buffer supplemented with protease inhibitors. Those extracts were cleared with protein G Sepharose beads (GE Healthcare) for 1 hr. The corresponding antibodies were incubated with the extracts for 2 hr and immunocomplexes were isolated by adsorption to protein G-Sepharose beads o/n. After washing, the proteins were eluted from the beads with 2xSDS gel-loading buffer 100 mM Tris-Hcl (pH 7), 4% SDS, 0.2% bromophenol blue, 200 mM β-mercaptoethanol and 20% glycerol, and loaded onto reducing polyacrylamide SDS gels. The proteins were detected by western blotting with the indicated antibodies. Immunoprecipitations were performed using mouse αFlag IgG (5 µg; F1804, Sigma-Aldrich), mouse αGFP IgG (4 µg; CSB-MA000051M0m, Cusabio), ChromPure mouse IgG (5 µg/1 mg prot; 015-000-003). Primary antibodies used for western blotting were rabbit αFlag IgG (1:2000; F7425 Sigma-Aldrich), goat αGFP IgG (sc-5385, Santa Cruz) (1:3000), rabbit αMyc Tag IgG (1:3000; #06–549, Millipore), rabbit αHSF2BP R2 (1:2000, ProteintechTM), rabbit αBRME1 R1 (1:3000, ProteintechTM), rat αRPA2 (1:1000, Cell Signaling (Cat 2208S)). Secondary horseradish peroxidase-conjugated α-mouse (715-035-150, Jackson ImmunoResearch), α-rabbit (711-035-152, Jackson ImmunoResearch), α-goat (705-035-147, Jackson ImmunoResearch) or α-rat (712-035-150, Jackson ImmunoResearch) antibodies were used at 1:5000 dilution. Antibodies were detected by using Immobilon Western Chemiluminescent HRP Substrate from Millipore. Secondary DyLight conjugated α-mouse (DyLight 680, 35518 Thermo-Scientific) and α-rabbit (DyLight 800, 35571 Thermo-Scientific) were used at 1:10,000 dilution. Antiboides were detected using a LI-COR Oddysey fluorescent Imager.

## Testis immunoprecipitation

Testis extracts were prepared in 50 mM Tris-HCl (pH8), 500 mM NaCl, 1 mM EDTA 1% Triton X100. 4 mg of protein were incubated with 10 µg of the specific antibody against the protein to be immunoprecipitated for 2 hr at 4°C rotating. Then 50 µl of sepharose beads (GE Healthcare) were added to the protein-Ab mixture and incubated overnight at 4°C with rotation. After that, the protein-bounded beads were washed four times with 500 µl of the extraction buffer by centrifugating 1 min at 10,000 rpm and 4°C. Finally, the co-immunoprecipitated proteins were eluted from the beads by resuspending the beads in 50 µl Laemmli buffer and boiling for 5 min. The samples were analyzed by western blot.

## Testis immunoprecipitation coupled to mass spectrometry analysis

200 µg of antibodies R1 and R2 against BRME1 (two independent IPs) and IgG from rabbit (negative control) were crosslinked to 100 ul of sepharose beads slurry (GE Healthcare). Testis extracts were prepared in 50 mM Tris-HCl (pH8), 500 mM NaCl, 1 mM EDTA 1% Triton X100. 20 mg of protein extracts were incubated o/n with the sepharose beads. Protein-bound beads were packed into columns and washed in extracting buffer for three times. Proteins were eluted in 100 mM glycine pH3 and analysed by Lc-MS/MS shotgun in LTQ Velos Orbitrap at the Proteomics facility of Centro de Investigación del Cáncer (CSIC/University of Salamanca).

## Mass spectrometry data analysis

Raw data were analysed using MaxQuant v 1.6.2.6 (*Cox and Mann, 2008*) against SwissProt Mouse database (UP000000589, Oct, 2019) and MaxQuant contaminants. All FDRs were of 1%. Variable modifications taken into account were oxidation of M and acetylation of the N-term, while fixed modifications included considered only carbamidomethylation of C. The maximum number of modifications allowed per peptide was of 5. Proteins were quantified using iBAQ (*Schwanhäusser et al., 2011*). Potential contaminants, reverse decoy sequences and proteins identified by site were removed. Proteins with less than two unique peptides in the R1 and R2 groups were not considered for ulterior analysis. Proteins with less than two unique peptides in the control group and more than

two in both groups R1 and R2 were selected as high-confidence candidates (group R1 and R2 only). An additional group of putative candidates was selected for those proteins with two or more unique peptides in one of the R1 or R2 groups and no unique peptides in the control sample (groups R1 only and R2 only, respectively).

## Statistics

In order to compare counts between genotypes, we used the Two-tailed Welch's t-test (unequal variances t-test), which was appropriate as the count data were not highly skewed (i.e., were reasonably approximated by a normal distribution) and in most cases showed unequal variance. We applied a two-sided test in all the cases. Asterisks denote statistical significance: *p-value<0.05, **p-value<0.01, ***p-value<0.001 and ****p-value<0.0001.

## Acknowledgements

We thank Dr. Barbero and Dr. Edyta Marcon, for providing antibodies against SYCP3 and RPA, respectively, Dr. Emmanuelle Martini for helpful advice with the DNA binding assay,Dr. Fabien Fauchereau for the genetic analysis of the family and Dr. Alex N Zelenskyy for helpful discussion. This study was supported by Université Paris Diderot and the Fondation pour la Recherche Médicale (Labelisation Equipes DEQ20150331757, SC, A-LT and RAV). This work was supported by MINECO (BFU2017-89408-R) and by Junta de Castilla y León (CSI239P18). NFM, FSS and LGH are supported by European Social Fund/JCyLe grants (EDU/310/2015, EDU/556/2019 and EDU/1083/2013). YBC is funded by a grant from MINECO (BS-2015–073993). The proteomic analysis was performed in the Proteomics Facility of Centro de Investigación del Cáncer, Salamanca, Grant PRB3(IPT17/0019 - ISCIII-SGEFI/ERDF). CIC-IBMCC is supported by the Programa de Apoyo a Planes Estratégicos de Investigación de Estructuras de Investigación de Excelencia cofunded by the Castilla–León autonomous government and the European Regional Development Fund (CLC–2017–01). The funders had no role in study design, data collection and analysis, decision to publish, or preparation of the manuscript.

## Additional information

### Funding

| Funder | Author |
| --- | --- |
| Ministerio de Economía y Competitividad | Alberto M Pendás |

The funders had no role in study design, data collection and interpretation, or the decision to submit the work for publication.

### Author contributions

Natalia Felipe-Medina, Resources, Formal analysis, Supervision, Investigation, Methodology, Writing - review and editing; Sandrine Caburet, Formal analysis, Investigation, Methodology, Writing - original draft, Writing - review and editing; Fernando Sánchez-Sáez, Yazmine B Condezo, Laura Gómez-H, Paloma Duque, Formal analysis, Methodology; Dirk G de Rooij, Anne Laure Todeschini, Formal analysis, Investigation; Rodrigo Garcia-Valiente, Formal analysis; Manuel Adolfo Sánchez-Martin, Resources, Methodology; Stavit A Shalev, Resources; Elena Llano, Conceptualization, Formal analysis, Investigation, Writing - original draft, Writing - review and editing; Reiner A Veitia, Conceptualization, Funding acquisition, Investigation, Writing - original draft, Writing - review and editing; Alberto M Pendás, Conceptualization, Formal analysis, Supervision, Funding acquisition, Investigation, Methodology, Writing - original draft, Writing - review and editing

### Author ORCIDs

Natalia Felipe-Medina (iD) https://orcid.org/0000-0001-6975-2524
Sandrine Caburet (iD) http://orcid.org/0000-0002-7404-8213
Rodrigo Garcia-Valiente (iD) http://orcid.org/0000-0003-0444-5587

Reiner A Veitia [iD] https://orcid.org/0000-0002-4100-2681
Alberto M Pendás [iD] https://orcid.org/0000-0001-9264-3721

## Ethics

Animal experimentation: All the experiments were approved by the Ethics Committee for Animal Experimentation of the University of Salamanca (USAL) and the Ethics committee of the Spanish Research Council (CSIC) under protocol #00-245. Accordingly, all the mouse protocols used in this work have been approved by the above mentioned Animal Experimentation committees. Specifically, mice were always housed in a temperature-controlled facility (specific pathogen free, spf) using individually ventilated cages, standard diet and a 12 h light/dark cycle, according to EU law (63/2010/UE) and the Spanish royal law (53/2013) at the "Servicio de Experimentación Animal, SEA. In addition, animal suffering was always minimized and we made every effort to improve animal welfare during the life of the animals.

## Decision letter and Author response

Decision letter https://doi.org/10.7554/eLife.56996.sa1
Author response https://doi.org/10.7554/eLife.56996.sa2

# Additional files

### Supplementary files

• Source data 1. Raw data for all figures and figure supplements.

• Supplementary file 1. Supplementary data and tables including quantification data. Supplementary file 1a. shows the whole exome sequencing and mapping metrics for the three genomic samples. Supplementary file 1b. shows the numbers of variants from the WES analysis and passing the various filters. Supplementary file 1c. shows the predictions of pathogenicity and conservations by 18 computational predictors. Supplementary files 1d and 1e. show the quantification of the colocalization between HSF2BP, BRME1, RPA1 and DMC1. Supplementary file 1f. shows the comparative alterations between all the mutants. Associated to *Figures 4*, *5*, *6* and *9*. Supplementary files 1g and 1h. show the putative BRME1 interactors identified by mass spectrometry. Supplementary files 1i and 1j. show respectively the crRNAs and the ssODN employed in the generation of the different mouse models. Supplementary file 1k. shows the primers and expected product sizes for genotyping mouse models. Supplementary file 1l. shows a summary of all the main and supplementary figures and their relationship.

• Transparent reporting form

### Data availability

All data generated or analysed during this study are included in the manuscript and supporting files.

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

# Appendix 1

## Key resources table

**Appendix 1—key resources table**

| Reagent type (species) or resource | Designation | Source or reference | Identifiers | Additional information |
|---|---|---|---|---|
| Genetic reagent (*M. musculus*) | *Hsf2bp*$^{-/-}$ | This paper | | Materials and methods section *Figure 2—figure supplement 1* Available from the authors upon request Dr. Alberto M. Pendás (amp@usal.es) |
| Genetic reagent (*M. musculus*) | *Hsf2bp*$^{S167L/S167L}$ | This paper | | Materials and methods section *Figure 2—figure supplement 1* Available from the authors upon request Dr. Alberto M. Pendás (amp@usal.es) |
| Genetic reagent (*M. musculus*) | *Brme1*$^{-/-}$ | This paper | | Materials and methods section *Figure 8—figure supplement 1* Available from the authors upon request Dr. Alberto M. Pendás (amp@usal.es) |
| Genetic reagent (*M. musculus*) | *Brme1*$^{\Delta142\text{-}472/\Delta142\text{-}472}$ | This paper | | Materials and methods section *Figure 7—figure supplement 1* Available from the authors upon request Dr. Alberto M. Pendás (amp@usal.es) |
| Genetic reagent (*M. musculus*) | *Rnf212*$^{-/-}$ | This paper | | Materials and methods section *Figure 7—figure supplement 5* Available from the authors upon request Dr. Alberto M. Pendás (amp@usal.es) |
| Genetic reagent (*M. musculus*) | *Hei10*$^{-/-}$ | This paper | | Materials and methods section *Figure 7—figure supplement 5* Available from the authors upon request Dr. Alberto M. Pendás (amp@usal.es) |
| Genetic reagent (*M. musculus*) | *Spo11*$^{-/-}$ | This paper | | Materials and methods section *Figure 7—figure supplement 6* Available from the authors upon request Dr. Alberto M. Pendás (amp@usal.es) |
| Genetic reagent (*M. musculus*) | *Psma8*$^{-/-}$ | PMID:31437213 | | |
| Genetic reagent (*M. musculus*) | *Six6os1*$^{-/-}$ | PMID:27796301 | | |
| Genetic reagent (*M. musculus*) | *Rec8*$^{-/-}$ | PMID:15515002 | | Dr. John C. Schimenti (Cornell university) |
| Cell line (*H. sapiens*) | U2OS | ATCC | HTB-96 | |

*Continued on next page*

*Appendix 1—key resources table continued*

| Reagent type (species) or resource | Designation | Source or reference | Identifiers | Additional information |
|---|---|---|---|---|
| Cell line (*H. sapiens*) | HEK293T | ATCC | CRL-11268 | |
| Recombinant DNA reagent | pEGFP-C1 | Clontech | Catalog: 6084–1 | |
| Recombinant DNA reagent | pcDNA3 | Invitrogen | A-150228 | |
| Recombinant DNA reagent | pcDNA3-2xFlag | This paper Generated from pcDNA3 | | Materials and methods section *Figure 6—figure supplement 2* Available from the authors upon request Dr. Alberto M. Pendás (amp@usal.es) |
| Recombinant DNA reagent | pEGFP-C1 HSF2BP | This paper | | Materials and methods section *Figure 7* Available from the authors upon request Dr. Alberto M. Pendás (amp@usal.es) |
| Recombinant DNA reagent | pcDNA3 2xFlag HSF2BP | This paper | | Materials and methods section *Figure 7* Available from the authors upon request Dr. Alberto M. Pendás (amp@usal.es) |
| Recombinant DNA reagent | pcDNA3 2xFlag HSF2BP-S167L | This paper | | Materials and methods section *Figure 7* Available from the authors upon request Dr. Alberto M. Pendás (amp@usal.es) |
| Recombinant DNA reagent | pcDNA3 2xHA HSF2BP | This paper | | Materials and methods section *Figure 10—figure supplement 1* Available from the authors upon request Dr. Alberto M. Pendás (amp@usal.es) |
| Recombinant DNA reagent | pEGFP-C1 BRME1 | This paper | | Materials and methods section *Figure 10—figure supplement 1* Available from the authors upon request Dr. Alberto M. Pendás (amp@usal.es) |
| Recombinant DNA reagent | pcDNA3 2xFlag BRME1 | This paper | | Materials and methods section *Figure 10—figure supplement 1* Available from the authors upon request Dr. Alberto M. Pendás (amp@usal.es) |
| Recombinant DNA reagent | pcDNA3 2xFlag BRME1Δ142–472 | This paper | | Materials and methods section *Figure 7* Available from the authors upon request Dr. Alberto M. Pendás (amp@usal.es) |

*Continued on next page*

*Appendix 1—key resources table continued*

| Reagent type (species) or resource | Designation | Source or reference | Identifiers | Additional information |
|---|---|---|---|---|
| Recombinant DNA reagent | pEGFP-C1 HSF2BP | This paper | | Materials and methods section *Figure 10* Available from the authors upon request Dr. Alberto M. Pendás (amp@usal.es) |
| Recombinant DNA reagent | pEGFP-C1 HSF2BP-S167L | This paper | | Materials and methods section *Figure 10* Available from the authors upon request Dr. Alberto M. Pendás (amp@usal.es) |
| Recombinant DNA reagent | pEGFP-C1 BRCA2-C | This paper | | Materials and methods section *Figure 6—figure supplement 2* Available from the authors upon request Dr. Alberto M. Pendás (amp@usal.es) |
| Recombinant DNA reagent | pcDNA3 2xFlag BRCA2-C | This paper | | Materials and methods section *Figure 10* Available from the authors upon request Dr. Alberto M. Pendás (amp@usal.es) |
| Recombinant DNA reagent | pcDNA3 2xFlag BRCA2-M | This paper | | Materials and methods section Available from the authors upon request Dr. Alberto M. Pendás (amp@usal.es) |
| Recombinant DNA reagent | pcDNA3 2xFlag BRCA2-N | This paper | | Materials and methods section Available from the authors upon request Dr. Alberto M. Pendás (amp@usal.es) |
| Recombinant DNA reagent | pcDNA3 2xFlag RPA1 | This paper | | Materials and methods section *Figure 10—figure supplement 1* Available from the authors upon request Dr. Alberto M. Pendás (amp@usal.es) |
| Recombinant DNA reagent | pcDNA3 2xFlag RAD51 | This paper | | Materials and methods section *Figure 10—figure supplement 1* Available from the authors upon request Dr. Alberto M. Pendás (amp@usal.es) |
| Recombinant DNA reagent | pEGFP-C1 PALB2 | This paper | | Materials and methods section *Figure 10—figure supplement 1* Available from the authors upon request Dr. Alberto M. Pendás (amp@usal.es) |
| Recombinant DNA reagent | pcDNA 2xHA PALB2 | This paper | | Materials and methods section *Figure 10—figure supplement 1* Available from the authors upon request Dr. Alberto M. Pendás (amp@usal.es) |

*Continued on next page*

*Appendix 1—key resources table continued*

| Reagent type (species) or resource | Designation | Source or reference | Identifiers | Additional information |
|---|---|---|---|---|
| Antibody | Anti-HSF2BP-R2 (rabbit polyclonal) | This paper (ProteintechTM) | | Materials and methods section IF (1:30) WB (1:2000) Available from the authors upon request Dr. Alberto M. Pendás (amp@usal.es) |
| Antibody | Anti-BRME1-R1 (rabbit polyclonal) | This paper (ProteintechTM) | | Materials and methods section WB (1:3000) Available from the authors upon request Dr. Alberto M. Pendás (amp@usal.es) |
| Antibody | Anti-BRME1-R2 (rabbit polyclonal) | This paper (ProteintechTM) | | Materials and methods section IF (1:100) Available from the authors upon request Dr. Alberto M. Pendás (amp@usal.es) |
| Antibody | Anti-DMC1 (rabbit polyclonal) | This paper (ProteintechTM) | | Materials and methods section IF (1:500) Available from the authors upon request Dr. Alberto M. Pendás (amp@usal.es) |
| Antibody | Anti-SYCP3 (mouse monoclonal) | Santa cruz | sc-74569 | IF (1:100) |
| Antibody | Anti-SYCP3 (rabbit polyclonal) | PMID:27796301 | K921 | Dr. José Luis Barbero (Centro de Investigaciones Biologicas) IF (1:60) |
| Antibody | Anti-SYCP1 (rabbit polyclonal) | Abcam | ab15090 | IF (1:200) |
| Antibody | anti-γH2AX (ser139) (rabbit polyclonal) | Millipore | #07–164 | IF (1:500) |
| Antibody | Anti-MLH1 (mouse monoclonal) | BD Biosciences | 51-1327GR | IF (1:20) |
| Antibody | Anti-CDK2 (mouse monoclonal) | Santa Cruz | sc-6248 | IF (1:20) |
| Antibody | αRAD51 (rabbit polyclonal) | Calbiochem | PC130 | IF (1:50) |
| Antibody | αRPA1 serum (rabbit polyclonal) | | ¨Molly¨ | Dr. Edyta Marcon (Medical Research University of Toronto) IF (1:30) |
| Antibody | αRPA2 (rat monoclonal) | Cell Signalling | 2208S | IF (1:100) WB (1:1000) |
| Antibody | Anti-SPATA22 (rabbit polyclonal) | Proteintech Europe | 16989–1-AP | IF (1:60) |
| Antibody | Anti-Flag (mouse monoclonal) | Sigma-Aldrich | F1804 | IF (1:100) IP (5 µg) |
| Antibody | Anti-GFP (mouse monoclonal) | Cusabio | CSB-MA000051M0m | IP (5 µg) |
| Antibody | Anti-HA (mouse monoclonal) | BioLegend | MMS-101P | IP (5 µg) |

*Continued on next page*

*Appendix 1—key resources table continued*

| Reagent type (species) or resource | Designation | Source or reference | Identifiers | Additional information |
|---|---|---|---|---|
| Antibody | Mouse IgGs (mouse polyclonal) | Jackson Immunoresearch | 015-000-003 | IP (5 μg) |
| Antibody | Anti-Flag (rabbit polyclonal) | Sigma-Aldrich | F7425 | WB (1:2000) |
| Antibody | Anti-GFP (goat polyclonal) | Santa Cruz | sc-5385 | WB (1:3000) |
| Antibody | Anti-GFP (rabbit polyclonal) | Life technologies | A-11122 | WB (1:3000) |
| Antibody | Anti-HA (rabbit polyclonal) | Sigma-Aldrich | H6908 | WB (1:3000) |
| Antibody | Goat α-mouse Alexa555 (goat polyclonal) | ThermoFisher | A-32727 | IF (1:200) |
| Antibody | Goat α-mouse Alexa488 (goat polyclonal) | ThermoFisher | A-11001 | IF (1:200) |
| Antibody | Donkey α-rabbit Alexa555 (donkey polyclonal) | ThermoFisher | A-31572 | IF (1:200) |
| Antibody | Goat α-rabbit Alexa488 (goat polyclonal) | ThermoFisher | A-32731 | IF (1:200) |
| Antibody | Goat α-rabbit Alexa488 Fab (goat polyclonal) | Jackson Immunoresearch | 111-547-003 | IF (1:100) |
| Antibody | Goat α-mouse AMCA (goat polyclonal) | Jackson Immunoresearch | 115-155-146 | IF (1:100) |
| Antibody | Donkey α-rabbit AMCA (donkey polyclonal) | Jackson Immunoresearch | 711-155-152 | IF (1:100) |
| Antibody | Goat α-rat Alexa488 (goat polyclonal) | ThermoFisher | A-11006 | IF (1:200) |
| Antibody | Secondary horseradish peroxidase-conjugated α-mouse (donkey polyclonal) | Jackson Immunoresearch | 715-035-150 | WB (1:5000) |
| Antibody | Secondary horseradish peroxidase-conjugated α-rabbit (donkey polyclonal) | Jackson Immunoresearch | 711-035-152 | WB (1:5000) |
| Antibody | Secondary horseradish peroxidase-conjugated α-goat (donkey polyclonal) | Jackson Immunoresearch | 705-035-147 | WB (1:5000) |
| Antibody | Secondary horseradish peroxidase-conjugated α-rat (donkey polyclonal) | Jackson Immunoresearch | 712-035-150 | WB (1:5000) |

*Continued on next page*

*Appendix 1—key resources table continued*

| Reagent type (species) or resource | Designation | Source or reference | Identifiers | Additional information |
|---|---|---|---|---|
| Antibody | Secondary DyLightTM 680 conjugated α-mouse (goat polyclonal) | Thermo Scientific | 35518 | WB (1:10000) |
| Antibody | Secondary DyLightTM 800 conjugated α-rabbit (goat polyclonal) | Thermo Scientific | 35571 | WB (1:10000) |
| Sequence-based reagent | sgRNA1 *Hsf2bp* | This paper (IDT) | CRISPR-Cas9 crRNA | Materials and methods section ***Supplementary file 1i*** ***Figure 2—figure supplement 1*** 5'-TCACAAAACTCTCCATCGTC-3' |
| Sequence-based reagent | sgRNA2 *Hsf2bp* | This paper (IDT) | CRISPR-Cas9 crRNA | Materials and methods section ***Supplementary file 1i*** ***Figure 2—figure supplement 1*** 5'-ATTGGATGGGGATGTCAAGG-3' |
| Sequence-based reagent | sgRNA3 *Brme1* | This paper (IDT) | CRISPR-Cas9 crRNA | Materials and methods section ***Supplementary file 1i*** ***Figure 7—figure supplement 1*** 5'-AACCTCAGGGACTCTCTCTG-3' |
| Sequence-based reagent | sgRNA4 *Brme1* | This paper (IDT) | CRISPR-Cas9 crRNA | Materials and methods section ***Supplementary file 1i*** ***Figure 7—figure supplement 1*** 5'-GAAGTCTAGTTCCATTGCTG-3' |
| Sequence-based reagent | sgRNA5 *Spo11* | This paper (IDT) | CRISPR-Cas9 crRNA | Materials and methods section ***Supplementary file 1i*** ***Figure 7—figure supplement 6*** 5'-TATGTCTCTATGCAGATGCA-3' |
| Sequence-based reagent | sgRNA6 *Spo11* | This paper (IDT) | CRISPR-Cas9 crRNA | Materials and methods section ***Supplementary file 1i*** ***Figure 7—figure supplement 6*** 5'- ACACTGACAGCCAGCTCTTT-3' |
| Sequence-based reagent | sgRNA7 *Rnf212* | This paper (IDT) | CRISPR-Cas9 crRNA | Materials and methods section ***Supplementary file 1i*** ***Figure 7—figure supplement 5*** 5'- ACCCACGTGAGACTCGCGCG-3' |
| Sequence-based reagent | sgRNA8 *Rnf212* | This paper (IDT) | CRISPR-Cas9 crRNA | Materials and methods section ***Supplementary file 1i*** ***Figure 7—figure supplement 5*** 5'- CCTCAAAGGTCCGCGTATTC-3' |
| Sequence-based reagent | sgRNA9 *Hei10* | This paper (IDT) | CRISPR-Cas9 crRNA | Materials and methods section ***Supplementary file 1i*** ***Figure 7—figure supplement 5*** 5'- GAAAGGGTACTGTTGCAAGC-3' |

*Appendix 1—key resources table continued*

| Reagent type (species) or resource | Designation | Source or reference | Identifiers | Additional information |
|---|---|---|---|---|
| Sequence-based reagent | ssODN $Hsf2bp^{S167L/S167L}$ | This paper (IDT) | | Materials and methods section *Supplementary file 1j* *Figure 2—figure supplement 1* 5'CTTTGGAAAGATGTGACAG TTCTATCTTTTTTATCTTTCA GGACAAAGCATTGAAGTTTT TCAACATAACTGGACAGACGA TGGAGAGTTTTGTGAAGTTA TTGGATGGGGATGTCAAGGAAG TTGATTCTGATGAAAATCAATTTGTC TTTGCACTGGCTGGAATTGTAAC AAGTAGGTAACTTTT CAGATACAGCGCT3' |
| Sequence-based reagent | ssODN $Brme1^{Δ142-472/Δ142-472}$ | This paper (IDT) | | Materials and methods section *Supplementary file 1j* *Figure 7—figure supplement 1* 5'CTTCAGAGTGCTTGCTTAT TGAAGGCCAGGACTGAATCT TCTTTTTCCACAGGAAACAA GGCCAGAGCTGGGAGCCCTC AAAGCAGCCAGCCAGCCACA GGCAATGGAACTAGACTTCC TGCCTGACAGCCAGATACAG GATGCCCTGGATGCCACTAA CATGGAGCAGGTAAGAGCT TTCTGTACTCAAATGTACACCC3' |
| Sequence-based reagent | ssODN $Spo11^{-/-}$ | This paper (IDT) | | Materials and methods section *Supplementary file 1j* *Figure 7—figure supplement 6* 5'GTTTCCTGCGGTATGTGT TCTCTGCCGTGGTCTGTGTT TGTCACCGTCCAGGAGCAA TGCTCATTCTGTGTTGAGCT TGCATCTGCATAGAGACAT ATTCTTCACTGACAGCCAGCT CTTTGGCAACCAGGCTGCG GTGGACAGCGCCATCGATG ACATTTCCTGTATGCTGA AAGTGCCCAGGAGGAG TCTGCACGTGG-3' |
| Sequence-based reagent | HSF2BP-F1 | This paper | PCR primer | Materials and methods section *Supplementary file 1j* *Figure 2—figure supplement 1* 5'-TTCTTTGGAAAGATGTGACAGTTC-3' |
| Sequence-based reagent | HSF2BP-R1 | This paper | PCR primer | Materials and methods section *Supplementary file 1j* *Figure 2—figure supplement 1* 5'-ACCTGGGTTTCCTTTAGATCAGTTA-3' |
| Sequence-based reagent | BRME1- F2 | This paper | PCR primer | Materials and methods section *Supplementary file 1j* *Figure 7—figure supplement 1* 5'-GAAAGTTCTTCAGAGTGCTTGCT-3' |

*Continued on next page*

*Appendix 1—key resources table continued*

| Reagent type (species) or resource | Designation | Source or reference | Identifiers | Additional information |
|---|---|---|---|---|
| Sequence-based reagent | BRME1- R2 | This paper | PCR primer | Materials and methods section *Supplementary file 1j* *Figure 7—figure supplement 1* 5'-AGCCCTATCTTGTCACCTAAAG-3' |
| Sequence-based reagent | BRME1- F3 | This paper | PCR primer | Materials and methods section *Supplementary file 1j* *Figure 7—figure supplement 1* 5'-CCCAGCAGATGCCTCTCTTAT-3' |
| Sequence-based reagent | BRME1- R3 | This paper | PCR primer | Materials and methods section *Supplementary file 1j* *Figure 7—figure supplement 1* 5'-CTCAGCAGAGTTCCAATGCAG-3' |
| Sequence-based reagent | SPO11-F4 | This paper | PCR primer | Materials and methods section *Supplementary file 1j* *Figure 7—figure supplement 6* 5'- AGAGCCCCCAGTGCTCTTAAC-3' |
| Sequence-based reagent | SPO11-R4 | This paper | PCR primer | Materials and methods section *Supplementary file 1j* *Figure 7—figure supplement 6* 5'- GGCAGACCCCTCTACCTCTGT-3' |
| Sequence-based reagent | RNF212-F5 | This paper | PCR primer | Materials and methods section *Supplementary file 1j* *Figure 7—figure supplement 5* 5'- TTTCTTTGCCTCCGTACTTTTGG-3' |
| Sequence-based reagent | RNF212-R5 | This paper | PCR primer | Materials and methods section *Supplementary file 1j* *Figure 7—figure supplement 5* 5'- CCCAGGCTTTACTTCAACAACAA−3' |
| Sequence-based reagent | HEI10-F6 | This paper | PCR primer | Materials and methods section *Supplementary file 1j* *Figure 7—figure supplement 5* 5'- CTGCCTGTTCTCACATCTTC-3' |
| Sequence-based reagent | HEI10-R6 | This paper | PCR primer | Materials and methods section *Supplementary file 1j* *Figure 7—figure supplement 5* 5'- AGCTTTCCAGAAAGGGTACTG−3' |
| Sequence-based reagent | ss60-mer F | PMID:24068956 | DNA Binding assay primer | Materials and methods section *Figure 7—figure supplement 4* 5'-GAT CTG CACGACGC ACACCGGACGTATCTGCTATC GCTCATGTCAACCGCT CAAGCTGC/3'BiotinTEG/ |
| Sequence-based reagent | ss60-mer R | PMID:24068956 | DNA Binding assay primer | Materials and methods section *Figure 7—figure supplement 4* 5'- GCAGCTTGAGCGGTTGACAT GAGCGATAGCAGATACGTCCG GTGTGCGTCGTGCAGATC-3' |

*Continued on next page*

*Appendix 1—key resources table continued*

| Reagent type (species) or resource | Designation | Source or reference | Identifiers | Additional information |
|---|---|---|---|---|
| Sequence-based reagent | HSF2BP-EX6F | This Paper | Sanger sequencing primer | Material and methods section *Figure 1—figure supplement 1* 5'-CTAGAATCTTCTGTATCCTGCA-3' |
| Sequence-based reagent | HSF2BP-EX6R2 | This Paper | Sanger sequencing primer | Material and methods section *Figure 1—figure supplement 1* 5'-GGTCTGGAAGCAAACAGGCAA-3' |
| Commercial assay or kit | TNT T7 Coupled Reticulocyte Lysate Systems | Promega | L4610 | *Figure 7—figure supplement 4* |
| Commercial assay or kit | In Situ Cell Death Detection Kit | Roche | 11684795910 | *Figure 2* |
| Commercial assay or kit | Matchmaker Gold Yeast Two-Hybrid System | Clontech | 630489 | Materials and methods section |
| Commercial assay or kit | Mouse testis Mate and Plate cDNA library | Clontech | 638852 | Materials and methods section |
| Commercial assay or kit | Jetpei | PolyPlus | 101–40N | Materials and methods section |
| Commercial assay or kit | GammaBind G Sepharose | GE Healthcare | 17-0885-02 | Materials and methods section |
| Commercial assay or kit | Dynabeads M-280 Streptavidin | Thermo Fisher | 11205D | Materials and methods section *Figure 7—figure supplement 4* |
| Chemical compound, drug | MG132 | Sigma-Aldrich PMDI:28059716 | M8699 | *Figure 10* |
| Other | DAPI stain | Invitrogen | D1306 | Materials and methods section |
| Other | Vectashield Mounting Medium | Vector Laboratories | H1000 | Materials and methods section |
| Other | ProLong Gold antifade reagent | Invitrogen | P10144 | Materials and methods section |

