## [Decision Letter]

Thank you for submitting your article "A missense in HSF2BP causing Primary Ovarian Insufficiency affects meiotic recombination by its novel interactor MIDAP" for consideration by *eLife*. Your article has been reviewed by three peer reviewers, including Bernard de Massy as the Reviewing Editor and Reviewer #1, and the evaluation has been overseen by Kevin Struhl as the Senior Editor. The following individuals involved in review of your submission have agreed to reveal their identity: Maria Jasin (Reviewer #2).

The reviewers have discussed the reviews with one another and the Reviewing Editor has drafted this decision to help you prepare a revised submission.

Summary:

This paper presents a primary ovarian insufficiency (POI) case study, proband and affected family members with a unique genetic variant in HSF2BP. To examine the functional consequences of this variant, they generated a mouse mutant with the equivalent allele (S167L). Despite ostensibly normal numbers of follicles, Hsf2bp- S167L female mice showed reduced fertility in females. The authors also identify an interaction partner of HSF2BP named MIDAP and provide evidence that these two proteins function interdependently to facilitate BRCA2-mediated assembly of RAD51/DMC1 strand-exchange complexes.

Essential revisions:

Overall, while there are a lot of data in this paper, the quality of some experiments is questionable and lack rigorous quantification. The text is often inaccurate, sometimes not correct and requires major clarification and improvement.

Below are mentioned essential points and important point to revise, that include new data, formatting figures, text modification and reorganization.

1) One essential issue is to validate the conclusion that the case of human POI is caused by a variant of HSF2BP. Thus, the mouse mutant Hsf2bp167L requires additional analysis.

2) Another major problem is order and clarity of data presentation.

a) When presenting the phenotype of the Hsf2bp point mutant, the authors should clarify first the already published phenotype of the null mutant. They can add the data of their own null mutant as comparison.

b) Order of data:

Males and female data are not presented in a consistent way for the analysis of Hsf2bp mutant (Figure 3, Figure 4 and Figure 5): Best would be to present male and female in parallel: Figure 3 (RPA), Figure 4 (DMC1, RAD51), Figure 5 (Mlh1). RAD51 data for female is required.

MIDAP localization in Hsf2bp mutants should be presented along with its colocalization with HSF2BP in wt.

Analysis of SPATA22 should be moved to main figures.

3) Removing poorly informative data

The in vitro assay fails to detect DNA binding activity of MIDAP or HSF2BP. This negative result obtained with non-purified proteins of undetermined concentration and molar ratio with the substrate do not allow to conclude on the DNA binding property of those proteins.

Only part of the co-transfection analysis (Figure 10) should be maintained (see details below).

4) The localization of HSF2BP/MIDAP at DSB repair foci is not shown in this study. This should be clarified either by referring to the previous study or by doing colocalization analysis (with RPA or DMC1 or RAD51).

5) Clarifying interpretations.

Many issues in the text are reported below.

Importantly the protein levels of HSF2BP and MIDAP is the mutants (Midap and Hsfp2bp) should be tested (in extracts from juvenile to avid confounding effect of cellular composition). Any coIP and western blot analysis from mouse testis to validate the interactions? and also the one with BRCA2? This would be expected given the Mass/spec data.

Figure 1

Fertility and rigorous follicle (at different stages) counting for Hsf2bp-S167L mice at different ages. One prediction is that oocyte reserve should decreased at least at advanced ages.

Mutants with decreased ovarian reserves generally activate normal numbers of follicles and thus can look normal, but pools will deplete faster. Oocyte counts using immunohistochemistry of ovaries from mice of different ages with an appropriate marker (e.g. MVH) is required.

1(D) – please provide quantification of testis weights corrected for animal body weight (mean {plus minus} standard deviation).

1(F) – Epididymis histology for Hsf2bp-S167L requires quantification. An easier approach would be to present sperm counts using standard methods. Scale bars in Figure 1B and 1C should be the same size to facilitate comparison. Also, it is difficult to see follicles, especially in (B). Immunostaining should be performed to identify follicles.

1(H). Spreads are not appropriate to quantify apoptotic cells. How was apoptosis monitored? The legend is incorrect: "(H).…spread preparations from Hsf2bp-/- and Hsf2bpS167L/S167L mice." It should be +/+ and S167L/S167L.

Figure 1 legend is much too long.

Figure 2

2 (A) – HSF2BP staining intensity -- were graphs showing HSF2BP staining intensity normalized against Sycp3 intensity? Were exposure times constant between images and samples? Please provide more details regarding image acquisition. This comment applies to other figures as well.

There is no information about image analysis in the Materials and methods section? How were foci counted? Was background subtracted? Which software? Which parameters?

2 (A,B). Which test was used, meaning of stars?

2 (C). A non-specific band of unknown origin cannot be used as loading control. Please add a proper loading control. The extracts should be from juvenile to interpret the result since wt and mutant do not have the same testicular cell composition.

What is the mouse background for the -/- and S167L/S167L mice? Are they the same? Differences in strain background (especially mixed versus inbred) can affect the penetrance of meiotic mutants.

Figure 3

H2AX plots Y axis: add γ

Figure 4

(A) – right hand side graph is missing a y-axis label. Labels should be more explicit, e.g. "number of DMC1 foci".

(B) In the right graph there is extraneous text ("S167L") under the data points.

Figure 5

(a) The evidence for XY univalent is weak. One may indeed consider the long univalent axis as X, but the smaller one could be any chromosome. This is only a deduction (On Figure 1G, impossible to tell). This observation needs further analysis: is the recombination defect (CO defect and univalent) mostly impacting the X and Y? Please, analyze the distributions of MLH1 foci, specifically the fraction of pachytene SCs that lack MLH1 foci.

(B,C) – Please comment on the observation that RPA counts are similar between control and mutant, while DMC1 counts are significantly lower. RAD51 counts should be analyzed in S167L females.

SPATA22 which appears to give a strong phenotype in MIDAP mutant should be tested in male and female Hsf2bp mutants.

Identification and analysis of MIDAP

The acronym MIDAP is not the best choice: It's unclear how MIDAP is an acronym for "meiotic double- stranded break BRCA2/HSF2BP complex associated protein". Where is the "I" from, and how is BRCA2/HSF2BP part of the acronym? What is the expression profile of MIDAP? And if the gene is not meiotic-specific it should not be labeled as such. Moreover, it's not clear that its major function is through BRCA2. Please indicate that C19ORF57 is the name of the human not the mouse gene (thus in the yeast two hybrid screen the gene identified was not C19ORF57; was the cDNA overlapping with the domain identified as interacting by CoIP?).

One learns very little from the study of the MIDAP internal deletion. Thus, this could be removed or added as supplementary information. In addition, this mutant allele does not show that the interaction domain is sufficient as the mutant protein is much larger than the interaction domain.

Midap∆/∆: we would suggest adding a number such as ∆6. Otherwise, it can be easily thought that this allele is a full deletion.

Figure 6:

(A): Add MW markers (and check this for all western blots)

(C) – IP with Δ475-600 – please provide evidence that IP worked – western blot for protein used for setting up IP reaction must be also provided in all cases. Which fraction of input, of IP was loaded? (these comments apply to all CoIPs).

(D) – The authors cannot conclude on a perfect colocalization without quantification. Please quantify. Squares indicating areas being magnified are not being reproduced properly. Also, these areas are not magnified very much- please make them larger! The insets are shifted, this should be mentioned.

Figure 7. Depletion of follicle pool in Midap-/- females should be quantified.

Males: provide evidence for the stage IV arrest; in the text explain what is stage IV for non specialist. Asterisks are barely visible, change color.

(E): 5 nuclei are shown. The four upper nuclei show various levels of synapsis defects. It is not clear what information is provided with the nucleus in the lower part which does not seem to be a pachytene but more likely a diplotene nucleus. Unless a specific conclusion is reached from this nucleus, it could be removed.

Figure 8:

(A) – Absence of HSF2BP staining in the Midap-/- mutant could be due to protein destabilization. See comment above; legend, Quantification is HSF2BP foci not MIDAP.

Figure 9. The absence of MIDAP foci should also be quantified in the Hsf2bp null mutant even if the number of foci is close to 0.

Figure 10:

(A) – and in general and for all western blots, the labeling of IPs is confusing and

ambiguous. It is unclear what the arrows are supposed to highlight. Best is to indicate which antibody was used to probe the blots, and to add arrows to identify proteins when required.

The comparison of Hsf2bp wt and S167L does not make sense in such overexpression system since the property of S167L reported above was a potential lower stability and/or level. Transfection with Hsf2bpS167L should be removed.

Overall, the information gained is the CoIP between the three proteins, the nuclear localization and stabilization of Hsf2Bp in the presence of MIDAP. Note that there is no evidence for a multimeric complex from this study. This would require performing sequential IP.

The one and half page paragraph should be reduced to half a page, which would improve the quality of the paper.

The legend should be reduced.

Figure 1: Please include the genotypes for the sequenced individuals. From the text, it appears as if the brother is heterozygous, so there does not appear to be a male homozygote to comment on human male fertility. This should be clearly stated early in the manuscript (see comment below).

Figure 2—figure supplement 1(A) legend: "Thin (non-coding) and thick (coding sequences) lines under exons represent the expected transcript derived from wild-type and Hsf2bp edited allele." There are no such lines in panel (A).

Figure 7—figure supplement 6 – What stage are these WT spermatocytes and approximate stage-matched mutants? Late zygonema?

Figure 8—figure supplement 1C. Please show the whole membrane. Any truncated protein detected? Where are the epitopes of the antibody?

Interpretation and discussion of the Hsf2bpS167L mutant

Subsection “Mice with the HSF2BP S167L variant show a partial reduction of fertility” "male and female mice were able to reproduce but females showed a significant reduction in the number and size of litters (Figure 1A) whilst males only showed a slight non-significant reduction in fertility (Figure 1A).This is in agreement with the fertility defects observed in Hsf2bp-/- mice (Figure 1C,E-F and (Brandsma et al., 2019; Zhang et al.,2019)), and suggests that the S167L variant might impact more specifically female fertility."

There are several problems with this statement:

a) This is not in agreement with the two published papers, but rather the opposite, since the published papers describe a more pronounced defect in males. Thus, this text needs to be changed.

b) However, importantly also, the authors appear to have a more pronounced phenotype in their female knockouts compared to the published reports. This needs to be discussed, included how the knockout alleles vary for the 3 knockouts that have now been generated.

c) Males with S167L have more variability in litter frequency and size, and so with the small numbers it is difficult to draw a conclusion about fertility. Moreover, testis weights and histology of tubules, which show a clear male defect.

Related to this in Subsection “Mice with the HSF2BP S167L variant show a partial reduction of fertility”: "These results suggest that mice bearing the POI-causing variant only partially phenocopy the human disease and fits well with the exacerbated sexual phenotypic dimorphism observed in the Hsf2bp-/- mice which might be less pronounced or absent in humans."

As above, no conclusion can be made about human male fertility. The lack of follicle counts/breeding over time in females makes it difficult to conclude about POI in mice. Moreover, what sexual dimorphism in -/- mice is this referring to? Both males and females are completely devoid of gametes in the HSF2BP-null.

Subsection “Mice with the HSF2BP S167L variant show a partial reduction of fertility”: "Overall, male and female Hsf2bpS167L/S167L mice show similar molecular alterations although with different severity".

There are obvious differences in molecular alterations between males and females:

- There is a dramatic increase in RPA foci in males during pachynema while no difference in RPA foci in females during pachynema.

- DMC1 is virtually unchanged in males (perhaps slight reduction in EZ), but greatly reduced in females.

- MLH1 is reduced 20-30% in males, but only ~5% (and not significant) in females.

- γ-H2AX is greatly increased during pachynema in males, but these data do not exist for females.

- RAD51 is decreased through zygonema to early pachynema with the biggest decrease seen in late zygonema, but these data don't exist for females.

- The only similarity is with DMC1, but even this is misleading as the defect in males is quite mild, but quite severe in females.

Supplementary file 1G, Supplementary file 1H

What are the two antibodies (Ab1 and Ab2) used for mass spec? What is the control sample (IP with IgG?). The Materials and methods section indicates the IP was performed with two antibodies R1 and R2 (against HSF2BP and MIDAP)? This is very confusing.

Please can you specify how many peptides of MIDAP identified in this screen.

With the large number of figures and different mutants examined, a summary table should be provided.

Comments on the Discussion section

The discussion is unfocused with no clear link back to the POI of the human family. Also, the data are generally over-interpreted into specific molecular mechanisms. This should be toned down and the authors should clarify when they are speculating.

Where is the evidence for a supramolecular complex? In the absence of validation this should also be removed from the abstract ("the higher order macromolecular structure").

Where is the evidence for direct interaction between HSF2BP and BRCA2 and MIDAP ?

Where is the evidence that the supramolecular (?) complex participate to the binding of RPA?

The known roles of DSS1 should be introduced noting that meiotic functions are not characterized.

"strongly suggesting a parallelism in the early recombination pathway where HSF2BP/MIDAP work" – this is unclear. Please rephrase.

"In general, meiotic recombination mutants seem to proceed further in female than in male meiosis perhaps because of the presence of less stringent checkpoints (Hunt and Hassold, 2002)."

But also, recombination mutants often show more severity in males due to X-Y asynapsis.

"importantly also of HSF2BP" – please rephrase.

---

## [Author Response]

Essential revisions:Overall, while there are a lot of data in this paper, the quality of some experiments is questionable and lack rigorous quantification. The text is often inaccurate, sometimes not correct and requires major clarification and improvement.Below are mentioned essential points and important point to revise, that include new data, formatting figures, text modification and reorganization.1) One essential issue is to validate the conclusion that the case of human POI is caused by a variant of HSF2BP. Thus, the mouse mutant Hsf2bp167L requires additional analysis.

We thank the referees for the comment. We have carried out additional analyses of the human mutation in both male and female mice including a comparative sperm count, TUNEL analysis of histological sections of mutant testis, follicle counts of mutant ovaries (5 dpp, 6 weeks, and 13 months), analysis of the distribution of COs along the autosomes and XY bivalents, analysis of the COs in a higher number of female mice and cells and a supplementary analysis of γH2AX as well as RAD51 and SPATA22 foci in mutant oocytes. Taking together, our new analysis corroborates and validates that the introduction of the point mutation in the *Hsf2bp* mouse gene found in the POI family leads to meiotic defects in the mouse germline.

2) Another major problem is order and clarity of data presentation.a) When presenting the phenotype of the Hsf2bp point mutant, the authors should clarify first the already published phenotype of the null mutant. They can add the data of their own null mutant as comparison.

We have included in the revised version of the MS the phenotype of the published null mutant at the beginning of the Results as suggested by the reviewers. We have also incorporated the data of our null mutant for a straight comparison. The added paragraph is:

“During the course of this work, two independent groups showed that HSF2BP is essential for meiotic recombination through its ability to interact with the armadillo repeats of BRCA2 (Zhang et al., 2019; Brandsma et al., 2019). Both groups showed that genetic disruption of *Hsf2bp* in mouse leads to the accumulation in the chromosomes axes of DNA repair proteins such as γH2AX (ATR-dependent phosphorylation of H2AX marks DSBs) and the single stranded-DNA binding protein RPA, a strong reduction of the recombinases DMC1 and RAD51 at the RNs and a lack of COs as labelled by MLH1 (Baker et al., 1996). The end result is male sterility (Brandsma et al., 2019; Zhang et al., 2019). However, loss of HSF2BP in female mice showed a milder meiotic phenotype (Zhang et al., 2019 and our own data, see below) and a weak albeit non statistical significant reduction of fertility (Brandsma et al., 2019) despite all of the mutants are nulls though in different genetic backgrounds.”

b) Order of data:Males and female data are not presented in a consistent way for the analysis of Hsf2bp mutant (Figure 3, Figure 4 and Figure 5): Best would be to present male and female in parallel: Figure 3 (RPA), Figure 4 (DMC1, RAD51), Figure 5 (Mlh1). RAD51 data for female is required.MIDAP localization in Hsf2bp mutants should be presented along with its colocalization with HSF2BP in wt.

We thank the referees for the comment. We have followed their advice and we have reorganized the data accordingly to improve the manuscript. Thus, we have presented in this revised version male and female in parallel corresponding to the reorganized Figure 4, Figure 5 and Figure 6.

We have also incorporated new data involving the analysis of RAD51 foci in *Hsf2bp*^-/-^ and *Hsf2bp*^S167L/S167L^ females and their corresponding quantification and comparison with the wild type (Figure 5D and Figure 5—figure supplement 1D).

Following the reviewers´ comments, we have placed MIDAP localization in *Hsf2bp* mutants in Figure 7 (7F-G and extended panel in Figure 7—figure supplement 4A) along with the colocalization of HSF2BP and MIDAP (Figure 7D-E).

Analysis of SPATA22 should be moved to main figures.

Following this suggestion, we have moved the SPATA22 analysis to the main figures for *Hsf2bp* mutants (Figure 6A) and also moved the SPATA22 analysis in *Midap* mutants to the main Figure 9E-F.

3) Removing poorly informative dataThe in vitro assay fails to detect DNA binding activity of MIDAP or HSF2BP. This negative result obtained with non-purified proteins of undetermined concentration and molar ratio with the substrate do not allow to conclude on the DNA binding property of those proteins.

We understand this comment. However, our experiments have been performed with purified protein extracts obtained from in vitro transcription and translation (TNT) and the protein concentration on each condition was estimated by loading the same amount of protein (estimated from the western blot signal) to avoid substantial differences on the molar ratio with the substrate. We have included the well-known DNA-binding activity of RPA as a positive control. This assay based on TNT produced proteins has been validated previously in several publications (Souquet et al., 2013 ; Loregian et al., 2004). We thus consider that these results point towards HSF2BP and MIDAP lack DNA binding activity.

Only part of the co-transfection analysis (Fig10) should be maintained (see details below)

We have carried out the modification (see below).

4) The localization of HSF2BP/MIDAP at DSB repair foci is not shown in this study. This should be clarified either by referring to the previous study or by doing colocalization analysis (with RPA or DMC1 or RAD51).

We agree with this commentary. We have carried out a new co-localization experiment by IF analysis of HSF2BP and MIDAP with RPA and DMC1 as well as the corresponding quantification. The new results have been included in a new Figure 7—figure supplement 3A-B and Supplementary files 1D and 1E.

5) Clarifying interpretations.Many issues in the text are reported below.Importantly the protein levels of HSF2BP and MIDAP is the mutants (Midap and Hsfp2bp) should be tested (in extracts from juvenile to avid confounding effect of cellular composition). Any coIP and western blot analysis from mouse testis to validate the interactions? and also the one with BRCA2? This would be expected given the Mass/spec data.

Following this criticism, we have analysed the protein levels of HSF2BP and MIDAP in juvenile mutants as suggested by the editor/referees and replaced the former WBs by the new ones (Figure 3C, Figure 7H and Figure 8H). The new results have been included in the MS: “Western blot analysis of WT, *Hsf2bp*^S167L/S167L^ and *Hsf2bp^-/-^* in whole testis extracts from 13 days post-partum (dpp) animals (Figure 3C) revealed that the reduced labelling observed by immunofluorescence correlated with a reduced protein expression level, suggesting that the mutation leads to a reduced expression and/or stability.”

We have analysed the protein levels of MIDAP in the *Hsf2bp* mutants (S167L variant and knock-out) revealing a drastic reduction of MIDAP expression in *Hsf2bp*^S167L/S167L^ and *Hsf2bp*^-/-^spermatocytes. The new results have been included in the MS: “Western blot analysis also revealed a drastic reduction of MIDAP expression in *Hsf2bp*^S167L/S167L^ and *Hsf2bp*^-/-^spermatocytes, suggesting an HSF2BP-dependent stabilization of MIDAP (Figure 7H).”

We also have analysed the protein level of HSF2BP in MIDAP mutants by Western blot analysis of 13 dpp testis extracts. The new results have been included in the manuscript: “Western blot analysis of HSF2BP in 13 dpp testis extracts from *Midap^-/-^* mice showed a strong reduction in comparison with the WT control (Figure 8H), suggesting again that MIDAP is necessary for HSF2BP protein stabilization.”

Additionally, we have carried out Co-IP experiments from mouse testis between HSF2BP and MIDAP to demonstrate the interaction in vivo. The results show positive co-IP and have been included in the MS: “We further validated this interaction in vivo by co-immunoprecipitation (co-IP) of both proteins from mouse whole testis extracts (Figure 7B).”

In the case of BRCA2, we would like to point out that we have not been able to detect endogenous BRCA2 with any antibody neither by immunofluorescence (this is a universal problem) nor by WB. Thus, in our hands, such an experiment is not possible.

Figure 1Fertility and rigorous follicle (at different stages) counting for Hsf2bp-S167L mice at different ages. One prediction is that oocyte reserve should decreased at least at advanced ages.Mutants with decreased ovarian reserves generally activate normal numbers of follicles and thus can look normal, but pools will deplete faster. Oocyte counts using immunohistochemistry of ovaries from mice of different ages with an appropriate marker (e.g. MVH) is required.

We thank the referees for the comment and accordingly we have included oocyte counts at 5 dpp, 6 weeks and 13 months from *Hsf2bp*^S167L/S167L^ mice. The new results have been included in the MS. “Histological analysis of *Hsf2bp*^S167L/S167L^ ovaries revealed no apparent differences in the number of follicles in comparison to wild-type (WT) animals (Figure 2B-C and Figure 2—figure supplement 2A), in contrast with the drastic reduction of the follicle pool in *Hsf2bp*^-/-^ ovaries (Figure 2B-C).” However, we consider that staining with a marker is not necessary since follicles and oocytes can be clearly identified just with a Hematoxylin-eosin staining of the histological sections as shown in Figure 2B and Figure 8A.

Alternative to the hypothesis of the erosion of the follicle pool, subfertility can be also caused by a defect in the quality of the secondary oocytes (i.e. absence of at least one COs per bivalent). Indeed, we have increased the number of cells and female mice analysed for interstitial CDK2 foci (as a measure of COs) and we have found a statistically significant reduction in the total number of COs and also an increased number of cells showing bivalents without CO. The new results have been included in the manuscript: “In agreement with the lower presence of recombinases, the number of COs (measured as interstitial CDK2 foci) was also reduced in *Hsf2bp*^S167L/S167L^oocytes, and a stronger reduction was observed in *Hsf2bp^-/-^*oocytes (Figure 6C and Figure 6—figure supplement 1D)”. These differences could explain the subfertility of our *Hsf2bp*^S167L/S167L^ females. We used interstitial CDK2 staining instead of MLH1 due to our anti-MLH1 antibody was finished and we could not get a new one due to the COVID-19 lockdown, so we decided to do this analysis using CDK2 for female analysis which is well accepted to be equivalent.

1(D) – please provide quantification of testis weights corrected for animal body weight (mean {plus minus} standard deviation).

We have modified these data following the advice of the referees and we have included a new graph showing the new data (Figure 2—figure supplement 2B). The sentences have been modified as follows:

“Testes from *Hsf2bp*^S167L/S167L^ mice displayed a reduced size (21% reduction compared to WT mice; testis/body weight ratio: S167L 0,26% ± 0,07 (n=12) vs 0,33% ± 0,05 for WT controls (n=14), **p<0,01, Figure 2D and Figure 2—figure supplement 2B) and this reduction was stronger in *Hsf2bp*^-/-^ testes (70% reduction compared to WT, testis/body weight ratio: *Hsf2bp*^-/-^ 0,10% ± 0,005 (n=6) vs 0,33% ± 0,05 for WT controls (n=14) **** p<0,001, Figure 2D and Figure 2—figure supplement 2B).”

“Male *Midap^-/-^*mice were infertile, the average size of their testes was severely reduced (76% reduction compared to WT; testis weight/body weight ratio: *Midap^-/-^*0,08% ± 0,004 (n=6) vs 0,33% ± 0,05 for WT controls (n=14), ****p<0.0001, Figure 8B and Figure 2—figure supplement 2B),”

1(F) – Epididymis histology for Hsf2bp-S167L requires quantification. An easier approach would be to present sperm counts using standard methods.

We have performed sperm counts of our S167L mutants following the referees’ advice and we show these data in the new Figure 2G. The new results have been included in the MS: “and a reduction in the number of spermatozoa in the epididymis (3.3x10^6^ in the *Hsf2bp*^S167L/S167L^ mutant vs 4.3x10^6^ in the WT; Figure 2G)”

Scale bars in Figure 1B and 1C should be the same size to facilitate comparison. Also, it is difficult to see follicles, especially in (B). Immunostaining should be performed to identify follicles.

We thank the referees for the comments. We have included new pictures (Figure 2B) where the follicles can be clearly distinguished by standard H&E staining which we are convinced is sufficient to show them clearly and unambiguously. On these new figures the scale bars are the same size, as requested by the reviewers.

1(H). Spreads are not appropriate to quantify apoptotic cells. How was apoptosis monitored? The legend is incorrect: "(H).…spread preparations from Hsf2bp-/- and Hsf2bpS167L/S167L mice." It should be +/+ and S167L/S167L.Figure 1 legend is much too long.

have carried out a TUNEL assay on histological sections (new panel on Figure 2F). The new results have been included in the MS: “Consistent with these results, *Hsf2bp*^S167L/S167L^ males showed increased numbers of meiotic divisions positive for TUNEL staining (Figure 2F)”.

We have also corrected the legend and shortened it by reducing it to a plain description. However, the need to incorporate the number of mice used in each experiment and the new results corresponding to the follicle count, TUNEL and the sperm count (new panels 2C, 2F and 2G) lead to a net increase of 23 words.

Figure 22 (a) – HSF2BP staining intensity -- were graphs showing HSF2BP staining intensity normalized against Sycp3 intensity? Were exposure times constant between images and samples? Please provide more details regarding image acquisition. This comment applies to other figures as well.There is no information about image analysis in the Materials and methods section? How were foci counted? Was background subtracted? Which software? Which parameters?

As recommended by the referees, we have provided more details regarding image acquisition and analysis/software. We would like to mention that chromosome spreads were prepared from mutants and controls at the same time by the same person (matched controls) and that at least two different slides from each genotype were used for direct IF comparison. Slides were not frozen to avoid differences in the background and antigen reactivity. The time acquisition was always constant among slides (different genotypes) for the same marker to allow direct comparison. We have included this new subsection “Image acquisition and analysis”:

**“**Slides were visualized at room temperature using a microscope (Axioplan 2; Carl Zeiss, Inc) with 63 × objectives with an aperture of 1.4 (Carl Zeiss, Inc). […] Secondary antibodies for STED imaging were conjugated to Alexa 555 and 488 (Invitrogen) and the slides were mounted in Prolong Antifade Gold without DAPI.”

2 (A,B). Which test was used, meaning of stars?

All statistical analyses have been carried out using the Two-tailed Welch´s t-test. As the meaning of the asterisks is the same in all graphs of a Figure, we usually add the interpretation only once at the end of the legend for all of them. However, we have now included it on each individual graph.

2 (C). A non-specific band of unknown origin cannot be used as loading control. Please add a proper loading control. The extracts should be from juvenile to interpret the result since wt and mutant do not have the same testicular cell composition.What is the mouse background for the -/- and S167L/S167L mice? Are they the same? Differences in strain background (especially mixed versus inbred) can affect the penetrance of meiotic mutants.

We thank the referee for the comment. We have performed the experiment using juvenile extracts to overcome this limitation (new panel in Figure 3C). We have also included new loading controls using anti-tubulin in the western blots (new panels in Figure 7—figure supplement 1C, Figure 7H and Figure 8H).

We would like to note that all the mutants generated in this work have the same mixed genetic background Bl6/CBA.

Figure 3H2AX plots Y axis: add γ

Thanks for the comment, we have added the corresponding symbol (Figure 4A).

Figure 4(a) – right hand side graph is missing a y-axis label. Labels should be more explicit, e.g. "number of DMC1 foci".(b) In the right graph there is extraneous text ("S167L") under the data points.

Thanks for the suggestions and remarks and apologies for the mistakes, we have corrected them and used more explicit labels.

Figure 5(a) The evidence for XY univalent is weak. One may indeed consider the long univalent axis as X, but the smaller one could be any chromosome. This is only a deduction (On Figure 1G, impossible to tell). This observation needs further analysis: is the recombination defect (CO defect and univalent) mostly impacting the X and Y? Please, analyze the distributions of MLH1 foci, specifically the fraction of pachytene SCs that lack MLH1 foci.

We thank the referees. We have included new analyses to address this point:

1) We have quantified the percentage of pachytene spermatocytes showing unsynapsed XY chromosomes by SYCP3 and γH2AX (sex body) labelling (Figure 6D) to assure the designation.

2) We have also carried out the quantification of the number of pachynemas showing XY/autosomic bivalents without COs (MLH1 staining) (new Figure 6B and Figure 6—figure supplement 1B).

These analyses show that in *Hsf2bp*^S167L/S167L^ males appear a significant number of spermatocytes with bivalents without CO (that will be eliminated) and this impacts more on the XY. The new data were included in the text: “and a decreased number of COs (measured as MLH1, Figure 6b and Figure 6—figure supplement 1B). In accordance with the reduction of COs, we observed the presence of univalents in the XY pair at pachynema as well as univalents in metaphase I spermatocytes from *Hsf2bp^S167L/S167L^* mice (Figure 6D and Figure 6—figure supplement 1C).

(B,C) – Please comment on the observation that RPA counts are similar between control and mutant, while DMC1 counts are significantly lower.

We have incorporated a new section in the refocused Discussion section about RPA and recombinases. Specifically, we have included de following text dealing with the sex differences in RPA foci: “Given the unknown function that RPA plays in vivo during oogenesis (Shi et al., 2019), it is tempting to speculate that the role of RPA in mediating the replacement of RAD51/DMC1 in female meiosis would be carried out by another protein complex such as SPATA22/MEIOB in a HSF2BP/MIDAP-dependent manner.”

RAD51 counts should be analyzed in S167L females.SPATA22 which appears to give a strong phenotype in MIDAP mutant should be tested in male and female Hsf2bp mutants.

We thank the referees’ comments. We have now included new counts of RAD51 foci (included in the new Figure 5D and Figure 5—figure supplement 1D) as well as SPATA22 analysis in females (included in the new Figure 6A and Figure 6—figure supplement 1A). The results for SPATA22 in male *Hsf2bp* mutants were already shown in former Figure S18 and have been moved to main Figure 6A and Figure 6—figure supplement 1A.

Identification and analysis of MIDAPThe acronym MIDAP is not the best choice: It's unclear how MIDAP is an acronym for "meiotic double- stranded break BRCA2/HSF2BP complex associated protein". Where is the "I" from, and how is BRCA2/HSF2BP part of the acronym? What is the expression profile of MIDAP? And if the gene is not meiotic-specific it should not be labeled as such. Moreover, it's not clear that its major function is through BRCA2. Please indicate that C19ORF57 is the name of the human not the mouse gene (thus in the yeast two hybrid screen the gene identified was not C19ORF57; was the cDNA overlapping with the domain identified as interacting by CoIP?).

We have indicated in the Y2H screen the name of the mouse RIKEN following referees’ advice indicating that C19ORF57 is the name of the human ORF.

The name of MIDAP was initially proposed and strongly suggested by the gene nomenclature committee (Dr. Ruth Seal <hgnc@genenames.org>). After receiving the reviewing report of the present MS, we have contacted again with the HGNC committee. To date, we have not received a definitive response to our query asking for a final name for the gene C19ORF57/4930432K21Rik. We are thus moving forward with the approved name decided by the authorized HGNC committee.

In relation with the Y2H analysis, all the clones identified and sequenced overlapped with the domain identified by CoIP. We only sequenced the full insert of a fraction of the positive clones and these included the domain identified in the CoIP. The remaining positive clones were only partially sequenced (due to their length) to determine the identity of the gene.

One learns very little from the study of the MIDAP internal deletion. Thus, this could be removed or added as supplementary information. In addition, this mutant allele does not show that the interaction domain is sufficient as the mutant protein is much larger than the interaction domain.Midap∆/∆: we would suggest adding a number such as ∆6. Otherwise, it can be easily thought that this allele is a full deletion.

We thank the referees for the comments. We have shortened this section and included a supplementary figure (Figure 7—figure supplement 1). We have also changed the nomenclature of the mutant to *Midap*^∆142-472/∆142-472^ to avoid any misunderstanding. It is clear that the mutant δ allele (lacking from 142 to 472) does include the interaction domain and also the N-terminal domain (1 to 141). Thus, one cannot formally prove that the interaction domain is sufficient for MIDAP function in vivo from this mutant. However, i) the unstructured nature of the MIDAP protein, ii) the lack of recognizable domains, iii) the lack of meiotic phenotype of the *Midap*^∆142-472/∆142-472^ , iv) the retained ability of interaction of the MIDAP-∆142-472 with HSF2BP,v) and the relatively proper loading of HSF2BP in the *Midap*^∆142-472/∆142-472^spermatocytes suggest that MIDAP carry out most of its meiotic function through a minimum portion of its sequence (270 residues out of the 600 residues of the wild type protein) which includes its interacting domain with HSF2BP (125 residues).

This part is now as follows:

“We mapped the HSF2BP/MIDAP-interacting domain to the C-term fragment of MIDAP (spanning residues 475-600 of the murine protein, Figure 7C). In line with this, the *Midap*^Δ142-472/Δ142-472^ mutant mice, expressing the MIDAP protein devoid of its central part, were fertile and did not show defects in chromosome synapsis or an alteration of HSF2BP loading to axes, further indicating that a large fraction of the coding protein of MIDAP is not essential for MIDAP/HSF2BP function in vivo (Figure 7—figure supplement 1)*.”*

Figure 6:(A): Add MW markers (and check this for all western blots)(C) – IP with Δ475-600 – please provide evidence that IP worked – western blot for protein used for setting up IP reaction must be also provided in all cases. Which fraction of input, of IP was loaded? (these comments apply to all CoIPs).

Following the advice of the referees, we have included in all the western blots the MW markers. We have also included in all the co-IPs the lane demonstrating that the IP works as well as the % of input loaded.

(D) – The authors cannot conclude on a perfect colocalization without quantification. Please quantify. Squares indicating areas being magnified are not being reproduced properly. Also, these areas are not magnified very much- please make them larger! The insets are shifted, this should be mentioned.

We have included the corresponding quantification of the colocalization between HSF2BP and MIDAP (Supplementary file 1D) and following the recommendations of the referees we have made the insets larger and properly reproduced. We have mentioned that the insets are shifted (Figure 7D-E).

Figure 7. Depletion of follicle pool in Midap-/- females should be quantified.Males: provide evidence for the stage IV arrest; in the text explain what is stage IV for non specialist. Asterisks are barely visible, change color.

As suggested by the referees we have quantified the pool of follicles in *Midap*^-/-^ females (Figure 8A).

-We have also explained the term stage IV of the epithelium cycle (referred to *Hsf2bp*^-/-^ arrest) as follows:

“During mouse spermatogenesis, the twelve stages of the epithelial cycle can be distinguished in seminiferous tubule sections by identifying groups of associated germ cell types (Ahmed and de Rooij, 2009). Following these criteria, the seminiferous epithelium of *Hsf2bp*^-/-^ mice showed a stage IV arrest, characterized by a massive apoptosis of zygotene-like spermatocytes occurring at the same time that in spermatogonia divide into B spermatogonia (Figure 2E)”

The color of asterisks has been changed to red following the editor/reviewer’s advice.

(E): 5 nuclei are shown. The four upper nuclei show various levels of synapsis defects. It is not clear what information is provided with the nucleus in the lower part which does not seem to be a pachytene but more likely a diplotene nucleus. Unless a specific conclusion is reached from this nucleus, it could be removed.

We have removed the non-informative nuclei from this figure. Additionally, we have reduced the panel of SYCP1 and SYCP3 (in oocytes) in the main Figure 8E and we have added a supplementary Figure 8—figure supplement 1E showing the remaining nuclei.

Figure 8:(A) – Absence of HSF2BP staining in the Midap-/- mutant could be due to protein destabilization. See comment above; legend, Quantification is HSF2BP foci not MIDAP.

We have carried out western blot analysis of HSF2BP in *Midap*^-/-^ 13 dpp testis extracts to determine whether the absence of HSF2BP labelling in the chromosome axes is due to the absence of MIDAP or due to a lower protein expression/stabilization. Our new WB results show that in *Midap* null mutant testis extracts there is also (as with the IF analysis) a strong reduction of HSF2BP protein expression suggesting lower protein stabilization. These results have been included in the manuscript: “Western blot analysis of HSF2BP in 13 dpp testis extracts from *Midap^-/-^* mice showed a strong reduction in comparison with the WT control (Figure 8H), suggesting again that MIDAP is necessary for HSF2BP protein stabilization.

Figure 9. The absence of MIDAP foci should also be quantified in the Hsf2bp null mutant even if the number of foci is close to 0.

We have included the requested quantification (Figure 7F). The figures showing these results have been relocated to the Figure 7 following the reviewers’ recommendation (moreover the Figure 7—figure supplement 4A shows the extended panel for MIDAP staining in *Hsf2bp* mutant males).

Figure 10:(A) – and in general and for all western blots, the labeling of IPs is confusing andambiguous. It is unclear what the arrows are supposed to highlight. Best is to indicate which antibody was used to probe the blots, and to add arrows to identify proteins when required.The comparison of Hsf2bp wt and S167L does not make sense in such overexpression system since the property of S167L reported above was a potential lower stability and/or level. Transfection with Hsf2bpS167L should be removed.

We have modified the WB labelling for the IPs. We agree with this consideration and accordingly we have removed the triple transfection with the Hsf2bp-S167L as suggested.

Overall, the information gained is the CoIP between the three proteins, the nuclear localization and stabilization of Hsf2Bp in the presence of MIDAP. Note that there is no evidence for a multimeric complex from this study. This would require performing sequential IP.The one and half page paragraph should be reduced to half a page, which would improve the quality of the paper.The legend should be reduced.

Regarding to the absence of evidence for a multimeric complex between BRCA2, MIDAP and HSF2BP, we consider that the sequential IP is not necessary (see the reasoning below) since triple transfections is equivalent in this case. We have provided direct evidence that this complex is indeed formed from the double transfections and triple transfection experiments in HEK293 cell:

Double transfections:

- BRCA2-C + HSF2BP: BRCA2 Co-IP with HSF2BP (Figure 6—figure supplement 2A).

- HSF2BP + MIDAP: HSF2BP Co-IP with MIDAP (Figure 7A).

- BRCA2 + MIDAP: BRCA2 does not Co-IP with MIDAP (Figure 10A). This negative result is essential for the conclusion (see below).

When we transfect the three proteins (BRCA2, HSF2BP and MIDAP):

HSF2BP co-IP with BRCA2-C (Figure 10B).

HSF2BP co-IP with MIDAP (Figure 10B).

BRCA2-C co-IP with MIDAP (but this Co-IP is not observed when we co-transfect of BRCA2 and MIDAP alone, see Figure 10A).

-Altogether, these Co-IP results can only be explained if the three proteins form a multimeric complex. The sequential IP experiment would be required if all pairwise Co-IPs would have been observed (which is not the case).

Please see (Liu et al., 2014) for a sequential IP example because Bag6 can interact with both gp78 and USP13 (subsection “MIDAP is a novel interactor of HSF2BP” and Figure 4F).

However, we agree that these results were too scattered in different figures which could lead to a difficult interpretation. We have now displayed the single negative co-IPs of BRCA2 with MIDAP together with the Triple co-IP (Figure 10A-B).

We agree that the manuscript would be improved by reducing this paragraph. We have shortened this paragraph (from 535 to 385 words).

In relation with the size of the legend, we have shortened the text. Despite that we have included new panels for clarity (scattered in the initial version) and that we also have incorporated the number of cells analysed and replicates (following the reviewers advice), the final number of words has been shortened from 332 to 286.

Figure 1: Please include the genotypes for the sequenced individuals. From the text, it appears as if the brother is heterozygous, so there does not appear to be a male homozygote to comment on human male fertility. This should be clearly stated early in the manuscript (see comment below).

We agree with the referees that this point should be clearer. We have added the genotypes on the pedigree in Figure 1 (former Figure S1), as requested (and modified the legend accordingly). We have added a sentence to highlight the fact that no homozygous male is available early in the Results section: "Therefore, there was no homozygous males identified in this family, preventing the analysis of the impact of this variant on male fertility."

Figure 2—figure supplement 1(A) legend: "Thin (non-coding) and thick (coding sequences) lines under exons represent the expected transcript derived from wild-type and Hsf2bp edited allele." There are no such lines in panel (A).

We have corrected the mistake.

Figure 7—figure supplement 6 – What stage are these WT spermatocytes and approximate stage-matched mutants? Late zygonema?

We have corrected the panel Figure 7—figure supplement 6B and we have indicated the stage at which the wild type and mutant spermatocytes are. We have also included a new description in the text: “We were able to show that none of the recombination-deficient mutants abrogate MIDAP labelling at zygotene (or the corresponding meiotic stage at which the mutant spermatocytes are arrested), in contrast to its absence of loading in SPO11-deficient mice (Figure 7—figure supplement 6B, left).”

Figure 8—figure supplement 1C. Please show the whole membrane. Any truncated protein detected? Where are the epitopes of the antibody?

We appreciate the referees for the comment, but no truncated protein was detected in the *Midap*-null mutant. The antibodies are rabbit polyclonal and have been raised against the full-length murine protein.

**Author response image 1. sa2fig1:** 

Interpretation and discussion of the Hsf2bpS167L mutantSubsection “Mice with the HSF2BP S167L variant show a partial reduction of fertility” "male and female mice were able to reproduce but females showed a significant reduction in the number and size of litters (Figure 1A) whilst males only showed a slight non-significant reduction in fertility (Figure 1A).This is in agreement with the fertility defects observed in Hsf2bp-/- mice (Figure 1C,E-F and (Brandsma et al., 2019; Zhang et al., 2019)), and suggests that the S167L variant might impact more specifically female fertility."There are several problems with this statement:a) This is not in agreement with the two published papers, but rather the opposite, since the published papers describe a more pronounced defect in males. Thus, this text needs to be changed.

We apologize for this error in this text. The fertility defects of the *Hsf2bp*^S167L/S167L^ and the *Hsf2bp*^-/-^ are not in agreement and they are the opposed (only the meiotic phenotype is in agreement and this is very early in the text to make any mention and is now included in the Discussion section). We have accordingly modified the text in the new revised version of the manuscript: “*Hsf2bp^S167L/S167L^*male and female mice were able to reproduce but females showed a significant reduction in the number of litters (Figure 2A) whilst males only showed a slight non-significant reduction in fertility (Figure 2A), suggesting that the S167L variant impacts murine fertility.”

b) However, importantly also, the authors appear to have a more pronounced phenotype in their female knockouts compared to the published reports. This needs to be discussed, included how the knockout alleles vary for the 3 knockouts that have now been generated.

We have incorporated the description of the 2 additional knockouts and their corresponding references. However, we would like also to highlight that our knockout is phenotypically similar to the one already published by Zhang et al., 2019 (males are arrested and females are subfertile) and stronger than the one published by Brandsma et al., 2019 (males are arrested but females show a non-significant reduction of fertility, only a trend). We have included it in the new manuscript: “However, loss of HSF2BP in female mice showed a milder meiotic phenotype (Zhang et al., 2019 and our own data, see below) and a weak albeit non statistical significant reduction of fertility (Brandsma et al., 2019) despite all of the mutants are nulls though in different genetic backgrounds.”

c) Males with S167L have more variability in litter frequency and size, and so with the small numbers it is difficult to draw a conclusion about fertility. Moreover, testis weights and histology of tubules, which show a clear male defect.

We have also the same impression. However, we have not been able to perform new crosses during the lockdown period because it was strictly forbidden in the animal facility. We have however carried out sperm count to analyse the reduction of the number of spermatozoa. The results show a 25% of reduction which is in the limit to have any physiological consequence in terms of fertility. This aspect has been analysed in more detailed in the Discussion section: “Our failure to detect any significant impact of the variant on male fertility could be due to the high variability in litter frequency and size in our study. […] This can be due to the very much fewer gametes available for fertilization in females in comparison to males but also to molecular differences in the meiotic recombination process in both sexes (Cahoon and Libuda, 2019), as displayed by the absence of RPA accumulation or the more pronounced decrease in DMC1 foci observed in HSF2BP-S167L oocytes.”

Related to this in Subsection “Mice with the HSF2BP S167L variant show a partial reduction of fertility”: "These results suggest that mice bearing the POI-causing variant only partially phenocopy the human disease and fits well with the exacerbated sexual phenotypic dimorphism observed in the Hsf2bp-/- mice which might be less pronounced or absent in humans."As above, no conclusion can be made about human male fertility. The lack of follicle counts/breeding over time in females makes it difficult to conclude about POI in mice. Moreover, what sexual dimorphism in -/- mice is this referring to? Both males and females are completely devoid of gametes in the HSF2BP-null.

We would like to highlight that *Hsf2bp*-null females from our lab and from the other two published works (Zhang et al., 2019 and Brandsma et al., 2019) are able to reproduce and have gametes. However, we agree that the sentences were not very descriptive. We have rephrased the text and reduced it to avoid misinterpretation.

“These results suggest that mice bearing the POI-causing variant only partially phenocopy the human disease.”

We have restricted the sexual dimorphism in the mice to the Discussion section “By contrast, female mice with the HSF2BP-S167L variant show a mild sub-fertility phenotype with a reduction of litter frequency. This can be due to the very much fewer gametes available for fertilization in females in comparison to males but also to molecular differences in the meiotic recombination process in both sexes (Cahoon & Libuda, 2019), as displayed by the absence of RPA accumulation or the more pronounced decrease in DMC1 foci observed in HSF2BP-S167L oocytes.”

We have also addressed the comparison between mouse and human infertility in a new restructured Discussion section. “Although we cannot exclude that […]. This reinforces a conserved function of HSF2BP in human male fertility.”

Subsection “Mice with the HSF2BP S167L variant show a partial reduction of fertility”: "Overall, male and female Hsf2bpS167L/S167L mice show similar molecular alterations although with different severity".There are obvious differences in molecular alterations between males and females:- There is a dramatic increase in RPA foci in males during pachynema while no difference in RPA foci in females during pachynema.- DMC1 is virtually unchanged in males (perhaps slight reduction in EZ), but greatly reduced in females.- MLH1 is reduced 20-30% in males, but only ~5% (and not significant) in females.- γ-H2AX is greatly increased during pachynema in males, but these data ds not exist for females.- RAD51 is decreased through zygonema to early pachynema with the biggest decrease seen in late zygonema, but these data don't exist for females.- The only similarity is with DMC1, but even this is misleading as the defect in males is quite mild, but quite severe in females.

We have reconsidered this sentence following the reviewers’ advice. Accordingly, we have changed the statement to: “Overall, male and female *Hsf2bp*^S167L/S167L^ mice share alterations in the meiotic recombination pathway although with different reproductive outcome.”

With the new experiments carried out, the meiotic alterations and their comparison between males and females and between genotypes has been summarized into a new Supplementary file 1f. The emerging picture make use of the expanded analysis (including now MLH1/CDK2 and the new RAD51 and SPATA22 analysis) show a similar trend in males and females for all the markers in all the mutants though with different intensity, with the exception of the RPA accumulation in males (HSF2BP KO=MIDAP KO>HSF2BPS167L) but not in females. Please see the new Supplementary file 1F. This has been addressed specifically in the Discussion section (as mentioned above).

Supplementary file 1G, Supplementary file 1HWhat are the two antibodies (Ab1 and Ab2) used for mass spec? What is the control sample (IP with IgG?). The Materials and methods section indicates the IP was performed with two antibodies R1 and R2 (against HSF2BP and MIDAP)? This is very confusing.Please can you specify how many peptides of MIDAP identified in this screen.

We thank the referees for the comment. We have immunized two independent rabbits with HSF2BP (HSF2BP-R1 and HSF2BP-R2) and two additional rabbits with MIDAP (MIDAP-R1 and MIDAP-R2). For the mass spec analysis, we performed two IP (one with MIDAP-R1 and another with MIDAP-R2). The control sample was IgGs from rabbit. We have explained this point as follows:

Subsection “Immunocytology and antibodies”:

“Rabbit polyclonal antibodies against HSF2BP and MIDAP were developed by ProteintechTM against a fusion protein of poly-His with full length HSF2BP or MIDAP (pUC57 vector) of mouse origin. Two antibodies (named as R1 and R2) were generated against each protein (HSF2BP-R1/2 or MIDAP-R1/2) by immunization of two different host rabbits.”

Subsection “Testis immunoprecipitation coupled to mass spectrometry analysis”:

“200 µg of antibodies R1 and R2 against MIDAP (two independent IPs) and IgG from rabbit (negative control) were crosslinked to 100 ul of sepharose beads slurry (GE Healthcare).”

The number of peptides identified for MIDAP (and all the putative interactors founded) are included in Supplementary files 1G and 1H.

With the large number of figures and different mutants examined, a summary table should be provided.

We have included a new table including a brief description of all the figures (Supplementary file 1L). We have included the title and all the panels that appear on each main figure. We have only included the title of the supplementary figures and its cross-relationship with the main figures for simplicity.

Additionally, we have also included a new table showing the comparative alterations found in the different mutants analysed and between genders (Supplementary file 1F) in a visual fashion so that allows for a rapid comparison.

Comments on the Discussion sectionThe discussion is unfocused with no clear link back to the POI of the human family. Also, the data are generally over-interpreted into specific molecular mechanisms. This should be toned down and the authors should clarify when they are speculating.

We have rewritten the Discussion section and toned down the conclusions. We have also paid particular attention to make it clear in the text when we are speculating. In brief, at first, we link our results of the mutant mice of HSF2BP (null and knock-in) to the human POI and discuss why the human HSF2BP-S167L variant is more severe in humans than in mouse attending to the fertility severity. We then discuss how the fertility defects in the mutant mice can be explained by the role of HSF2BP, together with its new identified partner MIDAP, in meiotic DSB repair and their relationship with other meiotic players involved in the recombination pathway. Finally, we speculate how HSF2BP gene variants, one of them recently known to modify human meiotic recombination rate genome wide (Halldorson et al., 2019), could also cause infertility by decreasing the recombination rate much below the lower limit of one CO per bivalent.

Where is the evidence for a supramolecular complex? In the absence of validation this should also be removed from the Abstract ("the higher order macromolecular structure").

We have removed this concept from the abstract given its speculative nature which is based on the fact that the proteins identified in the IP coupled to mass spec experiments have in common its known interaction with BRCA2 and the lack of direct interaction with MIDAP.

Where is the evidence for direct interaction between HSF2BP and BRCA2 and MIDAP ?

The evidence for the direct interaction between HSF2BP and BRCA2 comes from previous work (Brandsma et al., 2019) and also by our co-IP experiments (Figure 6—figure supplement 2A). Our evidence of direct interaction between HSF2BP and MIDAP comes from our experiments (Y2H, co-IP Figure 7A and testis IP Figure 7B). We have not detected direct interaction between MIDAP and BRCA2 (Figure 10A). However, we have observed that BRCA2 and MIDAP co-immunoprecipitates (Figure 10B) when HSF2BP is present.

Where is the evidence that the supramolecular (?) complex participate to the binding of RPA?

We have already replied to this point at the beginning of this question.

The known roles of DSS1 should be introduced noting that meiotic functions are not characterized.

We eliminated any mention relative to DSS1 given that nothing is known about its meiotic function.

"strongly suggesting a parallelism in the early recombination pathway where HSF2BP/MIDAP work" – this is unclear. Please rephrase.

We have restructured this part of the Discussion section following your advice and the comparison with TEX15 mutant is now out of focus after expanding the analysis of RAD51/DMC1 to females showing that foci formation follows the same trend than in males.

"In general, meiotic recombination mutants seem to proceed further in female than in male meiosis perhaps because of the presence of less stringent checkpoints (Hunt and Hassold, 2002)."But also, recombination mutants often show more severity in males due to X-Y asynapsis.

We agree and appreciate the comment since it is very well known the role that X-Y asynapsis play in the severity of the meiotic arrest in males. We have included this concept in the text: “In general, meiotic recombination mutants appear to proceed further in female than in male, because in males the asynapsis of the sex bivalent leads to a loss of silencing of the Y chromosome, and perhaps also because of the presence of less stringent checkpoints in oogenesis (Hunt and Hassold, 2002).”

"importantly also of HSF2BP" – please rephrase.

This has been corrected.